# Constrained Update Projection Approach to Safe Policy Optimization

Long Yang[1,2,*], Jiaming Ji[1,*], Juntao Dai[1], Linrui Zhang[3], Binbin Zhou,[4] Pengfei Li,[1] Yaodong Yang[2,5], Gang Pan[1,†]

[1]College of Computer Science and Technology, Zhejiang University, China
[2] School of Artificial Intelligence, Peking University, China
[3] Tsinghua Shenzhen International Graduate School, Tsinghua University, China
[4] Department of Computer Science and Computing, Zhejiang University City College, China
[5] Institute for Artificial Intelligence, Peking University & BIGAI, China
yanglong001@pku.edu.cn, gpan@zju.edu.cn

## Abstract

Safe reinforcement learning (RL) studies problems where an intelligent agent has to not only maximize reward but also avoid exploring unsafe areas. In this study, we propose CUP, a novel policy optimization method based on **C**onstrained **U**pdate **P**rojection framework that enjoys rigorous safety guarantee. Central to our CUP development is the newly proposed surrogate functions along with the performance bound. Compared to previous safe reinforcement learning methods, CUP enjoys the benefits of 1) CUP generalizes the surrogate functions to generalized advantage estimator (GAE), leading to strong empirical performance. 2) CUP unifies performance bounds, providing a better understanding and interpretability for some existing algorithms; 3) CUP provides a non-convex implementation via only first-order optimizers, which does not require any strong approximation on the convexity of the objectives. To validate our CUP method, we compared CUP against a comprehensive list of safe RL baselines on a wide range of tasks. Experiments show the effectiveness of CUP both in terms of reward and safety constraint satisfaction. We have opened the source code of CUP at https://github.com/zmsn-2077/CUP-safe-rl.

## 1 Introduction

Reinforcement learning (RL) [Sutton and Barto, 1998] has achieved significant successes in many fields (e.g., [Mnih *et al.*, 2015; Silver *et al.*, 2017; OpenAI, 2019; Afsar *et al.*, 2021; Yang *et al.*, 2022]). However, most RL algorithms improve the performance under the assumption that an agent is free to explore any behaviors. In real-world applications, only considering return maximization is not enough, and we also need to consider safe behaviors. For example, a robot agent should avoid playing actions that irrevocably harm its hardware, and a recommender system should avoid presenting offending items to users. Thus, it is crucial to consider *safe exploration* for RL, which is usually formulated as constrained Markov decision processes (CMDP) [Altman, 1999].

It is challenging to solve CMDP since traditional approaches (e.g., Q-learning [Watkins, 1989] & policy gradient [Williams, 1992]) usually violate the safe exploration constraints, which is undesirable for safe RL. Recently, Achiam *et al.* [2017]; Yang *et al.* [2020]; Bharadhwaj *et al.* [2021] suggest to use some surrogate functions to replace the objective and constraints. However, their implementations involve some convex approximations to the non-convex objective and safe constraints, which leads to

---

*L.Yang and J.Ji share equal contributions. † G.Pan is the corresponding author.

36th Conference on Neural Information Processing Systems (NeurIPS 2022).

many error sources and troubles. Concretely, Achiam *et al.* [2017]; Yang *et al.* [2020]; Bharadhwaj *et al.* [2021] approximate the non-convex objective (or constraints) with first-order or second Taylor expansion, but their implementations still lack a theory to show the error difference between the original objective (or constraints) and its convex approximations. Besides, their approaches involve the inverse of a high-dimension inverse Fisher information matrix, which causes their algorithms require a costly computation for each update when solving high-dimensional RL problems.

To address the above problems, we propose the *constrained update projection* (CUP) algorithm with a theoretical safety guarantee. We derive the CUP bases on the newly proposed surrogate functions with respect to objectives and safety constraints, and provide a practical implementation of CUP that does not depend on any convex approximation to adapt high-dimensional safe RL.

Concretely, in Section 3, Theorem 1 shows generalized difference bounds between two arbitrary policies for the objective and constraints. Those bounds provide principled approximations to the objective and constraints, which are theoretical foundations for us to use those bounds as surrogate functions to replace objective and constraints to design algorithms. Although using difference bounds as surrogate functions to replace the objective has appeared in previous works (e.g., [Schulman *et al.*, 2015; Achiam *et al.*, 2017]), Theorem 1 refines those bounds (or surrogate functions) at least two aspects: **(i)** Firstly, our rigorous theoretical analysis shows a bound with respect to generalized advantage estimator (GAE) [Schulman *et al.*, 2016]. GAE significantly reduces variance empirically while maintaining a tolerable level of bias, the proposed bound involves GAE is one of the critical steps for us to design efficient algorithms. **(ii)** Our new bounds unify the classic result of CPO Achiam *et al.* [2017], i.e., the classic performance bound of CPO is a special case of our bounds. Although existing work (e.g., Zhang *et al.* [2020]; Kang *et al.* [2021]) has applied the key idea of CPO with GAE to solve safe RL problems, their approaches are all empirical and lack a theoretical analysis. Thus, the proposed newly bound partially explains the effectiveness of the above safe RL algorithms. Finally, we should emphasize that although GAE has been empirically applied to extensive reinforcement learning tasks, this work is the first to show a rigorous theoretical analysis to extend the surrogate functions with respect to GAE.

In Section 4, we provide the necessary details of the proposed CUP. The CUP contains two steps: it first performs a policy improvement, which may produce a temporary policy violates the constraint. Then in the second step, CUP projects the policy back onto the safe region to reconcile the constraint violation. Theorem 2 shows the worst-case performance degradation guarantee and approximate satisfaction of safety constraints of CUP, result shows that with a relatively small parameter that controls the penalty of the distance between the old policy and current policy, CUP shares a desirable toleration for both policy improvements and safety constraints. Furthermore, we provide a practical implementation of sample-based CUP. This implementation allows us to use deep neural networks to train a model, which is an efficient iteration without strongly convex approximation of the objective or constraints (e.g., [Achiam *et al.*, 2017; Yang *et al.*, 2020]), and it optimizes the policy according to the first-order optimizer. Finally, extensive high-dimensional experiments on continuous control tasks show the effectiveness of CUP where the agent satisfies safe constraints.

## 2 Preliminaries

Reinforcement learning (RL) [Sutton and Barto, 1998] is often formulated as a *Markov decision process* (MDP) [Puterman, 2014] that is a tuple $\mathcal{M} = (\mathcal{S}, \mathcal{A}, \mathbb{P}, r, \rho_0, \gamma)$. Here $\mathcal{S}$ is state space, $\mathcal{A}$ is action space. $\mathbb{P}(s'|s, a)$ is probability of state transition from $s$ to $s'$ after playing $a$. $r(\cdot) : \mathcal{S} \times \mathcal{S} \times \mathcal{A} \to \mathbb{R}$, and $r(s'|s, a)$ denotes the reward that the agent observes when state transition from $s$ to $s'$ after it plays $a$. $\rho_0(\cdot) : \mathcal{S} \to [0, 1]$ is the initial state distribution and $\gamma \in (0, 1)$.

A stationary parameterized policy $\pi_{\boldsymbol{\theta}}$ is a probability distribution defined on $\mathcal{S} \times \mathcal{A}$, $\pi_{\boldsymbol{\theta}}(a|s)$ denotes the probability of playing $a$ in state $s$. We use $\Pi_{\boldsymbol{\theta}}$ to denote the set of all stationary policies, where $\Pi_{\boldsymbol{\theta}} = \{\pi_{\boldsymbol{\theta}} : \boldsymbol{\theta} \in \mathbb{R}^p\}$, and $\boldsymbol{\theta}$ is a parameter needed to be learned. Let $\mathbf{P}_{\pi_{\boldsymbol{\theta}}} \in \mathbb{R}^{|\mathcal{S}| \times |\mathcal{S}|}$ be a state transition probability matrix, and their components are: $\mathbf{P}_{\pi_{\boldsymbol{\theta}}}[s, s'] = \sum_{a \in \mathcal{A}} \pi_{\boldsymbol{\theta}}(a|s)\mathbb{P}(s'|s, a) =: \mathbb{P}_{\pi_{\boldsymbol{\theta}}}(s'|s)$, which denotes one-step state transformation probability from $s$ to $s'$ by executing $\pi_{\boldsymbol{\theta}}$. Let $\tau = \{s_t, a_t, r_{t+1}\}_{t \geq 0} \sim \pi_{\boldsymbol{\theta}}$ be a trajectory generated by $\pi_{\boldsymbol{\theta}}$, where $s_0 \sim \rho_0(\cdot)$, $a_t \sim \pi_{\boldsymbol{\theta}}(\cdot|s_t)$, $s_{t+1} \sim \mathbb{P}(\cdot|s_t, a_t)$, and $r_{t+1} = r(s_{t+1}|s_t, a_t)$. We use $\mathbb{P}_{\pi_{\boldsymbol{\theta}}}(s_t = s'|s)$ to denote the probability of visiting the state $s'$ after $t$ time steps from the state $s$ by executing $\pi_{\boldsymbol{\theta}}$. Due to the Markov property in

MDP, $\mathbb{P}_{\pi_{\boldsymbol{\theta}}}(s_t = s^{'}|s)$ is $(s, s^{'})$-th component of the matrix $\mathbf{P}_{\pi_{\boldsymbol{\theta}}}^t$, i.e., $\mathbb{P}_{\pi_{\boldsymbol{\theta}}}(s_t = s^{'}|s) = \mathbf{P}_{\pi_{\boldsymbol{\theta}}}^t[s, s^{'}]$. Finally, let $d_{\pi_{\boldsymbol{\theta}}}^{s_0}(s) = (1 - \gamma) \sum_{t=0}^{\infty} \gamma^t \mathbb{P}_{\pi_{\boldsymbol{\theta}}}(s_t = s|s_0)$ be the stationary state distribution of the Markov chain (starting at $s_0$) induced by policy $\pi_{\boldsymbol{\theta}}$. We define $d_{\pi_{\boldsymbol{\theta}}}^{\rho_0}(s) = \mathbb{E}_{s_0 \sim \rho_0(\cdot)}[d_{\pi_{\boldsymbol{\theta}}}^{s_0}(s)]$ as the discounted state visitation distribution on initial distribution $\rho_0(\cdot)$.

The *state value function* of $\pi_{\boldsymbol{\theta}}$ is defined as $V_{\pi_{\boldsymbol{\theta}}}(s) = \mathbb{E}_{\pi_{\boldsymbol{\theta}}}[\sum_{t=0}^{\infty} \gamma^t r_{t+1}|s_0 = s]$, where $\mathbb{E}_{\pi_{\boldsymbol{\theta}}}[\cdot|\cdot]$ denotes a conditional expectation on actions which are selected by $\pi_{\boldsymbol{\theta}}$. Its *state-action value function* is $Q_{\pi_{\boldsymbol{\theta}}}(s, a) = \mathbb{E}_{\pi_{\boldsymbol{\theta}}}[\sum_{t=0}^{\infty} \gamma^t r_{t+1}|s_0 = s, a_0 = a]$, and advantage function is $A_{\pi_{\boldsymbol{\theta}}}(s, a) = Q_{\pi_{\boldsymbol{\theta}}}(s, a) - V_{\pi_{\boldsymbol{\theta}}}(s)$. The goal of reinforcement learning is to maximize $J(\pi_{\boldsymbol{\theta}}) = \mathbb{E}_{s \sim \rho_0(\cdot)}[V_{\pi_{\boldsymbol{\theta}}}(s)]$.

## 2.1 Policy Gradient and Generalized Advantage Estimator (GAE)

Policy gradient [Williams, 1992; Sutton *et al.*, 2000] is widely used to solve policy optimization, which maximizes the expected total reward by repeatedly estimating the gradient $g = \nabla J(\pi_{\boldsymbol{\theta}})$. Schulman *et al.* [2016] summarize several different related expressions for the policy gradient:

$$g = \nabla J(\pi_{\boldsymbol{\theta}}) = \mathbb{E}\left[\sum_{t=0}^{\infty} \Psi_t \nabla \log \pi_{\boldsymbol{\theta}}(a_t|s_t)\right], \tag{1}$$

where $\Psi_t$ can be total discounted reward of the trajectory, value function, advantage function or temporal difference (TD) error. As stated by Schulman *et al.* [2016], the choice $\Psi_t = A(s_t, a_t)$ yields almost the lowest possible variance, which is consistent with the theoretical analysis [Greensmith *et al.*, 2004; Wu *et al.*, 2018]. Furthermore, Schulman *et al.* [2016] propose generalized advantage estimator (GAE) $\hat{A}_t^{\texttt{GAE}(\gamma,\lambda)}(s_t, a_t)$ to replace $\Psi_t$: for any $\lambda \in [0, 1]$,

$$\hat{A}_t^{\texttt{GAE}(\gamma,\lambda)}(s_t, a_t) = \sum_{\ell=0}^{\infty} (\gamma\lambda)^{\ell} \delta_{t+\ell}^V, \tag{2}$$

where $\delta_t^V = r_{t+1} + \gamma V(s_{t+1}) - V(s_t)$ is TD error, and $V(\cdot)$ is an estimator of value function. GAE is an efficient technique for data efficiency and reliable performance of reinforcement learning.

## 2.2 Safe Reinforcement Learning

Safe RL is often formulated as a constrained MDP (CMDP) $\mathcal{M} \cup \mathcal{C}$ [Altman, 1999], which is a standard MDP $\mathcal{M}$ augmented with an additional constraint set $\mathcal{C}$. The set $\mathcal{C} = \{(c_i, b_i)\}_{i=1}^{m}$, where $c_i$ are cost functions: $c_i : \mathcal{S} \times \mathcal{A} \to \mathbb{R}$, and limits are $b_i, i = 1, \cdot, m$. The *cost-return* is defined as: $J^{c_i}(\pi_{\boldsymbol{\theta}}) = \mathbb{E}_{\pi_{\boldsymbol{\theta}}}[\sum_{t=0}^{\infty} \gamma^t c_i(s_t, a_t)]$, then we define the feasible policy set $\Pi_{\mathcal{C}}$ as: $\Pi_{\mathcal{C}} = \cap_{i=1}^{m} \{\pi_{\boldsymbol{\theta}} \in \Pi_{\boldsymbol{\theta}}$ and $J^{c_i}(\pi_{\boldsymbol{\theta}}) \leq b_i\}$. The goal of CMDP is to search the optimal policy $\pi_{\star}$:

$$\pi_{\star} = \arg \max_{\pi_{\boldsymbol{\theta}} \in \Pi_{\mathcal{C}}} J(\pi_{\boldsymbol{\theta}}). \tag{3}$$

Furthermore, we define value functions, action-value functions, and advantage functions for the auxiliary costs in analogy to $V_{\pi_{\boldsymbol{\theta}}}, Q_{\pi_{\boldsymbol{\theta}}}$, and $A_{\pi_{\boldsymbol{\theta}}}$, with $c_i$ replacing $r$ respectively, we denote them as $V_{\pi_{\boldsymbol{\theta}}}^{c_i}, Q_{\pi_{\boldsymbol{\theta}}}^{c_i}$, and $A_{\pi_{\boldsymbol{\theta}}}^{c_i}$. For example, $V_{\pi_{\boldsymbol{\theta}}}^{c_i}(s) = \mathbb{E}_{\pi_{\boldsymbol{\theta}}}[\sum_{t=0}^{\infty} \gamma^t c_i(s_t, a_t)|s_0 = s]$. Without loss of generality, we will restrict our discussion to the case of one constraint with a cost function $c$ and upper bound $b$. Finally, we extend the GAE with respect to auxiliary cost function $c$:

$$\hat{A}_{C,t}^{\texttt{GAE}(\gamma,\lambda)}(s_t, a_t) = \sum_{\ell=0}^{\infty} (\gamma\lambda)^{\ell} \delta_{t+\ell}^C, \tag{4}$$

where $\delta_t^C = r_{t+1} + \gamma C(s_{t+1}) - C(s_t)$ is TD error, and $C(\cdot)$ is an estimator of cost function $c$.

## 3 Generalized Policy Performance Difference Bounds

In this section, we show some generalized policy optimization performance bounds for $J(\pi_{\boldsymbol{\theta}})$ and $J^c(\pi_{\boldsymbol{\theta}})$. The proposed bounds provide some new surrogate functions with respect to the objective and cost function, which are theoretical foundations for us to design efficient algorithms to improve policy

performance and satisfy constraints. Before we present the new performance difference bounds, let us revisit a classic performance difference from Kakade and Langford [2002],

$$J(\pi_{\boldsymbol{\theta}}) - J(\pi_{\boldsymbol{\theta}'}) = (1-\gamma)^{-1}\mathbb{E}_{s\sim d^{\rho_0}_{\pi_{\boldsymbol{\theta}}}(\cdot),a\sim\pi_{\boldsymbol{\theta}}(\cdot|s)}\left[A_{\pi_{\boldsymbol{\theta}'}}(s,a)\right]. \tag{5}$$

Eq.(5) shows a difference between two arbitrary policies $\pi_{\boldsymbol{\theta}}$ and $\pi_{\boldsymbol{\theta}'}$ with different parameters $\boldsymbol{\theta}$ and $\boldsymbol{\theta}'$. According to (5), we rewrite the policy optimization (3) as follows

$$\pi_{\star} = \arg \max_{\pi_{\boldsymbol{\theta}}\in\Pi_{\mathcal{C}}} \mathbb{E}_{s\sim d^{\rho_0}_{\pi_{\boldsymbol{\theta}}}(\cdot),a\sim\pi_{\boldsymbol{\theta}}(\cdot|s)}\left[A_{\pi_{\boldsymbol{\theta}'}}(s,a)\right]. \tag{6}$$

However, Eq.(5) or (6) is very intractable for sampling-based policy optimization since it requires the data comes from the (unknown) policy $\pi_{\boldsymbol{\theta}}$ that needed to be learned. In this section, we provide a bound refines the result (5), which provide the sights for surrogate functions to solve problem (3).

### 3.1 Some Additional Notations

We use a bold lowercase letter to denote a vector, e.g., $\mathbf{a} = (a_1, a_2, \cdots, a_n)$, and its $i$-th element $\mathbf{a}[i] =: a_i$. Let $\varphi(\cdot) : \mathcal{S} \to \mathbb{R}$ be a function defined on $\mathcal{S}$, $\delta^{\varphi}_t = r(s_{t+1}|s_t, a_t) + \gamma\varphi(s_{t+1}) - \varphi(s_t)$ is TD error with respect to $\varphi(\cdot)$. For two arbitrary policies $\pi_{\boldsymbol{\theta}}$ and $\pi_{\boldsymbol{\theta}'}$, we denote $\delta^{\varphi}_{\pi_{\boldsymbol{\theta}},t}(s)$ as the expectation of TD error, and define $\Delta^{\varphi}_t(\pi_{\boldsymbol{\theta}}, \pi_{\boldsymbol{\theta}'}, s)$ as the difference between $\delta^{\varphi}_{\pi_{\boldsymbol{\theta}},t}(s)$ and $\delta^{\varphi}_{\pi_{\boldsymbol{\theta}'},t}(s)$:

$$\delta^{\varphi}_{\pi_{\boldsymbol{\theta}},t}(s) = \mathop{\mathbb{E}}_{\substack{s_t\sim\mathbb{P}_{\pi_{\boldsymbol{\theta}}}(\cdot|s) \\ a_t\sim\pi_{\boldsymbol{\theta}}(\cdot|s_t) \\ s_{t+1}\sim\mathbb{P}(\cdot|s_t,a_t)}} \left[\delta^{\varphi}_t\right], \Delta^{\varphi}_t(\pi_{\boldsymbol{\theta}}, \pi_{\boldsymbol{\theta}'}, s) = \mathop{\mathbb{E}}_{\substack{s_t\sim\mathbb{P}_{\pi_{\boldsymbol{\theta}'}}(\cdot|s) \\ a_t\sim\pi_{\boldsymbol{\theta}'}(\cdot|s_t) \\ s_{t+1}\sim\mathbb{P}(\cdot|s_t,a_t)}} \left[\left(\frac{\pi_{\boldsymbol{\theta}}(a_t|s_t)}{\pi_{\boldsymbol{\theta}'}(a_t|s_t)} - 1\right)\delta^{\varphi}_t\right].$$

Furthermore, we introduce two vectors $\boldsymbol{\delta}^{\varphi}_{\pi_{\boldsymbol{\theta}},t}, \boldsymbol{\Delta}^{\varphi}_t(\pi_{\boldsymbol{\theta}}, \pi_{\boldsymbol{\theta}'}) \in \mathbb{R}^{|\mathcal{S}|}$, and their components are:

$$\boldsymbol{\delta}^{\varphi}_{\pi_{\boldsymbol{\theta}},t}[s] = \delta^{\varphi}_{\pi_{\boldsymbol{\theta}},t}(s), \quad \boldsymbol{\Delta}^{\varphi}_t(\pi_{\boldsymbol{\theta}}, \pi_{\boldsymbol{\theta}'})[s] = \Delta^{\varphi}_t(\pi_{\boldsymbol{\theta}}, \pi_{\boldsymbol{\theta}'}, s). \tag{7}$$

Let matrix $\mathbf{P}^{(\lambda)}_{\pi_{\boldsymbol{\theta}}} = (1-\gamma\lambda)\sum_{t=0}^{\infty}(\gamma\lambda)^t\mathbf{P}^{t+1}_{\pi_{\boldsymbol{\theta}}}$, where $\lambda \in [0,1]$. It is similar to the normalized discounted distribution $d^{\rho_0}_{\pi_{\boldsymbol{\theta}}}(s)$, we extend it to $\lambda$-version and denote it as $d^{\lambda}_{\pi_{\boldsymbol{\theta}}}(s)$:

$$d^{\lambda}_{\pi_{\boldsymbol{\theta}}}(s) = \mathbb{E}_{s_0\sim\rho_0(\cdot)}\left[(1-\tilde{\gamma})\sum_{t=0}^{\infty}\tilde{\gamma}^t\mathbb{P}^{(\lambda)}_{\pi_{\boldsymbol{\theta}}}(s_t = s|s_0)\right],$$

where $\tilde{\gamma} = \frac{\gamma(1-\lambda)}{1-\gamma\lambda}$, the probability $\mathbb{P}^{(\lambda)}_{\pi_{\boldsymbol{\theta}}}(s_t = s|s_0)$ is the $(s_0, s)$-th component of the matrix product $\left(\mathbf{P}^{(\lambda)}_{\pi_{\boldsymbol{\theta}}}\right)^t$. Finally, we introduce a vector $\mathbf{d}^{\lambda}_{\pi_{\boldsymbol{\theta}}} \in \mathbb{R}^{|\mathcal{S}|}$, and its components are: $\mathbf{d}^{\lambda}_{\pi_{\boldsymbol{\theta}}}[s] = d^{\lambda}_{\pi_{\boldsymbol{\theta}}}(s)$.

### 3.2 Main Results

**Theorem 1** (Generalized Policy Performance Difference). *For any function $\varphi(\cdot) : \mathcal{S} \to \mathbb{R}$, for two arbitrary policies $\pi_{\boldsymbol{\theta}}$ and $\pi_{\boldsymbol{\theta}'}$, for any $p, q \in [1, \infty)$ such that $\frac{1}{p} + \frac{1}{q} = 1$, we define two error terms:*

$$\epsilon^{\varphi,(\lambda)}_{p,q,t}(\pi_{\boldsymbol{\theta}}, \pi_{\boldsymbol{\theta}'}) =: \|\mathbf{d}^{\lambda}_{\pi_{\boldsymbol{\theta}}} - \mathbf{d}^{\lambda}_{\pi_{\boldsymbol{\theta}'}}\|_p\|\boldsymbol{\delta}^{\varphi}_{\pi_{\boldsymbol{\theta}},t}\|_q, \tag{8}$$

$$L^{\varphi,\pm}_{p,q}(\pi_{\boldsymbol{\theta}}, \pi_{\boldsymbol{\theta}'}) =: \frac{1}{1-\tilde{\gamma}}\sum_{t=0}^{\infty}\gamma^t\lambda^t\mathbb{E}_{s\sim d^{\lambda}_{\pi_{\boldsymbol{\theta}'}}(\cdot)}\left[\Delta^{\varphi}_t(\pi_{\boldsymbol{\theta}}, \pi_{\boldsymbol{\theta}'}, s) \pm \epsilon^{\varphi,(\lambda)}_{p,q,t}(\pi_{\boldsymbol{\theta}}, \pi_{\boldsymbol{\theta}'})\right]. \tag{9}$$

*Then, the following bound with respect to policy performance difference $J(\pi_{\boldsymbol{\theta}}) - J(\pi_{\boldsymbol{\theta}'})$ holds:*

$$L^{\varphi,-}_{p,q}(\pi_{\boldsymbol{\theta}}, \pi_{\boldsymbol{\theta}'}) \leq J(\pi_{\boldsymbol{\theta}}) - J(\pi_{\boldsymbol{\theta}'}) \leq L^{\varphi,+}_{p,q}(\pi_{\boldsymbol{\theta}}, \pi_{\boldsymbol{\theta}'}). \tag{10}$$

*Proof.* See Appendix E. □

The bound (10) is well-defined, i.e., if $\pi_{\boldsymbol{\theta}} = \pi_{\boldsymbol{\theta}'}$, all the three terms in Eq.(10) are zero identically. From Eq.(9), we know the performance difference bound $L^{\varphi,\pm}_{p,q}(\pi_{\boldsymbol{\theta}}, \pi_{\boldsymbol{\theta}'})$ (10) can be interpreted by two distinct difference parts: **(i)** the first difference part, i.e., the expectation $\Delta^{\varphi}_t(\pi_{\boldsymbol{\theta}}, \pi_{\boldsymbol{\theta}'}, s)$, which

is determined by the difference between TD errors of $\pi_{\boldsymbol{\theta}}$ and $\pi_{\boldsymbol{\theta}'}$; **(ii)** the second difference part, i.e., the discounted distribution difference $\epsilon_{p,q,t}^{\varphi,(\lambda)}(\pi_{\boldsymbol{\theta}},\pi_{\boldsymbol{\theta}'})$, which is determined by the gap between the normalized discounted distribution of $\pi_{\boldsymbol{\theta}}$ and $\pi_{\boldsymbol{\theta}'}$. Thus, the difference of both TD errors and discounted distribution determine the policy difference $J(\pi_{\boldsymbol{\theta}}) - J(\pi_{\boldsymbol{\theta}'})$.

The different choices of $p$ and $q$ lead Eq.(10) to be different bounds. If $p=1, q=\infty$, we denote $\epsilon_{\pi_{\boldsymbol{\theta}},t}^{\varphi} =: \|\boldsymbol{\delta}_{\pi_{\boldsymbol{\theta}},t}^{\varphi}\|_q = \max_{s_t \in \mathcal{S}} \mathbb{E}_{a_t \sim \pi_{\boldsymbol{\theta}}(\cdot|s_t), s_{t+1} \sim \mathbb{P}(\cdot|s_t,a_t)}[|\delta_t^{\varphi}|]$, then, according to Lemma 2 (see Appendix F.2), when $p=1, q=\infty$, then error $\epsilon_{p,q,t}^{\varphi,(\lambda)}(\pi_{\boldsymbol{\theta}},\pi_{\boldsymbol{\theta}'})$ is reduced to:

$$\epsilon_{p,q,t}^{\varphi,(\lambda)}(\pi_{\boldsymbol{\theta}},\pi_{\boldsymbol{\theta}'})\big|_{p=1,q=\infty} \leq \frac{\tilde{\gamma}\left(\gamma\lambda(|\mathcal{S}|-1)+1\right)\epsilon_{\pi_{\boldsymbol{\theta}},t}^{\varphi}}{(1-\tilde{\gamma})(1-\gamma\lambda)}\mathbb{E}_{s\sim d_{\pi_{\boldsymbol{\theta}}}^{\lambda}(\cdot)}\left[2D_{\mathrm{TV}}(\pi_{\boldsymbol{\theta}'},\pi_{\boldsymbol{\theta}})[s]\right],$$

where $D_{\mathrm{TV}}(\pi_{\boldsymbol{\theta}'},\pi_{\boldsymbol{\theta}})[s]$ is the total variational divergence between action distributions at state $s$, i.e.,

$$2D_{\mathrm{TV}}(\pi_{\boldsymbol{\theta}'},\pi_{\boldsymbol{\theta}})[s] = \sum_{a\in\mathcal{A}}|\pi_{\boldsymbol{\theta}'}(a|s) - \pi_{\boldsymbol{\theta}}(a|s)|.$$

Finally, let $\varphi = V_{\pi_{\boldsymbol{\theta}'}}$, the left side of (10) in Theorem 1 implies a lower bound of performance difference, which illustrates the worse case of approximation error, we present it in Proposition 1.

**Proposition 1.** *For any two policies $\pi_{\boldsymbol{\theta}}$ and $\pi_{\boldsymbol{\theta}'}$, let $\epsilon_{\pi_{\boldsymbol{\theta}}}^{V}(\pi_{\boldsymbol{\theta}'}) =: \sup_{t\in\mathbb{N}^+}\{\epsilon_{\pi_{\boldsymbol{\theta}},t}^{\varphi} : \varphi = V_{\pi_{\boldsymbol{\theta}'}}\}$, then*

$$J(\pi_{\boldsymbol{\theta}}) - J(\pi_{\boldsymbol{\theta}'}) \geq \frac{1}{1-\tilde{\gamma}}\mathbb{E}_{s\sim d_{\pi_{\boldsymbol{\theta}'}}^{\lambda}(\cdot), a\sim\pi_{\boldsymbol{\theta}}(\cdot|s)}\left[A_{\pi_{\boldsymbol{\theta}'}}^{\mathtt{GAE}(\gamma,\lambda)}(s,a)\right.$$
$$\left. - \frac{2\tilde{\gamma}\left(\gamma\lambda(|\mathcal{S}|-1)+1\right)\epsilon_{\pi_{\boldsymbol{\theta}}}^{V}(\pi_{\boldsymbol{\theta}'})}{(1-\tilde{\gamma})(1-\gamma\lambda)}D_{\mathrm{TV}}(\pi_{\boldsymbol{\theta}'},\pi_{\boldsymbol{\theta}})[s]\right]. \quad (11)$$

The refined bound (11) contains `GAE` technique that significantly reduces variance while maintaining a tolerable level of bias empirically [Schulman *et al.*, 2016], which implies using the bound (11) as a surrogate function could improve performance potentially for practice. Although `GAE` has been empirically applied to extensive reinforcement learning tasks, to the best of our knowledge, the result (11) is the first to show a rigorous theoretical analysis to extend the surrogate functions to `GAE`.

**Remark 1** (Unification of [Achiam *et al.*, 2017]). *If $\lambda \to 0$, then the distribution $d_{\pi_{\boldsymbol{\theta}'}}^{\lambda}(\cdot)$ is reduced to $d_{\pi_{\boldsymbol{\theta}'}}^{\rho_0}(\cdot)$ and the bound (11) is reduced to*

$$J(\pi_{\boldsymbol{\theta}}) - J(\pi_{\boldsymbol{\theta}'}) \geq \frac{1}{1-\gamma}\mathbb{E}_{s\sim d_{\pi_{\boldsymbol{\theta}'}}^{\rho_0}(\cdot), a\sim\pi_{\boldsymbol{\theta}}(\cdot|s)}\left[A_{\pi_{\boldsymbol{\theta}'}}(s,a) - 2\frac{\gamma}{1-\gamma}\epsilon_{\pi_{\boldsymbol{\theta}}}^{V}(\pi_{\boldsymbol{\theta}'})D_{\mathrm{TV}}(\pi_{\boldsymbol{\theta}'},\pi_{\boldsymbol{\theta}})[s]\right],$$
$$(12)$$

*which matches the result of [Achiam* et al.*, 2017, Corollary 1]. That is to say the proposed bound (11) unifies the classic bound (12)*

Let $\varphi = V_{\pi_{\boldsymbol{\theta}'}}^c$, Theorem 1 implies an upper bound of cost function as presented in the next Proposition 2, we will use it to make guarantee for safe policy optimization.

**Proposition 2.** *For any two policies $\pi_{\boldsymbol{\theta}}$ and $\pi_{\boldsymbol{\theta}'}$, let $\epsilon_{\pi_{\boldsymbol{\theta}}}^{C}(\pi_{\boldsymbol{\theta}'}) =: \sup_{t\in\mathbb{N}^+}\{\epsilon_{\pi_{\boldsymbol{\theta}},t}^{\varphi} : \varphi = V_{\pi_{\boldsymbol{\theta}'}}^c\}$, then*

$$J^c(\pi_{\boldsymbol{\theta}}) - J^c(\pi_{\boldsymbol{\theta}'}) \leq \frac{1}{1-\tilde{\gamma}}\mathbb{E}_{s\sim d_{\pi_{\boldsymbol{\theta}'}}^{\lambda}(\cdot), a\sim\pi_{\boldsymbol{\theta}}(\cdot|s)}\left[A_{\pi_{\boldsymbol{\theta}'},C}^{\mathtt{GAE}(\gamma,\lambda)}(s,a)\right.$$
$$\left. + \frac{2\tilde{\gamma}\left(\gamma\lambda(|\mathcal{S}|-1)+1\right)\epsilon_{\pi_{\boldsymbol{\theta}}}^{C}(\pi_{\boldsymbol{\theta}'})}{(1-\tilde{\gamma})(1-\gamma\lambda)}D_{\mathrm{TV}}(\pi_{\boldsymbol{\theta}'},\pi_{\boldsymbol{\theta}})[s]\right]. \quad (13)$$

*where we calculate $A_{\pi_{\boldsymbol{\theta}'},C}^{\mathtt{GAE}(\gamma,\lambda)}(s,a)$ according to the data sampled from $\pi_{\boldsymbol{\theta}'}$ and the estimator (4).*

All above bound results (11) and (13) can be extended for a total variational divergence to KL-divergence between policies, which are desirable for policy optimization.

**Proposition 3.** *All the bounds in (11) and (13) hold if we make the following substitution:*

$$\mathbb{E}_{s\sim d_{\pi_{\boldsymbol{\theta}'}}^{\lambda}(\cdot)}\left[D_{\mathrm{TV}}(\pi_{\boldsymbol{\theta}'},\pi_{\boldsymbol{\theta}})[s]\right] \leftarrow \sqrt{\frac{1}{2}\mathbb{E}_{s\sim d_{\pi_{\boldsymbol{\theta}'}}^{\lambda}(\cdot)}\left[\mathrm{KL}(\pi_{\boldsymbol{\theta}'},\pi_{\boldsymbol{\theta}})[s]\right]},$$

*where $\mathrm{KL}(\cdot,\cdot)$ is KL-divergence, and $\mathrm{KL}(\pi_{\boldsymbol{\theta}'},\pi_{\boldsymbol{\theta}})[s] = \mathrm{KL}(\pi_{\boldsymbol{\theta}'}(\cdot|s),\pi_{\boldsymbol{\theta}}(\cdot|s))$.*

# 4 CUP: Constrained Update Projection

It is challenging to implement CMDP (3) directly since it requires us to judge whether a proposed policy $\pi_{\boldsymbol{\theta}}$ is in the feasible region $\Pi_{\mathcal{C}}$. According to the bounds in Proposition 1-3, we develop new surrogate functions to replace the objective and constraints. We propose the CUP (constrained update projection) algorithm that is a two-step approach contains *performance improvement* and *projection*. Due to the limitation of space, we present all the details of the implementation in Appendix C and Algorithm 1.

## 4.1 Algorithm

**Step 1: Performance Improvement.** According to Proposition 1 and Proposition 3, for an appropriate coefficient $\alpha_k$, we update policy as:

$$\pi_{\boldsymbol{\theta}_{k+\frac{1}{2}}} = \arg\max_{\pi_{\boldsymbol{\theta}} \in \Pi_{\boldsymbol{\theta}}} \left\{ \mathop{\mathbb{E}}_{\substack{s \sim d^{\lambda}_{\pi_{\boldsymbol{\theta}_k}}(\cdot) \\ a \sim \pi_{\boldsymbol{\theta}_k}(\cdot|s)}} \left[ \frac{\pi_{\boldsymbol{\theta}}(a|s)}{\pi_{\boldsymbol{\theta}_k}(a|s)} A^{\mathtt{GAE}(\gamma,\lambda)}_{\pi_{\boldsymbol{\theta}_k}}(s,a) \right] - \alpha_k \sqrt{\mathbb{E}_{s \sim d^{\lambda}_{\pi_{\boldsymbol{\theta}_k}}(\cdot)} \left[ \mathrm{KL}(\pi_{\boldsymbol{\theta}_k}, \pi_{\boldsymbol{\theta}})[s] \right]} \right\}. \tag{14}$$

This step is a typical minimization-maximization (MM) algorithm [Hunter and Lange, 2004], it includes return maximization and minimization the distance between old policy and new policy. the expectation (14) by sample averages according to the trajectories collected by $\pi_{\boldsymbol{\theta}_k}$.

**Step 2: Projection.** According to Proposition 2 and Proposition 3, for an appropriate coefficient $\beta_k$, we project the policy $\pi_{\boldsymbol{\theta}_{k+\frac{1}{2}}}$ onto the safe constraint set,

$$\pi_{\boldsymbol{\theta}_{k+1}} = \arg\min_{\pi_{\boldsymbol{\theta}} \in \Pi_{\boldsymbol{\theta}}} D\left(\pi_{\boldsymbol{\theta}}, \pi_{\boldsymbol{\theta}_{k+\frac{1}{2}}}\right), \text{ s.t. } C_{\pi_{\boldsymbol{\theta}_k}}(\pi_{\boldsymbol{\theta}}, \beta_k) \leq b, \tag{15}$$

where $D(\cdot, \cdot)$ (e.g., KL divergence or $\ell_2$-norm) measures distance between $\pi_{\boldsymbol{\theta}_{k+\frac{1}{2}}}$ and $\pi_{\boldsymbol{\theta}}$,

$$C_{\pi_{\boldsymbol{\theta}_k}}(\pi_{\boldsymbol{\theta}}, \beta) = J^c(\pi_{\boldsymbol{\theta}_k}) + \frac{1}{1-\tilde{\gamma}} \mathbb{E}_{s \sim d^{\lambda}_{\pi_{\boldsymbol{\theta}_k}}(\cdot), a \sim \pi_{\boldsymbol{\theta}}(\cdot|s)} \left[ A^{\mathtt{GAE}(\gamma,\lambda)}_{\pi_{\boldsymbol{\theta}},C}(s,a) \right] + \beta \sqrt{\mathbb{E}_{s \sim d^{\lambda}_{\pi_{\boldsymbol{\theta}_k}}(\cdot)} \left[ \mathrm{KL}(\pi_{\boldsymbol{\theta}_k}, \pi_{\boldsymbol{\theta}})[s] \right]}.$$

Until now, the particular choice of surrogate function is heuristically motivated, we show the worst-case performance degradation guarantee and approximate satisfaction of safety constraints of CUP in Theorem 2, and its proof is shown in Appendix G.

**Theorem 2.** *Let* $\chi_k = \mathbb{E}_{s \sim d^{\lambda}_{\pi_{\boldsymbol{\theta}_k}}(\cdot)}[\mathrm{KL}(\pi_{\boldsymbol{\theta}_k}, \pi_{\boldsymbol{\theta}_{k+\frac{1}{2}}})[s]]$, $\iota =: \frac{\tilde{\gamma}(\gamma\lambda(|\mathcal{S}|-1)+1)}{(1-\tilde{\gamma})(1-\gamma\lambda)}$.*If* $\pi_{\boldsymbol{\theta}_k}$ *and* $\pi_{\boldsymbol{\theta}_{k+1}}$ *are generated according to (14)-(15), then the lower bound on policy improvement, and upper bound on constraint violation are*

$$J(\pi_{\boldsymbol{\theta}_{k+1}}) - J(\pi_{\boldsymbol{\theta}_k}) \geq -\frac{\iota \alpha_k \sqrt{2\chi_k}}{1-\tilde{\gamma}} \epsilon^V_{\pi_{\boldsymbol{\theta}_{k+1}}}(\pi_{\boldsymbol{\theta}_k}), \quad J^c(\pi_{\boldsymbol{\theta}_{k+1}}) \leq b + \frac{\iota \beta_k \sqrt{2\chi_k}}{1-\tilde{\gamma}} \epsilon^C_{\pi_{\boldsymbol{\theta}_{k+1}}}(\pi_{\boldsymbol{\theta}_k}).$$

**Remark 2** (Asymptotic Safety Guarantee)**.** *Let* $\alpha_k \to 0, \beta_k \to 0$ *as* $k \to \infty$, *Theorem 2 implies a monotonic policy improvement with an asymptotic safety guarantee.*

## 4.2 Practical Implementation

Now, we present our sample-based implementation for CUP (14)-(15). Our main idea is to estimate the objective and constraints in (14)-(15) with samples collected by current policy $\pi_{\boldsymbol{\theta}_k}$, then solving its optimization problem via first-order optimizer.

Let $\{(s_t, a_t, r_{t+1}, c_{t+1})\}_{t=1}^T \sim \pi_{\boldsymbol{\theta}_k}$, we denote the empirical KL-divergence with respect to $\pi_{\boldsymbol{\theta}}$ and $\pi_{\boldsymbol{\theta}'}$ as:

$$\hat{D}_{\mathrm{KL}}(\pi_{\boldsymbol{\theta}}, \pi_{\boldsymbol{\theta}'}) = \frac{1}{T} \sum_{t=1}^T \mathrm{KL}(\pi_{\boldsymbol{\theta}}(a_t|s_t), \pi_{\boldsymbol{\theta}'}(a_t|s_t)).$$

We update performance improvement (14) step as follows,

$$\pi_{\boldsymbol{\theta}_{k+\frac{1}{2}}} = \arg \max_{\pi_{\boldsymbol{\theta}} \in \Pi_{\boldsymbol{\theta}}} \left\{ \frac{1}{T} \sum_{t=1}^{T} \frac{\pi_{\boldsymbol{\theta}}(a_t|s_t)}{\pi_{\boldsymbol{\theta}_k}(a_t|s_t)} \hat{A}_t - \alpha_k \sqrt{\hat{D}_{\mathrm{KL}}(\pi_{\boldsymbol{\theta}_k}, \pi_{\boldsymbol{\theta}})} \right\},$$

where $\hat{A}_t$ is an estimator of $A_{\pi_{\boldsymbol{\theta}_k}}^{\mathtt{GAE}(\gamma,\lambda)}(s,a)$.

Then we update the projection step by replacing the distance $D$ by KL-divergence, the next Theorem 3 (for its proof, see Appendix C.2) provides a fundamental way for us to solve projection step (15).

**Theorem 3.** *The constrained problem (40) is equivalent to the following primal-dual problem:*

$$\max_{\nu \geq 0} \min_{\pi_{\boldsymbol{\theta}} \in \Pi_{\boldsymbol{\theta}}} \left\{ D\left(\pi_{\boldsymbol{\theta}}, \pi_{\boldsymbol{\theta}_{k+\frac{1}{2}}}\right) + \nu \left( C_{\pi_{\boldsymbol{\theta}_k}}(\pi_{\boldsymbol{\theta}}, \beta) - b \right) \right\}.$$

According to Theorem 3, we solve the constraint problem (15) by the following primal-dual approach,

$$(\pi_{\boldsymbol{\theta}_{k+1}}, \nu_{k+1}) = \arg \min_{\pi_{\boldsymbol{\theta}} \in \Pi_{\boldsymbol{\theta}}} \max_{\nu \geq 0} \left\{ \hat{D}_{\mathrm{KL}}(\pi_{\boldsymbol{\theta}_{k+\frac{1}{2}}}, \pi_{\boldsymbol{\theta}}) + \nu \hat{C}(\pi_{\boldsymbol{\theta}}, \pi_{\boldsymbol{\theta}_k}) \right\}$$

where $\hat{C}(\pi_{\boldsymbol{\theta}}, \pi_{\boldsymbol{\theta}_k}) = \hat{J}^C + \frac{1}{1-\tilde{\gamma}} \cdot \frac{1}{T} \sum_{t=1}^{T} \frac{\pi_{\boldsymbol{\theta}}(a_t|s_t)}{\pi_{\boldsymbol{\theta}_k}(a_t|s_t)} \hat{A}_t^C + \beta_k \sqrt{\hat{D}_{\mathrm{KL}}(\pi_{\boldsymbol{\theta}_k}, \pi_{\boldsymbol{\theta}})} - b$, $\hat{J}^C$ and $\hat{A}_t^C$ are estimators for cost-return and cost-advantage.

Finally, let

$$\hat{\mathcal{L}}_{\mathrm{c}}\left(\pi_{\boldsymbol{\theta}}, \pi_{\boldsymbol{\theta}_k}, \boldsymbol{\theta}_{k+\frac{1}{2}}, \nu\right) =: \hat{D}_{\mathrm{KL}}(\pi_{\boldsymbol{\theta}_{k+\frac{1}{2}}}, \pi_{\boldsymbol{\theta}}) + \nu \hat{C}(\pi_{\boldsymbol{\theta}}, \pi_{\boldsymbol{\theta}_k}), \tag{16}$$

we update the parameters $(\boldsymbol{\theta}_{k+1}, \nu_{k+1})$ as follows,

$$\boldsymbol{\theta}_{k+1} \leftarrow \boldsymbol{\theta}_k - \eta \frac{\partial}{\partial \boldsymbol{\theta}} \hat{\mathcal{L}}_{\mathrm{c}}\left(\pi_{\boldsymbol{\theta}}, \pi_{\boldsymbol{\theta}_k}, \boldsymbol{\theta}_{k+\frac{1}{2}}, \nu\right) \Big|_{\boldsymbol{\theta}=\boldsymbol{\theta}_k, \nu=\nu_k}, \tag{17}$$

$$\nu_{k+1} \leftarrow \left\{ \nu_k + \eta \frac{\partial}{\partial \nu} \hat{\mathcal{L}}_{\mathrm{c}}\left(\pi_{\boldsymbol{\theta}}, \pi_{\boldsymbol{\theta}_k}, \boldsymbol{\theta}_{k+\frac{1}{2}}, \nu\right) \Big|_{\boldsymbol{\theta}=\boldsymbol{\theta}_k, \nu=\nu_k} \right\}_+, \tag{18}$$

where $\eta > 0$ is step-size, $\{\cdot\}_+$ denotes the positive part, i.e., if $x \leq 0$, $\{x\}_+ = 0$, else $\{x\}_+ = x$. We have shown all the details of the implementation in Algorithm 1.

# 5 Related Work

Due to the limitation of space, for more discussions and comparisons, see Appendix B and Table 2.

## 5.1 Local Policy Search and Lagrangian Approach

A direct way to solve CMDP (3) is to apply *local policy search* [Peters and Schaal, 2008; Pirotta *et al.*, 2013] over the policy space $\Pi_{\mathcal{C}}$, i.e.,

$$\pi_{\boldsymbol{\theta}_{k+1}} = \arg \max_{\pi_{\boldsymbol{\theta}} \in \Pi_{\boldsymbol{\theta}}} J(\pi_{\boldsymbol{\theta}}), \text{ s.t. } J^c(\pi_{\boldsymbol{\theta}}) \leq b, \text{ and } D(\pi_{\boldsymbol{\theta}}, \pi_{\boldsymbol{\theta}_k}) < \delta, \tag{19}$$

where $\delta$ is a positive scalar, $D(\cdot, \cdot)$ is some distance measure. For practice, the local policy search (19) is challenging to implement because it requires evaluation of the constraint function $c$ to determine whether a proposed point $\pi$ is feasible [Zhang *et al.*, 2020]. Besides, Li and Belta [2019]; Cheng *et al.* [2019]; Liu *et al.* [2020] provide a local policy search via the barrier function. The key idea of the proposed CUP is parallel to Barrier functions. When updating policy according to samples, local policy search (19) requires off-policy evaluation [Achiam *et al.*, 2017], which is very challenging for high-dimension control problem [Duan *et al.*, 2016; Yang *et al.*, 2018, 2021a].

A way to solve CMDP (3) is Lagrangian approach that is also known as primal-dual problem:

$$(\pi_\star, \lambda_\star) = \arg \min_{\lambda \geq 0} \max_{\pi_{\boldsymbol{\theta}} \in \Pi_{\boldsymbol{\theta}}} \left\{ J(\pi_{\boldsymbol{\theta}}) - \lambda(J^c(\pi_{\boldsymbol{\theta}}) - b) \right\}. \tag{20}$$

Although extensive canonical algorithms are proposed to solve problem (20), e.g., [Liang *et al.*, 2018; Tessler *et al.*, 2019; Paternain *et al.*, 2019; Le *et al.*, 2019; Russel *et al.*, 2020; Satija *et al.*, 2020;

Chen *et al.*, 2021], the policy updated by Lagrangian approach may be infeasible w.r.t. CMDP (3). This is hazardous in reinforcement learning when one needs to execute the intermediate policy (which may be unsafe) during training [Chow *et al.*, 2018].

**Constrained Policy Optimization (CPO)**. Recently, Achiam *et al.* [2017] suggest to replace the cost constraint with a surrogate cost function which evaluates the constraint $J^c(\pi_{\boldsymbol{\theta}})$ according to the samples collected from the current policy $\pi_{\boldsymbol{\theta}_k}$, see Eq.(21)-(23). For a given policy $\pi_{\boldsymbol{\theta}_k}$, CPO [Achiam *et al.*, 2017] updates new policy $\pi_{\boldsymbol{\theta}_{k+1}}$ as follows:

$$\pi_{\boldsymbol{\theta}_{k+1}} = \arg \max_{\pi_{\boldsymbol{\theta}} \in \Pi_{\boldsymbol{\theta}}} \quad \mathbb{E}_{s \sim d_{\pi_{\boldsymbol{\theta}_k}}^{\rho_0}(\cdot), a \sim \pi_{\boldsymbol{\theta}}(\cdot|s)} \left[ A_{\pi_{\boldsymbol{\theta}_k}}(s, a) \right] \tag{21}$$

$$\text{s.t.} \quad J^c(\pi_{\boldsymbol{\theta}_k}) + \frac{1}{1 - \gamma} \mathbb{E}_{s \sim d_{\pi_{\boldsymbol{\theta}_k}}^{\rho_0}(\cdot), a \sim \pi_{\boldsymbol{\theta}}(\cdot|s)} \left[ A^c_{\pi_{\boldsymbol{\theta}_k}}(s, a) \right] \leq b, \tag{22}$$

$$\bar{D}_{\text{KL}}(\pi_{\boldsymbol{\theta}}, \pi_{\boldsymbol{\theta}_k}) = \mathbb{E}_{s \sim d_{\pi_{\boldsymbol{\theta}_k}}^{\rho_0}(\cdot)} [\text{KL}(\pi_{\boldsymbol{\theta}}, \pi_{\boldsymbol{\theta}_k})[s]] \leq \delta. \tag{23}$$

Existing recent works (e.g., [Achiam *et al.*, 2017; Vuong *et al.*, 2019; Yang *et al.*, 2020; Han *et al.*, 2020; Bisi *et al.*, 2020; Bharadhwaj *et al.*, 2021]) try to find some convex approximations to replace the term $A_{\pi_{\boldsymbol{\theta}_k}}(s, a)$ and $\bar{D}_{\text{KL}}(\pi_{\boldsymbol{\theta}}, \pi_{\boldsymbol{\theta}_k})$ in Eq.(24)-(26). Such first-order and second-order approximations turn a non-convex problem (24)-(26) to be a convex problem, it seems to make a simple solution, but this approach results in many error sources and troubles in practice. Firstly, it still lacks a theory analysis to show the difference between the non-convex problem (24)-(26) and its convex approximation. Policy optimization is a typical non-convex problem [Yang *et al.*, 2021b]; its convex approximation may introduce some error for its original issue. Secondly, CPO updates parameters according to conjugate gradient [Süli and Mayers, 2003], and its solution involves the inverse Fisher information matrix, which requires expensive computation for each update.

Instead of using a convex approximation for the objective function, the proposed CUP algorithm improves CPO and PCPO at least two aspects. Firstly, the CUP directly optimizes the surrogate objective function via the first-order method, and it does not depend on any convex approximation. Thus, the CUP effectively avoids the expensive computation for the inverse Fisher information matrix. Secondly, CUP extends the surrogate objective function to GAE. Although Zhang *et al.* [2020] has used the GAE technique in experiments, to the best of our knowledge, it still lacks a rigorous theoretical analysis involved GAE before we propose CUP.

# 6 Experiment

In this section, we aim to answer the following three issues:

- Does CUP satisfy the safety constraints in different environments? Does CUP performs well with different cost limits?
- How does CUP compare to the state-of-the-art safe RL algorithms?
- Does CUP play a sensibility during the hyper-parameters in the tuning processing?

We train different robotic agents using five MuJoCo physical simulators [Todorov *et al.*, 2012] which are open by OpenAI Gym API [Brockman *et al.*, 2016], and Safety Gym [Ray *et al.*, 2019]. For more details, see Appendix H.2. Baselines includes CPO [Achiam *et al.*, 2017], PCPO [Yang *et al.*, 2020], TRPO Lagrangian (TRPO-L), PPO Lagrangian (PPO-L) and FOCOPS [Zhang *et al.*, 2020]. TRPO-L and PPO-L are improved by [Chow *et al.*, 2018; Ray *et al.*, 2019], which are based on TRPO [Schulman *et al.*, 2015] and PPO [Schulman *et al.*, 2017]. These two algorithms use the Lagrangian method [Bertsekas, 1997], which applies adaptive penalty coefficients to satisfy the constraint.

## 6.1 Evaluation CUP and Comparison Analysis

We have shown the Learning curves for CUP, and other baselines in Figure 1-2, and Table 1 summarizes the performance of all algorithms. Results show that CUP quickly stabilizes the constraint return around the limit value while converging the objective return to higher values faster. In most cases, the traces of constraint from CUP almost coincide with the dashed black line of the limit. By contrast, the baseline algorithms frequently suffer from over or under the correction.

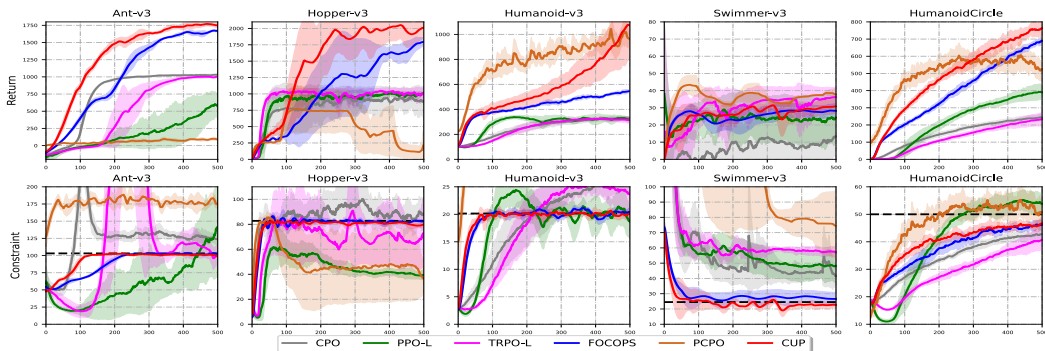

Figure 1: Comparison of CUP to baseline algorithms over 10 seeds on Mujoco.

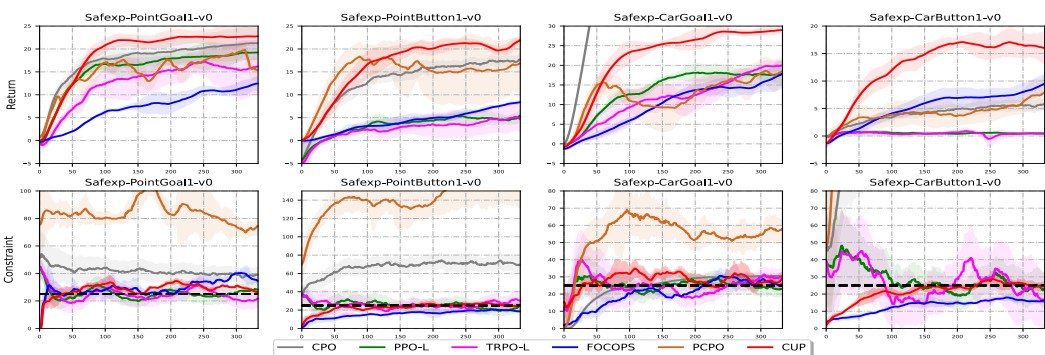

Figure 2: Comparison of CUP to to baseline algorithms over 3 seeds on Satety-Gym.

From Figure 1, we know initial policies of the baseline algorithms are not guaranteed to be feasible, such as in Swimmer-v3, while CUP performs the best and keeps safety learning in Swimmer-v3 tasks. In the HumanoidCircle task, all the algorithms learn steadily to obtain a safe policy, except PPO-L. Additionally, we observed that CUP brings the policy back to the feasible range faster than other baselines in the HumanoidCircle task. In the Ant-v3 task, only the FOCOPS and the proposed CUP learn safely, and both CPO and TRPO-L violate the safety constraints significantly. Besides, although FOCOPS and CUP converge to a safe policy, CUP obtains a better reward performance than FOCOPS in the Ant-v3 task. The result of Figure 2 is relatively complex, the initial policies of the CPO and PCPO are not guaranteed to be feasible on both Safexp-PointGoal1-v0 and Safexp-PointButton1-v0. We think it is not accidental, but it partially provides corroboration of the previous discussions in Appendix B. Both CPO and PCPO use first-order and second-order approximation to approximate a non-convex problem as a convex problem, which inevitably produces a significant deviation from the original RL problem, and it is more serious in large-scale and complex control systems.

From Table 1, we know although PPO-L achieves a reward of $35.58 \pm 5.68$ outperforms CUP in Swimmer-v3, PPO-L obtains a cost of $54.91 \pm 3.93$ that violates the cost limit of $24.5$ significantly, which implies PPO-L learns a dangerous policy under this setting. On the other hand, Figure 1 has shown that CUP generally gains higher returns than different baselines while enforcing the cost constraint. Mainly, CUP achieves a reward performance of $2025.56 \pm 122.35$ that significantly outperforms all the baseline algorithms. Additionally, after equal iterations, CUP performs a greater speed of stabilizing the constraint return around the limit value and is quicker to find feasible policies to gain a more significant objective return.

## 6.2 Sensitivity Analysis for Hyper-Parameters Tuning

Hyper-parameters tuning is necessary to achieve efficient policy improvement and enforce constraints. We investigate the performance with respect to the parameters: $\nu$, step-size $\alpha$, and cost limit $b$. From Figure 3 (a), we know if the estimated cost under the target threshold $b$, then $\nu$ keeps calm, which implies $\nu$ is not activated. Such an empirical phenomenon gives significant expression to the Ant-v3,

| Environment | | CPO | TRPO-L | PPO-L | PCPO | FOCOPS | **CUP** |
|---|---|---|---|---|---|---|---|
| Ant-v3 | Return | $1030.17 \pm 8.15$ | $480.86 \pm 161.05$ | $1012.02 \pm 17.26$ | $90.83 \pm 17.66$ | $1662.53 \pm 17.40$ | $\mathbf{1743.66 \pm 40.5}$ |
| cost limit: 103 | Constraint | $120.76 \pm 4.80$ | $131.07 \pm 67.9$ | $112.45 \pm 15.48$ | $174.80 \pm 5.53$ | $101.31 \pm 0.41$ | $99.11 \pm 0.93$ |
| Hopper-v3 | Return | $875.89 \pm 285.17$ | $1025.49 \pm 10.68$ | $1010.2 \pm 61.48$ | $214.90 \pm 101.22$ | $1687.72 \pm 24.38$ | $\mathbf{2025.56 \pm 122.35}$ |
| cost limit: 83 | Constraint | $76.6 \pm 10.62$ | $40.36 \pm 4.75$ | $83.28 \pm 31.19$ | $36.63 \pm 12.54$ | $102.3 \pm 1.455$ | $79.98 \pm 2.306$ |
| Swimmer-v3 | Return | $18.77 \pm 6.56$ | $27.35 \pm 10.07$ | $\mathbf{35.58 \pm 5.68}$ | $37.73 \pm 3.56$ | $28.15 \pm 4.30$ | $33.38 \pm 0.54$ |
| cost limit: 24.5 | Constraint | $42.07 \pm 3.31$ | $49.58 \pm 7.46$ | $54.91 \pm 3.93$ | $74.39 \pm 22.71$ | $26.54 \pm 4.16$ | $23.31 \pm 0.052$ |
| Humanoid-v3 | Return | $326.95 \pm 16.00$ | $307.71 \pm 24.71$ | $322.11 \pm 25.54$ | $962.13 \pm 57.94$ | $542.5 \pm 4.76$ | $\mathbf{1066.83 \pm 266.12}$ |
| cost limit: 20.0 | Constraint | $26.13 \pm 2.13$ | $18.22 \pm 3.04$ | $20.04 \pm 4.54$ | $48.66 \pm 3.52$ | $20.04 \pm 0.19$ | $19.91 \pm 0.36$ |
| Humanoid-Circle | Return | $237.54 \pm 23.20$ | $384.45 \pm 47.66$ | $243.35 \pm 37.90$ | $525.23 \pm 48.32$ | $713.04 \pm 9.25$ | $\mathbf{768.65 \pm 63.70}$ |
| cost limit: 50.0 | Constraint | $43.64 \pm 1.91$ | $53.77 \pm 1.48$ | $41.17 \pm 3.98$ | $50.80 \pm 4.57$ | $47.73 \pm 0.64$ | $48.23 \pm 0.65$ |

Table 1: Average results for baseline algorithms and CUP over 10 seeds the last 500 iterations.

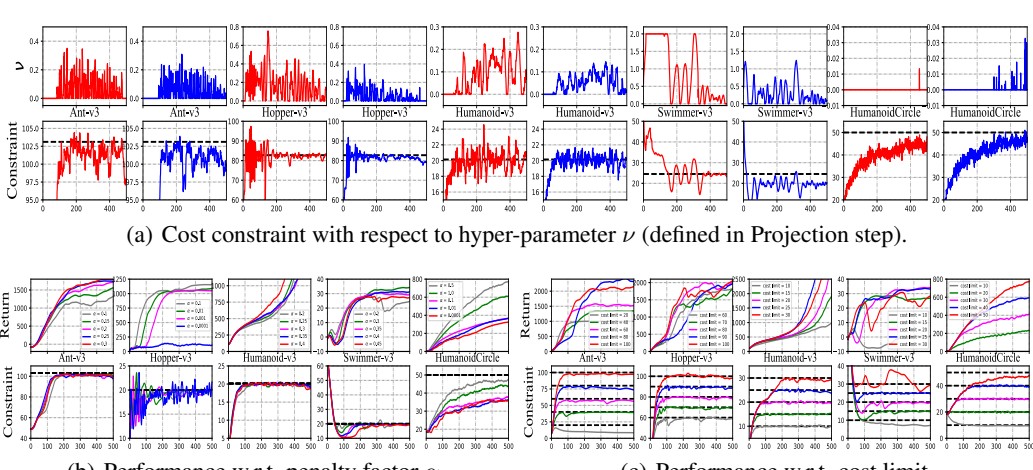

(a) Cost constraint with respect to hyper-parameter $\nu$ (defined in Projection step).

(b) Performance w.r.t. penalty factor $\alpha$.

(c) Performance w.r.t. cost limit.

Figure 3: Sensitivity analysis for hyper-parameters tuning with respect to $\nu$, $\alpha$ and cost limit.

Humanoid-v3, and Hopper-v3 environments. While if the estimated cost exceeds the target threshold $b$, $\nu$ will be activated, which requires the agent to play a policy on the safe region. Those empirical results are consistent with the update rule of $\nu$: $\nu_{k+1} = \{\nu_k + \eta(\hat{J}_k^C - b)\}_+$, which implies the projection of CUP plays an important role for the agent to learn a safe policy. Additionally, Figure 3 (a) provides a visualization way to show the difficulty of different tasks, where the task actives much quantification of $\nu$, such a task is more challenging to obtain a safe policy. Furthermore, Figure 3 (b) shows that the performance of CUP is still very stable for different settings of $\alpha$, and the constraint value of CUP also still fluctuates around the target value. The different value achieved by CUP in different setting $\alpha$ is affected by the simulated environment and constraint thresholds, which are easy to control. Finally, Figure 3 (c) shows that CUP learns safe policies stably under the cost limit thresholds. We compare policy performance and cost under different cost limit settings. For example, in the Swimmer-v3, we set cost limit $b$ among $\{10, 15, 20, 25, 30\}$. Different cost limit setting implies different difficulty for learning, results show that CUP is scalable to various complex tasks, which means CUP is robust to different cost limit settings for various safe RL tasks.

## 7  Conclusion

This paper proposes the CUP algorithm with a theoretical safety guarantee. We derive the CUP based on the newly proposed surrogate functions with respect to objectives and constraints and provide a practical implementation of CUP that does not depend on any convex approximation. We compared CUP against a comprehensive list of safe RL baselines on a wide range of tasks, which shows the effectiveness of CUP where the agent satisfies safe constraints.

## Acknowledgements

The National Key R&D Program of China (2021ZD0200400).Natural Science Foundation of China (No. 61925603, and No.62102349).

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
