| 1030.17 ± 8.15 | 480.86 ± 161.05 | 1012.02 ± 17.26 | 90.83 ± 17.66 | 1662.53 ± 17.40 | **1743.66 ± 40.5** |
| cost limit: 103 | Constraint | 120.76 ± 4.80 | 131.07 ± 67.9 | 112.45 ± 15.48 | 174.80 ± 5.53 | 101.31 ± 0.41 | 99.11 ± 0.93 |
| Hopper-v3 | Return | 875.89 ± 285.17 | 1025.49 ± 10.68 | 1010.2 ± 61.48 | 214.90 ± 101.22 | 1687.72 ± 24.38 | **2025.56 ± 122.35** |
| cost limit: 83 | Constraint | 76.6 ± 10.62 | 40.36 ± 4.75 | 83.28 ± 31.19 | 36.63 ± 12.54 | 102.3 ± 1.455 | 79.98 ± 2.306 |
| Swimmer-v3 | Return | 18.77 ± 6.56 | 27.35 ± 10.07 | **35.58 ± 5.68** | 37.73 ± 3.56 | 28.15 ± 4.30 | 33.38 ± 0.54 |
| cost limit: 24.5 | Constraint | 42.07 ± 3.31 | 49.58 ± 7.46 | 54.91 ± 3.93 | 74.39 ± 22.71 | 26.54 ± 4.16 | 23.31 ± 0.052 |
| Humanoid-v3 | Return | 326.95 ± 16.00 | 307.71 ± 24.71 | 322.11 ± 25.54 | 962.13 ± 57.94 | 542.5 ± 4.76 | **1066.83 ± 266.12** |
| cost limit: 20.0 | Constraint | 26.13 ± 2.13 | 18.22 ± 3.04 | 22.94 ± 4.54 | 48.66 ± 3.52 | 20.04 ± 0.19 | 19.91 ± 0.36 |
| Humanoid-Circle | Return | 237.54 ± 23.20 | 384.45 ± 47.66 | 243.35 ± 37.90 | 525.23 ± 48.32 | 713.04 ± 9.25 | **768.65 ± 63.70** |
| cost limit: 50.0 | Constraint | 43.64 ± 1.91 | 53.77 ± 1.48 | 41.17 ± 3.98 | 50.80 ± 4.57 | 47.73 ± 0.64 | 48.23 ± 0.65 |

Table 1: Average results for baseline algorithms and CUP over 10 seeds the last 500 iterations.

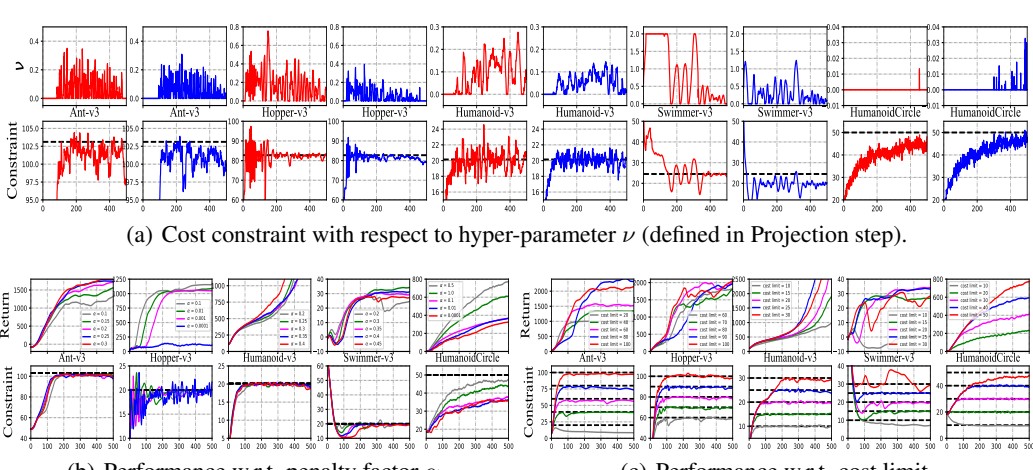

(a) Cost constraint with respect to hyper-parameter $\nu$ (defined in Projection step).

(b) Performance w.r.t. penalty factor $\alpha$.      (c) Performance w.r.t. cost limit.

Figure 3: Sensitivity analysis for hyper-parameters tuning with respect to $\nu$, $\alpha$ and cost limit.

Humanoid-v3, and Hopper-v3 environments. While if the estimated cost exceeds the target threshold $b$, $\nu$ will be activated, which requires the agent to play a policy on the safe region. Those empirical results are consistent with the update rule of $\nu$: $\nu_{k+1} = \{\nu_k + \eta(\hat{J}_k^C - b)\}_+$, which implies the projection of CUP plays an important role for the agent to learn a safe policy. Additionally, Figure 3 (a) provides a visualization way to show the difficulty of different tasks, where the task actives much quantification of $\nu$, such a task is more challenging to obtain a safe policy. Furthermore, Figure 3 (b) shows that the performance of CUP is still very stable for different settings of $\alpha$, and the constraint value of CUP also still fluctuates around the target value. The different value achieved by CUP in different setting $\alpha$ is affected by the simulated environment and constraint thresholds, which are easy to control. Finally, Figure 3 (c) shows that CUP learns safe policies stably under the cost limit thresholds. We compare policy performance and cost under different cost limit settings. For example, in the Swimmer-v3, we set cost limit $b$ among $\{10, 15, 20, 25, 30\}$. Different cost limit setting implies different difficulty for learning, results show that CUP is scalable to various complex tasks, which means CUP is robust to different cost limit settings for various safe RL tasks.

## 7   Conclusion

This paper proposes the CUP algorithm with a theoretical safety guarantee. We derive the CUP based on the newly proposed surrogate functions with respect to objectives and constraints and provide a practical implementation of CUP that does not depend on any convex approximation. We compared CUP against a comprehensive list of safe RL baselines on a wide range of tasks, which shows the effectiveness of CUP where the agent satisfies safe constraints.

## Acknowledgements

The National Key R&D Program of China (2021ZD0200400).Natural Science Foundation of China (No. 61925603, and No.62102349).

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

# A Notations

## A.1 Matrix Index

In this paper, we use a bold capital letter to denote matrix, e.g., $\mathbf{A} = (a_{i,j}) \in \mathbb{R}^{m \times n}$, and its $(i,j)$-th element denoted as

$$\mathbf{A}[i,j] =: a_{i,j},$$

where $1 \leq i \leq m, 1 \leq j \leq n$. Similarly, a bold lowercase letter denotes a vector, e.g., $\mathbf{a} = (a_1, a_2, \cdots, a_n) \in \mathbb{R}^n$, and its $i$-th element denoted as

$$\mathbf{a}[i] =: a_i,$$

where $1 \leq i \leq n$.

## A.2 Key Notations of Reinforcement Learning

For convenience of reference, we list key notations that have be used in this paper.

### A.2.1 Value Function and Dynamic System of MDP.

| | | |
|---|---|---|
| $\mathbf{r}_{\pi_\theta}$, $R_{\pi_\theta}(s)$, | : | $\mathbf{r}_{\pi_\theta} \in \mathbb{R}^{|\mathcal{S}|}$ is the expected vector reward according to $\pi_\theta$, i.e., their components are: $\mathbf{r}_{\pi_\theta}[s] = \sum_{a \in \mathcal{A}} \sum_{s' \in \mathcal{S}} \pi_\theta(a|s) r(s'|s,a) =: R_{\pi_\theta}(s)$, $s \in \mathcal{S}$. |
| $\mathbf{v}_{\pi_\theta}$, $V_{\pi_\theta}(s)$, | : | $\mathbf{v}_{\pi_\theta} \in \mathbb{R}^{|\mathcal{S}|}$ is the vector that stores all the state value functions, and its components are: $\mathbf{v}_{\pi_\theta}[s] = V_{\pi_\theta}(s)$, $s \in \mathcal{S}$. |
| $\rho(\cdot)$, $\boldsymbol{\rho}$ | : | $\rho(s)$: the initial state distribution of state $s$; $\boldsymbol{\rho} \in \mathbb{R}^{|\mathcal{S}|}$, and $\boldsymbol{\rho}[s] = \rho(s)$. |
| $\mathbf{P}_{\pi_\theta}$ | : | Single-step state transition matrix by executing $\pi_\theta$. |
| $\mathbb{P}_{\pi_\theta}(s'|s)$ | : | Single-step state transition probability from $s$ to $s'$ by executing $\pi_\theta$, and it is the $(s,s')$-th component of the matrix $\mathbf{P}_{\pi_\theta}$, i.e., $\mathbf{P}_{\pi_\theta}[s,s'] = \mathbb{P}_{\pi_\theta}(s'|s)$. |
| $\mathbb{P}_{\pi_\theta}(s_t = s'|s)$ | : | The probability of visiting the state $s'$ after $t$ time steps from the state $s$ by executing $\pi_\theta$, and it is the $(s,s')$-th component of the matrix $\mathbf{P}_{\pi_\theta}$, i.e., $\mathbf{P}_{\pi_\theta}^t[s,s'] = \mathbb{P}_{\pi_\theta}(s_t = s'|s)$. |
| $d_{\pi_\theta}^{s_0}(s)$, $d_{\pi_\theta}^{\rho_0}(s)$ | : | The normalized discounted distribution of the future state $s$ encountered starting at $s_0$ by executing $\pi_\theta$: $d_{\pi_\theta}^{s_0}(s) =: (1-\gamma) \sum_{t=0}^{\infty} \gamma^t \mathbb{P}_{\pi_\theta}(s_t = s|s_0)$. Since $s_0 \sim \rho(\cdot)$, we define $d_{\pi_\theta}^{\rho_0}(s) =: \mathbb{E}_{s_0 \sim \rho(\cdot)}[d_{\pi_\theta}^{s_0}(s)]$. |
| $\mathbf{d}_{\pi_\theta}^{\rho_0}$ | : | It stores all the normalized discounted state distributions $d_{\pi_\theta}^{\rho_0}(s)$, $\in \mathcal{S}$, i.e., $\mathbf{d}_{\pi_\theta}^{\rho_0} \in \mathbb{R}^{|\mathcal{S}|}$, and its components are: $\mathbf{d}_{\pi_\theta}^{\rho_0}[s] = d_{\pi_\theta}^{\rho_0}(s)$. |

### A.2.2 Extend them to $\lambda$-version.

| | | |
|---|---|---|
| $\mathbf{P}_{\pi_\theta}^{(\lambda)}$ | : | $\mathbf{P}_{\pi_\theta}^{(\lambda)} = (1 - \gamma\lambda) \sum_{t=0}^{\infty} (\gamma\lambda)^t \mathbf{P}_{\pi_\theta}^{t+1}$. |
| $\mathbb{P}_{\pi_\theta}^{(\lambda)}(s'|s)$ | : | $\mathbb{P}_{\pi_\theta}^{(\lambda)}(s'|s) =: \mathbf{P}_{\pi_\theta}^{(\lambda)}[s,s'] = (1 - \gamma\lambda) \sum_{t=0}^{\infty} (\gamma\lambda)^t \mathbb{P}_{\pi_\theta}(s_{t+1} = s'|s)$. |
| $\mathbf{r}_{\pi_\theta}^{(\lambda)}$, $R_{\pi_\theta}^{(\lambda)}(s)$ | : | $\mathbf{r}_{\pi_\theta}^{(\lambda)} = \sum_{t=0}^{\infty} (\gamma\lambda \mathbf{P}_{\pi_\theta})^t \mathbf{r}_{\pi_\theta}$; $R_{\pi_\theta}^{(\lambda)}(s) =: \mathbf{r}_{\pi_\theta}^{(\lambda)}[s]$. |
| $\tilde{\gamma}$ | : | $\tilde{\gamma} = \dfrac{\gamma(1-\lambda)}{1-\gamma\lambda}$. |
| $d_{\pi_\theta}^{s_0,\lambda}(s)$ | : | $d_{\pi_\theta}^{s_0,\lambda}(s) = (1 - \tilde{\gamma}) \sum_{t=0}^{\infty} \tilde{\gamma}^t \mathbb{P}_{\pi_\theta}^{(\lambda)}(s_t = s|s_0)$. |
| $d_{\pi_\theta}^{\lambda}(s)$, $\mathbf{d}_{\pi_\theta}^{\lambda}$ | : | $d_{\pi_\theta}^{\lambda}(s) = \mathbb{E}_{s_0 \sim \rho_0(\cdot)}\left[d_{\pi_\theta}^{s_0,\lambda}(s)\right]$, $\mathbf{d}_{\pi_\theta}^{\lambda}[s] = d_{\pi_\theta}^{\lambda}(s)$. |

### A.2.3 TD error w.r.t. any function $\varphi(\cdot)$.

| | | |
|---|---|---|
| $\delta_t^{\varphi}$ | : | $\delta_t^{\varphi} = r(s_{t+1}|s_t, a_t) + \gamma\varphi(s_{t+1}) - \varphi(s_t)$. |
| $\delta_{\pi_\theta,t}^{\varphi}(s)$ | : | $\delta_{\pi_\theta,t}^{\varphi}(s) = \mathbb{E}_{s_t \sim \mathbb{P}_{\pi_\theta}(\cdot|s), a_t \sim \pi_\theta(\cdot|s_t), s_{t+1} \sim \mathbb{P}(\cdot|s_t,a_t)}[\delta_t^{\varphi}]$. |
| $\boldsymbol{\delta}_{\pi_\theta,t}^{\varphi}$ | : | $\boldsymbol{\delta}_{\pi_\theta,t}^{\varphi}[s] = \delta_{\pi_\theta,t}^{\varphi}(s)$. |
| $\Delta_t^{\varphi}(\pi_\theta, \pi_{\theta'}, s)$ | : | $\mathbb{E}_{s_t \sim \mathbb{P}_{\pi_{\theta'}}(\cdot|s), a_t \sim \pi_{\theta'}(\cdot|s_t), s_{t+1} \sim \mathbb{P}(\cdot|s_t,a_t)}\left[\left(\dfrac{\pi_\theta(a_t|s_t)}{\pi_{\theta'}(a_t|s_t)} - 1\right)\delta_t^{\varphi}\right]$. |
| $\boldsymbol{\Delta}_t^{\varphi}(\pi_\theta, \pi_{\theta'})$ | : | $\boldsymbol{\Delta}_t^{\varphi}(\pi_\theta, \pi_{\theta'})[s] = \Delta_t^{\varphi}(\pi_\theta, \pi_{\theta'}, s)$. |

# B  Additional Discussion about Related Work

This section reviews three typical safe reinforcement learning algorithms: CPO [Achiam *et al.*, 2017], PCPO [Yang *et al.*, 2020] and FOCOPS [Zhang *et al.*, 2020]. Those algorithms also use new surrogate functions to replace the objective and constraints, which resembles the proposed CUP algorithm. The goal is to present the contribution of our work.

## B.1  CPO [Achiam *et al.*, 2017]

For a given policy $\pi_{\boldsymbol{\theta}_k}$, CPO updates new policy $\pi_{\boldsymbol{\theta}_{k+1}}$ as follows:

$$\pi_{\boldsymbol{\theta}_{k+1}} = \arg\max_{\pi_{\boldsymbol{\theta}} \in \Pi_{\boldsymbol{\theta}}} \quad \mathbb{E}_{s \sim d_{\pi_{\boldsymbol{\theta}_k}}^{\rho_0}(\cdot), a \sim \pi_{\boldsymbol{\theta}}(\cdot|s)} \left[ A_{\pi_{\boldsymbol{\theta}_k}}(s,a) \right] \tag{24}$$

$$\text{s.t.} \quad J^c(\pi_{\boldsymbol{\theta}_k}) + \frac{1}{1-\gamma} \mathbb{E}_{s \sim d_{\pi_{\boldsymbol{\theta}_k}}^{\rho_0}(\cdot), a \sim \pi_{\boldsymbol{\theta}}(\cdot|s)} \left[ A_{\pi_{\boldsymbol{\theta}_k}}^c(s,a) \right] \le b, \tag{25}$$

$$\bar{D}_{\mathrm{KL}}(\pi_{\boldsymbol{\theta}}, \pi_{\boldsymbol{\theta}_k}) = \mathbb{E}_{s \sim d_{\pi_{\boldsymbol{\theta}_k}}^{\rho_0}(\cdot)}[\mathrm{KL}(\pi_{\boldsymbol{\theta}}, \pi_{\boldsymbol{\theta}_k})[s]] \le \delta. \tag{26}$$

It is impractical to solve the problem (24) directly due to the computational cost. [Achiam *et al.*, 2017] suggest to find some convex approximations to replace the term $A_{\pi_{\boldsymbol{\theta}_k}}(s,a)$ and $\bar{D}_{\mathrm{KL}}(\pi_{\boldsymbol{\theta}}, \pi_{\boldsymbol{\theta}_k})$ Eq.(24)-(26).

Concretely, according to (5), Achiam *et al.* [2017] suggest to use first-order Taylor expansion of $J(\pi_{\boldsymbol{\theta}})$ to replace the objective (24) as follows,

$$\frac{1}{1-\gamma} \mathbb{E}_{s \sim d_{\pi_{\boldsymbol{\theta}_k}}^{\rho_0}(\cdot), a \sim \pi_{\boldsymbol{\theta}_k}(\cdot|s)} \left[ \frac{\pi_{\boldsymbol{\theta}}(a|s)}{\pi_{\boldsymbol{\theta}_k}(a|s)} A_{\pi_{\boldsymbol{\theta}_k}}(s,a) \right] = J(\pi_{\boldsymbol{\theta}}) - J(\pi_{\boldsymbol{\theta}_k}) \approx (\boldsymbol{\theta} - \boldsymbol{\theta}_k)^\top \nabla_{\boldsymbol{\theta}} J(\pi_{\boldsymbol{\theta}}).$$

Similarly, Achiam *et al.* [2017] use the following approximations to turn the constrained policy optimization (24)-(26) to be a convex problem,

$$\frac{1}{1-\gamma} \mathbb{E}_{s \sim d_{\pi_{\boldsymbol{\theta}_k}}^{\rho_0}(\cdot), a \sim \pi_{\boldsymbol{\theta}_k}(\cdot|s)} \left[ \frac{\pi_{\boldsymbol{\theta}}(a|s)}{\pi_{\boldsymbol{\theta}_k}(a|s)} A_{\pi_{\boldsymbol{\theta}_k}}^c(s,a) \right] \approx (\boldsymbol{\theta} - \boldsymbol{\theta}_k)^\top \nabla_{\boldsymbol{\theta}} J^c(\pi_{\boldsymbol{\theta}}), \tag{27}$$

$$\bar{D}_{\mathrm{KL}}(\pi_{\boldsymbol{\theta}}, \pi_{\boldsymbol{\theta}_k}) \approx (\boldsymbol{\theta} - \boldsymbol{\theta}_k)^\top \mathbf{H}(\boldsymbol{\theta} - \boldsymbol{\theta}_k), \tag{28}$$

where $\mathbf{H}$ is Hessian matrix of $\bar{D}_{\mathrm{KL}}(\pi_{\boldsymbol{\theta}}, \pi_{\boldsymbol{\theta}_k})$, i.e.,

$$\mathbf{H}[i,j] =: \frac{\partial^2}{\partial \boldsymbol{\theta}_i \partial \boldsymbol{\theta}_j} \mathbb{E}_{s \sim d_{\pi_{\boldsymbol{\theta}_k}}^{\rho_0}(\cdot)} \left[ \mathrm{KL}(\pi_{\boldsymbol{\theta}}, \pi_{\boldsymbol{\theta}_k})[s] \right],$$

Eq.(28) is the second-oder approximation of (26).

Let $\lambda_\star, \nu_\star$ is the dual solution of the following problem

$$\lambda_\star, \nu_\star = \arg\max_{\lambda \ge 0, \nu \ge 0} \left\{ \frac{-1}{2\lambda} \left( \mathbf{g}^\top \mathbf{H}^{-1} \mathbf{g} - 2\nu r + s\nu^2 \right) + \nu c - \frac{\lambda \delta}{2} \right\};$$

where $\mathbf{g} = \nabla_{\boldsymbol{\theta}} \mathbb{E}_{s \sim d_{\pi_{\boldsymbol{\theta}_k}}^{\rho_0}(\cdot), a \sim \pi_{\boldsymbol{\theta}}(\cdot|s)} \left[ A_{\pi_{\boldsymbol{\theta}_k}}(s,a) \right]$, $\mathbf{a} = \nabla_{\boldsymbol{\theta}} \mathbb{E}_{s \sim d_{\pi_{\boldsymbol{\theta}_k}}^{\rho_0}(\cdot), a \sim \pi_{\boldsymbol{\theta}}(\cdot|s)} \left[ A_{\pi_{\boldsymbol{\theta}_k}}^c(s,a) \right]$, $r = \mathbf{g}^\top \mathbf{H} \mathbf{a}$, $s = \mathbf{a}^\top \mathbf{H}^{-1} \mathbf{a}$, and $c = J^c(\pi_{\boldsymbol{\theta}_k}) - b$.

Finally, CPO updates parameters according to conjugate gradient as follows: if approximation to CPO is feasible:

$$\boldsymbol{\theta}_{k+1} = \boldsymbol{\theta}_k + \frac{1}{\lambda_\star} \mathbf{H}^{-1}(\mathbf{g} - \nu_\star \mathbf{a}),$$

else,

$$\boldsymbol{\theta}_{k+1} = \boldsymbol{\theta}_k - \sqrt{\frac{2\delta}{\mathbf{a}^\top \mathbf{H}^{-1} \mathbf{a}}} \mathbf{H}^{-1} \mathbf{a}.$$

## B.2 PCPO [Yang *et al.*, 2020]

Projection-Based Constrained Policy Optimization (PCPO) is an iterative method for optimizing policies in a two-step process: the first step performs a local reward improvement update, while the second step reconciles any constraint violation by projecting the policy back onto the constraint set.

**Reward Improvement:**

$$\pi_{\boldsymbol{\theta}_{k+\frac{1}{2}}} = \arg \max_{\pi_{\boldsymbol{\theta}} \in \Pi_{\boldsymbol{\theta}}} \mathbb{E}_{s \sim d_{\pi_{\boldsymbol{\theta}_k}}^{\rho_0}(\cdot), a \sim \pi_{\boldsymbol{\theta}}(\cdot|s)} \left[ A_{\pi_{\boldsymbol{\theta}_k}}(s,a) \right],$$

$$\text{s.t.} \bar{D}_{\mathrm{KL}}(\pi_{\boldsymbol{\theta}}, \pi_{\boldsymbol{\theta}_k}) = \mathbb{E}_{s \sim d_{\pi_{\boldsymbol{\theta}_k}}^{\rho_0}(\cdot)}[\mathrm{KL}(\pi_{\boldsymbol{\theta}}, \pi_{\boldsymbol{\theta}_k})[s]] \leq \delta;$$

**Projection:**

$$\pi_{\boldsymbol{\theta}_{k+1}} = \arg \min_{\pi_{\boldsymbol{\theta}} \in \Pi_{\boldsymbol{\theta}}} D\left( \pi_{\boldsymbol{\theta}}, \pi_{\boldsymbol{\theta}_{k+\frac{1}{2}}} \right),$$

$$\text{s.t.} \ J^c(\pi_{\boldsymbol{\theta}_k}) + \frac{1}{1-\gamma} \mathbb{E}_{s \sim d_{\pi_{\boldsymbol{\theta}_k}}^{\rho_0}(\cdot), a \sim \pi_{\boldsymbol{\theta}}(\cdot|s)} \left[ A^c_{\pi_{\boldsymbol{\theta}_k}}(s,a) \right] \leq b.$$

Then, Yang *et al.* [2020] follows CPO [Achiam *et al.*, 2017] uses convex approximation to original problem, and calculate the update rule as follows,

$$\boldsymbol{\theta}_{k+1} = \boldsymbol{\theta}_k - \sqrt{\frac{2\delta}{\mathbf{g}^\top \mathbf{H}^{-1} \mathbf{g}}} \mathbf{H}^{-1} \mathbf{g} - \max\left( 0, \frac{\sqrt{\frac{2\delta}{\mathbf{g}^\top \mathbf{H}^{-1} \mathbf{g}}} \mathbf{a}^\top \mathbf{H}^{-1} \mathbf{g} + c}{\mathbf{a}^\top \mathbf{L}^{-1} \mathbf{a}} \right) \mathbf{L}^{-1} \mathbf{a},$$

where $\mathbf{L} = \mathbf{I}$ if $D$ is $\ell_2$-norm, and $\mathbf{L} = \mathbf{H}$ if $D$ is KL-divergence.

## B.3 FOCOPS [Zhang *et al.*, 2020]

Zhang *et al.* [2020] propose the First Order Constrained Optimization in Policy Space (FOCOPS) that is a two-step approach. We present it as follows.

**Step1: Finding the optimal update policy.** Firstly, for a given policy $\pi_{\boldsymbol{\theta}k}$, we find an optimal update policy $\pi^\star$ by solving the optimization problem (24)-(26) in the non-parameterized policy space.

$$\pi^\star = \arg \max_{\pi \in \Pi} \quad \mathbb{E}_{s \sim d_{\pi_{\boldsymbol{\theta}_k}}^{\rho_0}(\cdot), a \sim \pi(\cdot|s)} \left[ A_{\pi_{\boldsymbol{\theta}_k}}(s,a) \right] \tag{29}$$

$$\text{s.t.} \ J^c(\pi_{\boldsymbol{\theta}_k}) + \frac{1}{1-\gamma} \mathbb{E}_{s \sim d_{\pi_{\boldsymbol{\theta}_k}}^{\rho_0}(\cdot), a \sim \pi(\cdot|s)} \left[ A^c_{\pi_{\boldsymbol{\theta}_k}}(s,a) \right] \leq b, \tag{30}$$

$$\bar{D}_{\mathrm{KL}}(\pi_{\boldsymbol{\theta}}, \pi_{\boldsymbol{\theta}_k}) = \mathbb{E}_{s \sim d_{\pi_{\boldsymbol{\theta}_k}}^{\rho_0}(\cdot)}[\mathrm{KL}(\pi, \pi_{\boldsymbol{\theta}_k})[s]] \leq \delta. \tag{31}$$

If $\pi_{\boldsymbol{\theta}_k}$ is feasible, then the optimal policy for (29)-(31) takes the following form:

$$\pi^\star(a|s) = \frac{\pi_{\boldsymbol{\theta}_k}(a|s)}{Z_{\lambda,\nu}(s)} \exp\left( \frac{1}{\lambda} \left( A_{\pi_{\boldsymbol{\theta}_k}}(s,a) - \nu A^c_{\pi_{\boldsymbol{\theta}_k}}(s,a) \right) \right), \tag{32}$$

where $Z_{\lambda,\nu}(s)$ is the partition function which ensures (32) is a valid probability distribution, $\lambda$ and $\nu$ are solutions to the optimization problem:

$$\min_{\lambda, \nu \geq 0} \lambda \nu + \nu \tilde{b} + \lambda \mathbb{E}_{s \sim d_{\pi_{\boldsymbol{\theta}_k}}^{\rho_0}(\cdot), a \sim \pi^\star(\cdot|s)} \left[ Z_{\lambda,\nu}(s) \right],$$

the term $\tilde{b} = (1-\gamma)(b - J^c(\pi_{\boldsymbol{\theta}_k}))$.

**Step 2: Projection** Then, we project the policy found in the previous step back into the parameterized policy space $\Pi_{\boldsymbol{\theta}}$ by solving for the closest policy $\pi_{\boldsymbol{\theta}} \in \Pi_{\boldsymbol{\theta}}$ to $\pi^\star$ in order to obtain $\pi_{\boldsymbol{\theta}_{k+1}}$:

$$\boldsymbol{\theta}_{k+1} = \arg \min_{\boldsymbol{\theta}} \mathbb{E}_{s \sim d_{\pi_{\boldsymbol{\theta}_k}}^{\rho_0}(\cdot)}[\mathrm{KL}(\pi_{\boldsymbol{\theta}}, \pi^\star)[s]].$$

Table 2: Comparison of some safe reinforcement algorithms.

| Algorithm | Optimization problem | Implementation | Remark |
|---|---|---|---|
| CPO [Achiam *et al.*, 2017] | $\pi_{\boldsymbol{\theta}_{k+1}} = \arg\max_{\pi_{\boldsymbol{\theta}} \in \Pi_{\boldsymbol{\theta}}} \mathbb{E}_{s \sim d^{\rho_0}_{\pi_{\boldsymbol{\theta}_k}}(\cdot), a \sim \pi_{\boldsymbol{\theta}}(\cdot\|s)} \left[ A_{\pi_{\boldsymbol{\theta}_k}}(s,a) \right],$ s.t. $J^c(\pi_{\boldsymbol{\theta}_k}) + \mathbb{E}_{s \sim d^{\rho_0}_{\pi_{\boldsymbol{\theta}_k}}(\cdot), a \sim \pi_{\boldsymbol{\theta}}(\cdot\|s)} \left[ A^c_{\pi_{\boldsymbol{\theta}_k}}(s,a) \right] \leq b,$ $\bar{D}_{\mathrm{KL}}(\pi_{\boldsymbol{\theta}}, \pi_{\boldsymbol{\theta}_k}) = \mathbb{E}_{s \sim d^{\rho_0}_{\pi_{\boldsymbol{\theta}_k}}(\cdot)} [\mathrm{KL}(\pi_{\boldsymbol{\theta}}, \pi_{\boldsymbol{\theta}_k})[s]] \leq \delta.$ | $\boldsymbol{\theta}_{k+1} = \arg\max_{\boldsymbol{\theta}} \mathbf{g}^\top (\boldsymbol{\theta} - \boldsymbol{\theta}_k),$ s.t. $c + \mathbf{b}^\top(\boldsymbol{\theta} - \boldsymbol{\theta}_k) \leq 0,$ $\frac{1}{2}(\boldsymbol{\theta} - \boldsymbol{\theta}_k)^\top \mathbf{H}(\boldsymbol{\theta} - \boldsymbol{\theta}_k) \leq \delta.$ | Convex Implementation |
| PCPO [Yang *et al.*, 2020] | Reward Improvement $\pi_{\boldsymbol{\theta}_{k+\frac{1}{2}}} = \arg\max_{\pi_{\boldsymbol{\theta}} \in \Pi_{\boldsymbol{\theta}}} \mathbb{E}_{s \sim d^{\rho_0}_{\pi_{\boldsymbol{\theta}_k}}(\cdot), a \sim \pi_{\boldsymbol{\theta}}(\cdot\|s)} \left[ A_{\pi_{\boldsymbol{\theta}_k}}(s,a) \right],$ s.t. $\bar{D}_{\mathrm{KL}}(\pi_{\boldsymbol{\theta}}, \pi_{\boldsymbol{\theta}_k}) = \mathbb{E}_{s \sim d^{\rho_0}_{\pi_{\boldsymbol{\theta}_k}}(\cdot)} [\mathrm{KL}(\pi_{\boldsymbol{\theta}}, \pi_{\boldsymbol{\theta}_k})[s]] \leq \delta;$ Projection $\pi_{\boldsymbol{\theta}_{k+1}} = \arg\min_{\pi_{\boldsymbol{\theta}} \in \Pi_{\boldsymbol{\theta}}} D\left(\pi_{\boldsymbol{\theta}}, \pi_{\boldsymbol{\theta}_{k+\frac{1}{2}}}\right),$ s.t. $J^c(\pi_{\boldsymbol{\theta}_k}) + \frac{1}{1-\gamma} \mathbb{E}_{s \sim d^{\rho_0}_{\pi_{\boldsymbol{\theta}_k}}(\cdot), a \sim \pi_{\boldsymbol{\theta}}(\cdot\|s)} \left[ A^c_{\pi_{\boldsymbol{\theta}_k}}(s,a) \right] \leq b.$ | Reward Improvement $\boldsymbol{\theta}_{k+\frac{1}{2}} = \arg\max_{\boldsymbol{\theta}} \mathbf{g}^\top (\boldsymbol{\theta} - \boldsymbol{\theta}_k),$ s.t. $\frac{1}{2}(\boldsymbol{\theta} - \boldsymbol{\theta}_k)^\top \mathbf{H}(\boldsymbol{\theta} - \boldsymbol{\theta}_k) \leq \delta;$ Projection $\boldsymbol{\theta}_{k+1} = \arg\min_{\boldsymbol{\theta}} \frac{1}{2}(\boldsymbol{\theta} - \boldsymbol{\theta}_k)^\top \mathbf{L}(\boldsymbol{\theta} - \boldsymbol{\theta}_k),$ s.t. $c + \mathbf{b}^\top(\boldsymbol{\theta} - \boldsymbol{\theta}_k) \leq 0.$ | Convex Implementation |
| FOCOPS [Zhang *et al.*, 2020] | Optimal update policy $\pi^\star = \arg\max_{\pi \in \Pi} \mathbb{E}_{s \sim d^{\rho_0}_{\pi_{\boldsymbol{\theta}_k}}(\cdot), a \sim \pi(\cdot\|s)} \left[ A_{\pi_{\boldsymbol{\theta}_k}}(s,a) \right],$ s.t. $J^c(\pi_{\boldsymbol{\theta}_k}) + \mathbb{E}_{s \sim d^{\rho_0}_{\pi_{\boldsymbol{\theta}_k}}(\cdot), a \sim \pi(\cdot\|s)} \left[ A^c_{\pi_{\boldsymbol{\theta}_k}}(s,a) \right] \leq b,$ $\bar{D}_{\mathrm{KL}}(\pi_{\boldsymbol{\theta}}, \pi_{\boldsymbol{\theta}_k}) = \mathbb{E}_{s \sim d^{\rho_0}_{\pi_{\boldsymbol{\theta}_k}}(\cdot)} [\mathrm{KL}(\pi, \pi_{\boldsymbol{\theta}_k})[s]] \leq \delta;$ Projection $\pi_{\boldsymbol{\theta}_{k+1}} = \arg\min_{\pi_{\boldsymbol{\theta}} \in \Pi_{\boldsymbol{\theta}}} \mathbb{E}_{s \sim d^{\rho_0}_{\pi_{\boldsymbol{\theta}_k}}(\cdot)} [\mathrm{KL}(\pi_{\boldsymbol{\theta}}, \pi^\star)[s]].$ | Optimal update policy $\pi^\star(a\|s) = \frac{\pi_{\boldsymbol{\theta}_k}(a\|s)}{Z_{\lambda,\nu}(s)} \exp\left( \frac{1}{\lambda} \left( A_{\pi_{\boldsymbol{\theta}_k}}(s,a) - \nu A^c_{\pi_{\boldsymbol{\theta}_k}}(s,a) \right) \right);$ Projection $\boldsymbol{\theta}_{k+1} = \arg\min_{\boldsymbol{\theta}} \mathbb{E}_{s \sim d^{\rho_0}_{\pi_{\boldsymbol{\theta}_k}}(\cdot)} [\mathrm{KL}(\pi_{\boldsymbol{\theta}}, \pi^\star)[s]].$ | Non-Convex Implementation |
| CUP (Our Work) | Policy Improvement $\pi_{\boldsymbol{\theta}_{k+\frac{1}{2}}} = \arg\max_{\pi_{\boldsymbol{\theta}} \in \Pi_{\boldsymbol{\theta}}} \left\{ \mathbb{E}_{s \sim d^{\lambda}_{\pi_{\boldsymbol{\theta}_k}}(\cdot), a \sim \pi_{\boldsymbol{\theta}}(\cdot\|s)} \left[ A^{\mathrm{GAE}(\gamma,\lambda)}_{\pi_{\boldsymbol{\theta}_k}}(s,a) \right] -\alpha_k \sqrt{\mathbb{E}_{s \sim d^{\lambda}_{\pi_{\boldsymbol{\theta}_k}}(\cdot)} [\mathrm{KL}(\pi_{\boldsymbol{\theta}_k}, \pi_{\boldsymbol{\theta}})[s]]} \right\},$ Projection $\pi_{\boldsymbol{\theta}_{k+1}} = \arg\min_{\pi_{\boldsymbol{\theta}} \in \Pi_{\boldsymbol{\theta}}} D\left(\pi_{\boldsymbol{\theta}}, \pi_{\boldsymbol{\theta}_{k+\frac{1}{2}}}\right),$ s.t. $J^c(\pi_{\boldsymbol{\theta}_k}) + \frac{1}{1-\tilde{\gamma}} \mathbb{E}_{s \sim d^{\lambda}_{\pi_{\boldsymbol{\theta}_k}}(\cdot), a \sim \pi_{\boldsymbol{\theta}}(\cdot\|s)} \left[ A^{\mathrm{GAE}(\gamma,\lambda),C}_{\pi_{\boldsymbol{\theta}_k}}(s,a) \right]$ $+ \beta_k \sqrt{\mathbb{E}_{s \sim d^{\lambda}_{\pi_{\boldsymbol{\theta}_k}}(\cdot)} [\mathrm{KL}(\pi_{\boldsymbol{\theta}_k}, \pi_{\boldsymbol{\theta}})[s]]} \leq b.$ | Policy Improvement $\boldsymbol{\theta}_{k+\frac{1}{2}} = \arg\max_{\boldsymbol{\theta}} \left\{ \frac{1}{T} \sum_{t=1}^{T} \frac{\pi_{\boldsymbol{\theta}}(a_t\|s_t)}{\pi_{\boldsymbol{\theta}_k}(a_t\|s_t)} \hat{A}_t -\alpha \sqrt{\frac{1}{T} \sum_{t=1}^{T} \mathrm{KL}(\pi_{\boldsymbol{\theta}_k}(\cdot\|s_t), \pi_{\boldsymbol{\theta}}(\cdot\|s_t))} \right\};$ Projection $\boldsymbol{\theta}_{k+1} = \arg\min_{\boldsymbol{\theta}} \frac{1}{T} \sum_{t=1}^{T} \left\{ \mathrm{KL}\left( \pi_{\boldsymbol{\theta}_{k+\frac{1}{2}}}(\cdot\|s_t), \pi_{\boldsymbol{\theta}}(\cdot\|s_t) \right) + \nu_k \frac{1-\gamma\lambda}{1-\gamma} \frac{\pi_{\boldsymbol{\theta}}(a_t\|s_t)}{\pi_{\boldsymbol{\theta}_k}(a_t\|s_t)} \hat{A}^C_t \right\}.$ | Non-Convex Implementation |

# C Constrained Update Projection Algorithm

## C.1 Practical Implementation of Performance Improvement

### C.1.1 Sample-based Performance Improvement

Let the trajectory $\{(s_t, a_t, r_{t+1}, c_{t+1})\}_{t=1}^{T}$ be sampled according to $\pi_{\boldsymbol{\theta}_k}$, then we denote the empirical KL-divergence with respect to $\pi_{\boldsymbol{\theta}}$ and $\pi_{\boldsymbol{\theta}'}$ as follows,

$$\hat{D}_{\mathrm{KL}}(\pi_{\boldsymbol{\theta}}, \pi_{\boldsymbol{\theta}'}) = \frac{1}{T} \sum_{t=1}^{T} \mathrm{KL}(\pi_{\boldsymbol{\theta}}(a_t|s_t), \pi_{\boldsymbol{\theta}'}(a_t|s_t)).$$

We defined the following $\hat{\mathcal{L}}_{\mathrm{R}}(\pi_{\boldsymbol{\theta}}, \pi_{\boldsymbol{\theta}_k})$,

$$\hat{\mathcal{L}}_{\mathrm{R}}(\pi_{\boldsymbol{\theta}}, \pi_{\boldsymbol{\theta}_k}) = \frac{1}{T} \sum_{t=1}^{T} \frac{\pi_{\boldsymbol{\theta}}(a_t|s_t)}{\pi_{\boldsymbol{\theta}_k}(a_t|s_t)} \hat{A}_t - \alpha_k \sqrt{\hat{D}_{\mathrm{KL}}(\pi_{\boldsymbol{\theta}_k}, \pi_{\boldsymbol{\theta}})}, \tag{33}$$

where $\hat{A}_t$ is an estimator of $A_{\pi_{\boldsymbol{\theta}_k}}^{\mathtt{GAE}(\gamma,\lambda)}(s, a)$. The term $\hat{\mathcal{L}}_{\mathrm{R}}(\pi_{\boldsymbol{\theta}}, \pi_{\boldsymbol{\theta}_k})$ (33) is an estimator of the next expectation that appears in (14)

$$\mathbb{E}_{s \sim d_{\pi_{\boldsymbol{\theta}_k}}^{\lambda}(\cdot), \, a \sim \pi_{\boldsymbol{\theta}_k}(\cdot|s)} \left[ \frac{\pi_{\boldsymbol{\theta}}(a|s)}{\pi_{\boldsymbol{\theta}_k}(a|s)} A_{\pi_{\boldsymbol{\theta}_k}}^{\mathtt{GAE}(\gamma,\lambda)}(s, a) \right] - \alpha_k \sqrt{\mathbb{E}_{s \sim d_{\pi_{\boldsymbol{\theta}_k}}^{\lambda}(\cdot)} \left[ \mathrm{KL}(\pi_{\boldsymbol{\theta}_k}, \pi_{\boldsymbol{\theta}})[s] \right]}.$$

Then we implement the performance improvement as follows,

$$\pi_{\boldsymbol{\theta}_{k+\frac{1}{2}}} = \arg \max_{\pi_{\boldsymbol{\theta}} \in \Pi_{\boldsymbol{\theta}}} \left\{ \hat{\mathcal{L}}_{\mathrm{R}}(\pi_{\boldsymbol{\theta}}, \pi_{\boldsymbol{\theta}_k}) \right\}. \tag{34}$$

### C.1.2 Clipped Surrogate Objective

How can the implementation (34) take the biggest possible improvement step on a policy using the data we currently have, without stepping so far that we accidentally cause performance collapse? Now, we present a clip implementation for policy improvement, which is very efficient in practice.

Instead of the previous policy improvement (14), according to PPO Schulman *et al.* [2017], we update the policy as follows,

$$\pi_{\boldsymbol{\theta}_{k+\frac{1}{2}}} = \arg \max_{\pi_{\boldsymbol{\theta}} \in \Pi_{\boldsymbol{\theta}}} \left\{ \mathbb{E}_{s \sim d_{\pi_{\boldsymbol{\theta}_k}}^{\lambda}(\cdot), \, a \sim \pi_{\boldsymbol{\theta}}(\cdot|s)} \left[ \mathcal{L}_{\mathrm{clip}}(s, a, \pi_{\boldsymbol{\theta}}, \pi_{\boldsymbol{\theta}_k}, \epsilon) \right] \right\},$$

where the the objective $\mathcal{L}_{\mathrm{clip}}$ is defined as follows,

$$\mathcal{L}_{\mathrm{clip}}(s, a, \pi_{\boldsymbol{\theta}}, \pi_{\boldsymbol{\theta}_k}, \epsilon) = \min \left\{ \frac{\pi_{\boldsymbol{\theta}}(a|s)}{\pi_{\boldsymbol{\theta}_k}(a|s)} A_{\pi_{\boldsymbol{\theta}_k}}^{\mathtt{GAE}(\gamma,\lambda)}(s, a), \, \mathrm{clip}\left( \frac{\pi_{\boldsymbol{\theta}}(a|s)}{\pi_{\boldsymbol{\theta}_k}(a|s)}, 1 - \epsilon, 1 + \epsilon \right) A_{\pi_{\boldsymbol{\theta}_k}}^{\mathtt{GAE}(\gamma,\lambda)}(s, a) \right\}, \tag{35}$$

$\epsilon$ is a hyperparameter which roughly says how far away the policy $\pi_{\boldsymbol{\theta}_{k+\frac{1}{2}}}$ is allowed to go from the current policy $\pi_{\boldsymbol{\theta}_k}$. The objective $\mathcal{L}_{\mathrm{clip}}(s, a, \pi_{\boldsymbol{\theta}}, \pi_{\boldsymbol{\theta}_k}, \epsilon)$ is complex, we present the insights of this clip mechanism Schulman *et al.* [2017] to make CUP learn stably.

**Positive GAE:** $A_{\pi_{\boldsymbol{\theta}_k}}^{\mathtt{GAE}(\gamma,\lambda)}(s, a) > 0$. Firstly, we consider the positive advantage, which implies the objective $\mathcal{L}_{\mathrm{clip}}(s, a, \pi_{\boldsymbol{\theta}}, \pi_{\boldsymbol{\theta}_k}, \epsilon)$ reduces to

$$\mathcal{L}_{\mathrm{clip}}(s, a, \pi_{\boldsymbol{\theta}}, \pi_{\boldsymbol{\theta}_k}, \epsilon) = \min \left\{ \frac{\pi_{\boldsymbol{\theta}}(a|s)}{\pi_{\boldsymbol{\theta}_k}(a|s)}, 1 + \epsilon \right\} A_{\pi_{\boldsymbol{\theta}_k}}^{\mathtt{GAE}(\gamma,\lambda)}(s, a). \tag{36}$$

Since $A_{\pi_{\boldsymbol{\theta}_k}}^{\mathtt{GAE}(\gamma,\lambda)}(s, a) > 0$, to improve the performance, we need to increase $\pi_{\boldsymbol{\theta}}$. The $\min\{\cdot\}$ operator determines the quantization how much the CUP improves. If the policy improves too much such that

$$\pi_{\boldsymbol{\theta}}(a|s) > (1 + \epsilon)\pi_{\boldsymbol{\theta}_k}(a|s),$$

The $\min\{\cdot\}$ operator hit the objective with a ceiling of $(1 + \epsilon)A_{\pi_{\boldsymbol{\theta}_k}}^{\mathtt{GAE}(\gamma,\lambda)}(s, a)$. The clip technique requires CUP learns a policy $\pi_{\boldsymbol{\theta}_{k+\frac{1}{2}}}$ does not benefit by going far away from the current policy $\pi_{\boldsymbol{\theta}_k}$.

**Negative GAE:** $A_{\pi_{\theta_k}}^{\text{GAE}(\gamma,\lambda)}(s,a) < 0$. Let us consider the negative advantage, which implies the objective $\mathcal{L}_{\text{clip}}(s,a,\pi_{\theta},\pi_{\theta_k},\epsilon)$ reduces to

$$\mathcal{L}_{\text{clip}}(s,a,\pi_{\theta},\pi_{\theta_k},\epsilon) = \max\left\{\frac{\pi_{\theta}(a|s)}{\pi_{\theta_k}(a|s)}, 1-\epsilon\right\} A_{\pi_{\theta_k}}^{\text{GAE}(\gamma,\lambda)}(s,a). \tag{37}$$

Since $A_{\pi_{\theta_k}}^{\text{GAE}(\gamma,\lambda)}(s,a) < 0$, to improve the performance, we need to decrease the policy $\pi_{\theta}(a|s)$. The $\max\{\cdot\}$ operator determines the quantization how much the CUP improves. If the policy decrease too much such that

$$\pi_{\theta}(a|s) < (1-\epsilon)\pi_{\theta_k}(a|s),$$

The $\max\{\cdot\}$ operator hit the objective with a ceiling of $(1-\epsilon)A_{\pi_{\theta_k}}^{\text{GAE}(\gamma,\lambda)}(s,a)$. Thus, similar to the positive GAE the clip technique requires CUP learns a policy $\pi_{\theta_{k+\frac{1}{2}}}$ does not benefit by going far away from the current policy $\pi_{\theta_k}$.

### C.1.3  Learning from Sampling

To short the expression, we introduce a function $g(\epsilon, A)$ as follows,

$$g(\epsilon, A) = \begin{cases} (1+\epsilon)A & A \geq 0 \\ (1+\epsilon)A & A < 0. \end{cases}$$

Then we rewrite the objective (35) as follows,

$$\mathcal{L}_{\text{clip}}(s,a,\pi_{\theta},\pi_{\theta_k},\epsilon) = \min\left\{\frac{\pi_{\theta}(a|s)}{\pi_{\theta_k}(a|s)}A_{\pi_{\theta_k}}^{\text{GAE}(\gamma,\lambda)}(s,a), g\left(\epsilon, A_{\pi_{\theta_k}}^{\text{GAE}(\gamma,\lambda)}(s,a)\right)\right\}. \tag{38}$$

Recall the trajectory $\{(s_t, a_t, r_{t+1}, c_{t+1})\}_{t=1}^T$ be sampled according to $\pi_{\theta_k}$, we defined the following $\hat{\mathcal{L}}_{\text{clip}}(\pi_{\theta}, \pi_{\theta_k})$,

$$\hat{\mathcal{L}}_{\text{clip}}(\pi_{\theta}, \pi_{\theta_k}, \epsilon) = \min\left\{\frac{1}{T}\sum_{t=1}^T \frac{\pi_{\theta}(a_t|s_t)}{\pi_{\theta_k}(a_t|s_t)}\hat{A}_t - g\left(\epsilon, \frac{1}{T}\sum_{t=1}^T \hat{A}_t\right)\right\}, \tag{39}$$

where $\hat{A}_t$ is an estimator of $A_{\pi_{\theta_k}}^{\text{GAE}(\gamma,\lambda)}(s,a)$. The term $\hat{\mathcal{L}}_{\text{clip}}(\pi_{\theta}, \pi_{\theta_k}, \epsilon)$ (39) is an estimator of the next expectation that appears in (38).

Then we implement the performance improvement as follows,

$$\pi_{\theta_{k+\frac{1}{2}}} = \arg\max_{\pi_{\theta}\in\Pi_{\theta}}\left\{\hat{\mathcal{L}}_{\text{clip}}(\pi_{\theta}, \pi_{\theta_k}, \epsilon)\right\},$$

i.e., we obtain the parameter $\theta_{k+\frac{1}{2}}$ according to

$$\theta_{k+\frac{1}{2}} = \theta_k + \eta_1 \frac{\partial}{\partial\theta}\hat{\mathcal{L}}_{\text{clip}}(\pi_{\theta}, \pi_{\theta_k}, \epsilon)\Big|_{\theta=\theta_k},$$

where $\eta_1$ is step-size.

### C.2  Practical Implementation of Projection

Recall (15), we introduce the new surrogate function with respected to cost function as follows,

$$C_{\pi_{\theta'}}(\pi_{\theta}, \beta) = J^c(\pi_{\theta'}) + \frac{1}{1-\tilde{\gamma}}\mathbb{E}_{s\sim d_{\pi_{\theta'}}^{\lambda}(\cdot), a\sim\pi_{\theta}(\cdot|s)}\left[A_{\pi_{\theta'},C}^{\text{GAE}(\gamma,\lambda)}(s,a) + \beta\sqrt{\mathbb{E}_{s\sim d_{\pi_{\theta'}}^{\lambda}(\cdot)}[\text{KL}(\pi_{\theta'}, \pi_{\theta})[s]]}\right],$$

where $\beta$ is adaptive to the term $\frac{\sqrt{2}\tilde{\gamma}(\gamma\lambda(|\mathcal{S}|-1)+1)\epsilon_{\pi_{\theta}}^V(\pi_{\theta'})}{(1-\gamma\lambda)}$. Now, we rewrite the projection step (15) as follows,

$$\pi_{\theta_{k+1}} = \arg\min_{\pi_{\theta}\in\Pi_{\theta}} D\left(\pi_{\theta}, \pi_{\theta_{k+\frac{1}{2}}}\right), \quad \text{s.t. } C_{\pi_{\theta_k}}(\pi_{\theta}, \beta) \leq b. \tag{40}$$

We update the projection step (15) by replacing the distance function $D(\cdot, \cdot)$ by KL-divergence, and we solve the constraint problem (15) by the primal-dual approach.

**Theorem 4.** *The constrained problem (40) is equivalent to the following primal-dual problem:*

$$\max_{\nu \geq 0} \min_{\pi_{\boldsymbol{\theta}} \in \Pi_{\boldsymbol{\theta}}} \left\{ D\left(\pi_{\boldsymbol{\theta}}, \pi_{\boldsymbol{\theta}_{k+\frac{1}{2}}}\right) + \nu\left(C_{\pi_{\boldsymbol{\theta}_k}}(\pi_{\boldsymbol{\theta}}, \beta) - b\right)\right\}.$$

*Proof.* This result is a direct application of [Boyd and Vandenberghe, 2004, Chapter 5.9], and we also present it in D.1. Firstly, we notice if $D\left(\cdot, \pi_{\boldsymbol{\theta}_{k+\frac{1}{2}}}\right)$ is KL divergence or $\ell_2$-norm, then the constrained problem (40) is a convex problem [2]. In fact, for a given policy $\pi_{\boldsymbol{\theta}_{k+\frac{1}{2}}}$, $D\left(\cdot, \pi_{\boldsymbol{\theta}_{k+\frac{1}{2}}}\right)$ is convex over the policy $\Pi_{\boldsymbol{\theta}}$, and $C_{\pi_{\boldsymbol{\theta}'}}(\cdot, \beta)$ is also convex over the policy $\Pi_{\boldsymbol{\theta}}$. Additionally, Slater's condition alway holds since $C_{\pi_{\boldsymbol{\theta}'}}(\pi_{\boldsymbol{\theta}'}, \beta) = 0$. $\qquad\square$

According to Theorem 4, we turn the projection step (40) as the following unconstrained problem,

$$\max_{\nu \geq 0} \min_{\pi_{\boldsymbol{\theta}} \in \Pi_{\boldsymbol{\theta}}} \left\{ D\left(\pi_{\boldsymbol{\theta}}, \pi_{\boldsymbol{\theta}_{k+\frac{1}{2}}}\right) + \nu\left(C_{\pi_{\boldsymbol{\theta}_k}}(\pi_{\boldsymbol{\theta}}, \beta) - b\right)\right\}. \tag{41}$$

In our implementation, we use KL-divergence as the distance $D(\cdot, \cdot)$ to measure the difference between two policies, then

$$D\left(\pi_{\boldsymbol{\theta}}, \pi_{\boldsymbol{\theta}_{k+\frac{1}{2}}}\right) = \mathbb{E}_{s \sim d^\lambda_{\pi_{\boldsymbol{\theta}_k}}(\cdot)}\left[\mathrm{KL}\left(\pi_{\boldsymbol{\theta}_{k+\frac{1}{2}}}, \pi_{\boldsymbol{\theta}}\right)[s]\right], \tag{42}$$

which implies we can rewrite the problem (41) as follows,

$$\max_{\nu \geq 0} \min_{\pi_{\boldsymbol{\theta}}} \left\{ \mathbb{E}_{s \sim d^\lambda_{\pi_{\boldsymbol{\theta}_k}}(\cdot)}\left[\mathrm{KL}\left(\pi_{\boldsymbol{\theta}_{k+\frac{1}{2}}}, \pi_{\boldsymbol{\theta}}\right)[s]\right] + \nu\left(C_{\pi_{\boldsymbol{\theta}_k}}(\pi_{\boldsymbol{\theta}}, \beta) - b\right)\right\}. \tag{43}$$

Furthermore, we update the projection step as follows,

$$(\pi_{\boldsymbol{\theta}_{k+1}}, \nu_{k+1}) = \arg \min_{\pi_{\boldsymbol{\theta}} \in \Pi_{\boldsymbol{\theta}}} \max_{\nu \geq 0} \left\{ \hat{\mathcal{L}}_{\mathrm{c}}\left(\pi_{\boldsymbol{\theta}}, \pi_{\boldsymbol{\theta}_k}, \boldsymbol{\theta}_{k+\frac{1}{2}}, \nu\right)\right\},$$

where

$$\hat{\mathcal{L}}_{\mathrm{c}}\left(\pi_{\boldsymbol{\theta}}, \pi_{\boldsymbol{\theta}_k}, \boldsymbol{\theta}_{k+\frac{1}{2}}, \nu\right) = \hat{D}_{\mathrm{KL}}(\pi_{\boldsymbol{\theta}_{k+\frac{1}{2}}}, \pi_{\boldsymbol{\theta}}) + \nu \hat{C}(\pi_{\boldsymbol{\theta}}, \pi_{\boldsymbol{\theta}_k}),$$

$$\hat{C}(\pi_{\boldsymbol{\theta}}, \pi_{\boldsymbol{\theta}_k}) = \hat{J}^C + \frac{1}{1-\tilde{\gamma}} \cdot \frac{1}{T} \sum_{t=1}^{T} \frac{\pi_{\boldsymbol{\theta}}(a_t|s_t)}{\pi_{\boldsymbol{\theta}_k}(a_t|s_t)} \hat{A}^C_t + \beta_k \sqrt{\hat{D}_{\mathrm{KL}}(\pi_{\boldsymbol{\theta}_k}, \pi_{\boldsymbol{\theta}})} - b,$$

$\hat{J}^C$ and $\hat{A}^C_t$ are estimators for cost-return and cost-advantage.

**Remark 3** (Track for Learning $\nu$). *Particularly, after some simple algebra, we obtain the derivation of $\hat{\mathcal{L}}_c(\cdot)$ with respect to $\nu$ as follows,*

$$\frac{\partial \hat{\mathcal{L}}_c\left(\pi_{\boldsymbol{\theta}}, \pi_{\boldsymbol{\theta}_k}, \boldsymbol{\theta}_{k+\frac{1}{2}}, \nu\right)}{\partial \nu} = J^c(\pi_{\boldsymbol{\theta}_k}) + \frac{1}{1-\tilde{\gamma}} \mathbb{E}_{s \sim d^\lambda_{\pi_{\boldsymbol{\theta}_k}}(\cdot), a \sim \pi_{\boldsymbol{\theta}}(\cdot|s)}\left[A^{\mathtt{GAE}(\gamma,\lambda)}_{\pi_{\boldsymbol{\theta}_k}, C}(s, a)\right] - b. \tag{44}$$

*But recall (14) is a minimization-maximization iteration, i.e., we require to minimize the distance $\mathbb{E}_{s \sim d^\lambda_{\pi_{\boldsymbol{\theta}_k}}(\cdot)}\mathrm{KL}(\pi_{\boldsymbol{\theta}}, \pi_{\boldsymbol{\theta}_k})[s]$, which implies $\pi_{\boldsymbol{\theta}}$ is close to $\pi_{\boldsymbol{\theta}_k}$. Thus it is reasonable to consider*

$$\mathbb{E}_{s \sim d^\lambda_{\pi_{\boldsymbol{\theta}_k}}(\cdot), a \sim \pi_{\boldsymbol{\theta}}(\cdot|s)}\left[A^{\mathtt{GAE}(\gamma,\lambda)}_{\pi_{\boldsymbol{\theta}_k}, C}(s, a)\right] \approx 0.$$

*Thus, in practice, we update $\nu$ following a simple way*

$$\nu \leftarrow \left\{\nu + \eta(J^c(\pi_{\boldsymbol{\theta}_k}) - b)\right\}_+.$$

Finally, we obtain the parameters $(\boldsymbol{\theta}_{k+1}, \nu_{k+1})$ as follows,

$$\boldsymbol{\theta}_{k+1} \leftarrow \boldsymbol{\theta}_k - \eta_2 \frac{\partial}{\partial \boldsymbol{\theta}} \hat{\mathcal{L}}_{\mathrm{c}}\left(\pi_{\boldsymbol{\theta}}, \pi_{\boldsymbol{\theta}_k}, \boldsymbol{\theta}_{k+\frac{1}{2}}, \nu\right)\Big|_{\boldsymbol{\theta}=\boldsymbol{\theta}_k, \nu=\nu_k}, \tag{45}$$

$$\nu_{k+1} \leftarrow \left\{\nu_k + \eta_2 \frac{\partial}{\partial \nu} \hat{\mathcal{L}}_{\mathrm{c}}\left(\pi_{\boldsymbol{\theta}}, \pi_{\boldsymbol{\theta}_k}, \boldsymbol{\theta}_{k+\frac{1}{2}}, \nu\right)\Big|_{\boldsymbol{\theta}=\boldsymbol{\theta}_k, \nu=\nu_k}\right\}_+, \tag{46}$$

where $\{\cdot\}_+$ denotes the positive part, i.e., if $x \leq 0$, $\{x\}_+ = 0$, else $\{x\}_+ = x$. We have shown all the details of the implementation in Algorithm 1.

---

[2]It is worth noting that $\min_{\pi_{\boldsymbol{\theta}} \in \Pi_{\boldsymbol{\theta}}} D\left(\pi_{\boldsymbol{\theta}}, \pi_{\boldsymbol{\theta}_{k+\frac{1}{2}}}\right)$ is a convex problem, while $\min_{\boldsymbol{\theta} \in \mathbb{R}^p} D\left(\pi_{\boldsymbol{\theta}}, \pi_{\boldsymbol{\theta}_{k+\frac{1}{2}}}\right)$ can be a non-convex problem.

**Algorithm 1** Constrained Update Projection (CUP)

---

**Initialize:** policy network parameters $\boldsymbol{\theta}_0$; value network parameter $\boldsymbol{\omega}_0$; cost value function parameter $\boldsymbol{\nu}_0$, step-size $\nu_0$;

**Hyper-parameters:** trajectory horizon $T$; discount rate $\gamma$; episode number $M, N$, mini-batch size $B$, positive constant $\alpha, \eta$;

**for** $k = 0, 1, 2, \ldots$ **do**

    Collect batch data of $M$ episodes of horizon $T$ in $\cup_{i=1}^{M}\cup_{t=0}^{T}\{(s_{i,t}, a_{i,t}, r_{i,t+1}, c_{i,t+1})\}$ according to current policy $\pi_{\boldsymbol{\theta}_k}$;

    Estimate $c$-return by discount averaging on each episode: $\hat{J}_i^C = \sum_{t=0}^{T} \gamma^t c_{i,t+1}$;

    Compute TD errors $\cup_{i=1}^{M}\cup_{t=0}^{T}\{\delta_{i,t}\}$, cost TD errors $\cup_{i=1}^{M}\cup_{t=0}^{T}\{\delta_{i,t}^C\}$:

$$\delta_{i,t} = r_{i,t} + \gamma V_{\boldsymbol{\omega}_k}(s_{i,t}) - V_{\boldsymbol{\omega}_k}(s_{i,t-1}), \ \delta_{i,t}^C = c_{i,t} + \gamma V_{\boldsymbol{\nu}_k}^C(s_{i,t}) - V_{\boldsymbol{\nu}_k}^C(s_{i,t-1});$$

    Compute GAE: $\cup_{i=1}^{M}\cup_{t=0}^{T}\{\hat{A}_{i,t}, \hat{A}_{i,t}^C\}$: $\hat{A}_{i,t} = \sum_{j=t}^{T}(\gamma\lambda)^{j-t}\delta_{i,j}$, $\hat{A}_{i,t}^C = \sum_{j=t}^{T}(\gamma\lambda)^{j-t}\delta_{i,j}^C$;

    Compute target function for value function and cost value function as follows,

$$V_{i,t}^{\text{target}} = \hat{A}_{i,t} + V_{\boldsymbol{\omega}_k}(s_{i,t}), \ \ V_{i,t}^{\text{target},C} = \hat{A}_{i,t}^C + V_{\boldsymbol{\nu}_k}^C(s_{i,t});$$

    Store data: $\mathcal{D}_k = \cup_{i=1}^{M}\cup_{t=0}^{T}\left\{(a_{i,t}, s_{i,t}, \hat{A}_{i,t}, \hat{A}_{i,t}^C, V_{i,t}^{\text{target}}, V_{i,t}^{\text{target},C})\right\}$;

    $\pi_{\text{old}} \leftarrow \pi_{\boldsymbol{\theta}_k}$;                                                      Policy Improvement

    **for** $i = 0, 1, 2, \ldots, M$ **do**

$$\boldsymbol{\theta}_{k+\frac{1}{2}} = \arg\max_{\boldsymbol{\theta}}\left\{\frac{1}{T}\sum_{t=1}^{T}\frac{\pi_{\boldsymbol{\theta}}(a_{i,t}|s_{i,t})}{\pi_{\text{old}}(a_{i,t}|s_{i,t})}\hat{A}_{i,t} - g\left(\epsilon, \frac{1}{T}\sum_{t=1}^{T}\hat{A}_{i,t}\right)\right\};$$

    **end for**

    $\pi_{\text{old}} \leftarrow \pi_{\boldsymbol{\theta}_{k+\frac{1}{2}}}$;                                                                 Projection

    $\nu_{k+1} = (\nu_k + \eta(\hat{J}_i^C - b))_+$;

    **for** $i = 0, 1, 2, \ldots, M$ **do**

$$\boldsymbol{\theta}_{k+1} = \arg\min_{\boldsymbol{\theta}}\frac{1}{T}\sum_{t=1}^{T}\left\{\text{KL}(\pi_{\boldsymbol{\theta}_{\text{old}}}(\cdot|s_{i,t}), \pi_{\boldsymbol{\theta}}(\cdot|s_{i,t})) + \nu_k\frac{1-\gamma\lambda}{1-\gamma}\frac{\pi_{\boldsymbol{\theta}}(a_{i,t}|s_{i,t})}{\pi_{\boldsymbol{\theta}_k}(a_{i,t}|s_{i,t})}\hat{A}_{i,t}^C\right\};$$

    **end for**

    **for** each mini-batch $\{(a_j, s_j, \hat{A}_j, \hat{A}_j^C, V_j^{\text{target}}, V_j^{\text{target},C})\}$ of size $B$ from $\mathcal{D}_k$ **do**

$$\boldsymbol{\omega}_{k+1} = \arg\min_{\boldsymbol{\omega}}\sum_{j=1}^{B}\left(V_{\boldsymbol{\omega}}(s_j) - V_j^{\text{target}}\right)^2, \boldsymbol{\nu}_{k+1} = \arg\min_{\boldsymbol{\nu}}\sum_{j=1}^{B}\left(V_{\boldsymbol{\nu}}^c(s_j) - V_j^{\text{target},C}\right)^2;$$

    **end for**

**end for**

# D   Preliminaries

In this section, we introduce some new notations and results about convex optimization, state distribution, policy optimization and $\lambda$-returns.

## D.1   Strong Duality via Slater's Condition

We consider a convex optimization problem:

$$p_\star = \min_x f_0(x), \tag{47}$$

$$\text{s.t. } f_i(x) \leq 0, \ i = 1, 2, \cdots, m, \tag{48}$$

$$h_i(x) = 0, \ i = 1, 2, \cdots, p, \tag{49}$$

where the functions $f_0, f_1, \cdots, f_m$ are convex, and $h_1, \cdots, h_p$ are affine. We denote by $\mathcal{D}$ the domain of the problem (which is the intersection of the domains of all the functions involved), and by $\mathcal{X} \subset \mathcal{D}$ its feasible set.

To the problem we associate the Lagrangian $\mathcal{L} : \mathbb{R}^n \times \mathbb{R}^m \times \mathbb{R}^p \to \mathbb{R}$, with values

$$\mathcal{L}(x, \lambda, \nu) = f_0 + \sum_{i=1}^m \lambda_i f_i(x) + \sum_{i=1}^p \nu_i h_i(x). \tag{50}$$

The dual function is $g : \mathbb{R}^m \times \mathbb{R}^p \to \mathbb{R}$, with values

$$g(\lambda, \nu) = \min_x \mathcal{L}(x, \lambda, \nu). \tag{51}$$

The associated dual problem is

$$d_\star = \max_{\lambda \succeq 0, \nu} g(\lambda, \nu). \tag{52}$$

**Slater's condition**. We say that the problem satisfies Slater's condition if it is strictly feasible, that is:

$$\exists x_0 \in \mathcal{D} : f_i(x_0) < 0, i = 1, \cdots, m, \ h_i(x_0) = 0, i = 1, \cdots, p. \tag{53}$$

**Theorem 5** (Strong duality via Slater condition). *If the primal problem (8.1) is convex, and satisfies the weak Slater's condition, then strong duality holds, that is, $p_\star = d_\star$.*

We omit the proof of Theorem 5, for more discussions, please refer to [Boyd and Vandenberghe, 2004, Chapter 5.9].

## D.2   State Distribution

We use $\mathbf{P}_{\pi_\theta} \in \mathbb{R}^{|\mathcal{S}| \times |\mathcal{S}|}$ to denote the state transition matrix by executing $\pi_\theta$, and their components are:

$$\mathbf{P}_{\pi_\theta}[s, s'] = \sum_{a \in \mathcal{A}} \pi_\theta(a|s) \mathbb{P}(s'|s, a) =: \mathbb{P}_{\pi_\theta}(s'|s), \ s, s' \in \mathcal{S},$$

which denotes one-step state transformation probability from $s$ to $s'$.

We use $\mathbb{P}_{\pi_\theta}(s_t = s|s_0)$ to denote the probability of visiting $s$ after $t$ time steps from the initial state $s_0$ by executing $\pi_\theta$. Particularly, we notice if $t = 0$, $s_t \neq s_0$, then $\mathbb{P}_{\pi_\theta}(s_t = s|s_0) = 0$, i.e.,

$$\mathbb{P}_{\pi_\theta}(s_t = s|s_0) = 0, \ t = 0 \text{ and } s \neq s_0. \tag{54}$$

Then for any initial state $s_0 \sim \rho(\cdot)$, the following holds,

$$\mathbb{P}_{\pi_\theta}(s_t = s|s_0) = \sum_{s' \in \mathcal{S}} \mathbb{P}_{\pi_\theta}(s_t = s|s_{t-1} = s') \mathbb{P}_{\pi_\theta}(s_{t-1} = s'|s_0). \tag{55}$$

Recall $d_{\pi_\theta}^{s_0}(s)$ denotes the normalized discounted distribution of the future state $s$ encountered starting at $s_0$ by executing $\pi_\theta$,

$$d_{\pi_\theta}^{s_0}(s) = (1 - \gamma) \sum_{t=0}^\infty \gamma^t \mathbb{P}_{\pi_\theta}(s_t = s|s_0).$$

Furthermore, since $s_0 \sim \rho_0(\cdot)$, we define

$$d_{\pi_{\boldsymbol{\theta}}}^{\rho_0}(s) = \mathbb{E}_{s_0 \sim \rho_0(\cdot)}[d_{\pi_{\boldsymbol{\theta}}}^{s_0}(s)] = \int_{s_0 \in \mathcal{S}} \rho_0(s_0) d_{\pi_{\boldsymbol{\theta}}}^{s_0}(s) \mathrm{d}s_0$$

as the discounted state visitation distribution over the initial distribution $\rho_0(\cdot)$. We use $\mathbf{d}_{\pi_{\boldsymbol{\theta}}}^{\rho_0} \in \mathbb{R}^{|\mathcal{S}|}$ to store all the normalized discounted state distributions, and its components are:

$$\mathbf{d}_{\pi_{\boldsymbol{\theta}}}^{\rho_0}[s] = d_{\pi_{\boldsymbol{\theta}}}^{\rho_0}(s), \quad s \in \mathcal{S}.$$

We use $\boldsymbol{\rho}_0 \in \mathbb{R}^{|\mathcal{S}|}$ to denote initial state distribution vector, and their components are:

$$\boldsymbol{\rho}_0[s] = \rho_0(s), \quad s \in \mathcal{S}.$$

Then, we rewrite $\mathbf{d}_{\pi_{\boldsymbol{\theta}}}^{\rho_0}$ as the following matrix version,

$$\mathbf{d}_{\pi_{\boldsymbol{\theta}}}^{\rho_0} = (1 - \gamma) \sum_{t=0}^{\infty} (\gamma \mathbf{P}_{\pi_{\boldsymbol{\theta}}})^t \boldsymbol{\rho}_0 = (1 - \gamma)(\mathbf{I} - \gamma \mathbf{P}_{\pi_{\boldsymbol{\theta}}})^{-1} \boldsymbol{\rho}_0. \tag{56}$$

### D.3 Objective of MDP

Recall $\tau = \{s_t, a_t, r_{t+1}\}_{t \geq 0} \sim \pi_{\boldsymbol{\theta}}$, according to $\tau$, we define the expected return $J(\pi_{\boldsymbol{\theta}}|s_0)$ as follows,

$$J(\pi_{\boldsymbol{\theta}}|s_0) = \mathbb{E}_{\tau \sim \pi_{\boldsymbol{\theta}}}[R(\tau)] = \frac{1}{1-\gamma} \mathbb{E}_{s \sim d_{\pi_{\boldsymbol{\theta}}}^{s_0}(\cdot), a \sim \pi_{\boldsymbol{\theta}}(\cdot|s), s' \sim \mathbb{P}(\cdot|s,a)} \left[ r(s'|s, a) \right], \tag{57}$$

where $R(\tau) = \sum_{t \geq 0} \gamma^t r_{t+1}$, and the notation $J(\pi_{\boldsymbol{\theta}}|s_0)$ is "conditional" on $s_0$ is to emphasize the trajectory $\tau$ starting from $s_0$.

Since $s_0 \sim \rho_0(\cdot)$, we define the objective of MDP as follows,

$$J(\pi_{\boldsymbol{\theta}}) = \frac{1}{1-\gamma} \mathbb{E}_{s \sim d_{\pi_{\boldsymbol{\theta}}}^{\rho_0}(\cdot), a \sim \pi_{\boldsymbol{\theta}}(\cdot|s), s' \sim \mathbb{P}(\cdot|s,a)} \left[ r(s'|s, a) \right]. \tag{58}$$

The goal of reinforcement learning is to solve the following optimization problem:

$$\boldsymbol{\theta}_\star = \arg\max_{\boldsymbol{\theta} \in \mathbb{R}^p} J(\pi_{\boldsymbol{\theta}}). \tag{59}$$

### D.4 $\lambda$-Return

Let $\mathcal{B}_{\pi_{\boldsymbol{\theta}}}$ be the *Bellman operator*:

$$\mathcal{B}_{\pi_{\boldsymbol{\theta}}} : \mathbb{R}^{|\mathcal{S}|} \to \mathbb{R}^{|\mathcal{S}|}, \quad v \mapsto \mathbf{r}_{\pi_{\boldsymbol{\theta}}} + \gamma \mathbf{P}_{\pi_{\boldsymbol{\theta}}} v, \tag{60}$$

where $\mathbf{r}_{\pi_{\boldsymbol{\theta}}} \in \mathbb{R}^{|\mathcal{S}|}$ is the expected reward according to $\pi_{\boldsymbol{\theta}}$, i.e., their components are:

$$\mathbf{r}_{\pi_{\boldsymbol{\theta}}}[s] = \sum_{a \in \mathcal{A}} \sum_{s' \in \mathcal{S}} \pi_{\boldsymbol{\theta}}(a|s) r(s'|s, a) =: R_{\pi_{\boldsymbol{\theta}}}(s), \quad s \in \mathcal{S}.$$

Let $\mathbf{v}_{\pi_{\boldsymbol{\theta}}} \in \mathbb{R}^{|\mathcal{S}|}$ be a vector that stores all the state value functions, and its components are:

$$\mathbf{v}_{\pi_{\boldsymbol{\theta}}}[s] = V_{\pi_{\boldsymbol{\theta}}}(s), \quad s \in \mathcal{S}.$$

Then, according to Bellman operator (60), we rewrite Bellman equation [Bellman, 1957] as the following matrix version:

$$\mathcal{B}_{\pi_{\boldsymbol{\theta}}} \mathbf{v}_{\pi_{\boldsymbol{\theta}}} = \mathbf{v}_{\pi_{\boldsymbol{\theta}}}. \tag{61}$$

Furthermore, we define $\lambda$-*Bellman operator* $\mathcal{B}_{\pi_{\boldsymbol{\theta}}}^{\lambda}$ as follows,

$$\mathcal{B}_{\pi_{\boldsymbol{\theta}}}^{\lambda} = (1 - \lambda) \sum_{t=0}^{\infty} \lambda^t (\mathcal{B}_{\pi_{\boldsymbol{\theta}}})^{t+1},$$

which implies

$$\mathcal{B}_{\pi_\theta}^\lambda : \mathbb{R}^{|\mathcal{S}|} \to \mathbb{R}^{|\mathcal{S}|}, \quad v \mapsto \mathbf{r}_{\pi_\theta}^{(\lambda)} + \tilde{\gamma} \mathbf{P}_{\pi_\theta}^{(\lambda)} v, \tag{62}$$

where

$$\mathbf{P}_{\pi_\theta}^{(\lambda)} = (1 - \gamma\lambda) \sum_{t=0}^{\infty} (\gamma\lambda)^t \mathbf{P}_{\pi_\theta}^{t+1}, \quad \mathbf{r}_{\pi_\theta}^{(\lambda)} = \sum_{t=0}^{\infty} (\gamma\lambda \mathbf{P}_{\pi_\theta})^t \mathbf{r}_{\pi_\theta}, \quad \tilde{\gamma} = \frac{\gamma(1-\lambda)}{1-\gamma\lambda}. \tag{63}$$

Let

$$\mathbb{P}_{\pi_\theta}^{(\lambda)}(s'|s) = \mathbf{P}_{\pi_\theta}^{(\lambda)}[s, s'] =: (1 - \gamma\lambda) \sum_{t=0}^{\infty} (\gamma\lambda)^t \left( \mathbf{P}_{\pi_\theta}^{t+1}[s, s'] \right), \tag{64}$$

where $\mathbf{P}_{\pi_\theta}^{t+1}[s, s']$ is the $(s, s')$-th component of matrix $\mathbf{P}_{\pi_\theta}^{t+1}$, which is the probability of visiting $s'$ after $t + 1$ time steps from the state $s$ by executing $\pi_\theta$, i.e.,

$$\mathbf{P}_{\pi_\theta}^{t+1}[s, s'] = \mathbb{P}_{\pi_\theta}(s_{t+1} = s'|s). \tag{65}$$

Thus, we rewrite $\mathbb{P}_{\pi_\theta}^{(\lambda)}(s'|s)$ (64) as follows

$$\mathbb{P}_{\pi_\theta}^{(\lambda)}(s'|s) = (1 - \gamma\lambda) \sum_{t=0}^{\infty} (\gamma\lambda)^t \mathbb{P}_{\pi_\theta}(s_{t+1} = s'|s), \quad s \in \mathcal{S}. \tag{66}$$

**Remark 4.** *Furthermore, recall the following visitation sequence $\tau = \{s_t, a_t, r_{t+1}\}_{t\geq 0}$ induced by $\pi_\theta$, it is similar to the probability $\mathbb{P}_{\pi_\theta}(s_t = s'|s_0)$, we introduce $\mathbb{P}_{\pi_\theta}^{(\lambda)}(s_t = s'|s_0)$ as the probability of transition from state $s$ to state $s'$ after $t$ time steps under the dynamic transformation matrix $\mathbf{P}_{\pi_\theta}^{(\lambda)}$. Then, the following equity holds*

$$\mathbb{P}_{\pi_\theta}^{(\lambda)}(s_t = s|s_0) = \sum_{s' \in \mathcal{S}} \mathbb{P}_{\pi_\theta}^{(\lambda)}(s_t = s|s_{t-1} = s') \mathbb{P}_{\pi_\theta}^{(\lambda)}(s_{t-1} = s'|s_0). \tag{67}$$

Similarly, let

$$\begin{aligned} R_{\pi_\theta}^{(\lambda)}(s) =: \mathbf{r}_{\pi_\theta}^{(\lambda)}[s] = \sum_{t=0}^{\infty} (\gamma\lambda \mathbf{P}_{\pi_\theta})^t \mathbf{r}_{\pi_\theta}[s] = \sum_{t=0}^{\infty} (\gamma\lambda)^t \left( \sum_{s' \in \mathcal{S}} \mathbb{P}_{\pi_\theta}(s_t = s'|s) R_{\pi_\theta}(s') \right) \\ = \sum_{t=0}^{\infty} \sum_{s' \in \mathcal{S}} (\gamma\lambda)^t \mathbb{P}_{\pi_\theta}(s_t = s'|s) R_{\pi_\theta}(s'). \end{aligned} \tag{68}$$

It is similar to normalized discounted distribution $d_{\pi_\theta}^{\rho_0}(s)$, we introduce $\lambda$-return version of discounted state distribution $d_{\pi_\theta}^\lambda(s)$ as follows: $\forall s \in \mathcal{S}$,

$$d_{\pi_\theta}^{s_0, \lambda}(s) = (1 - \tilde{\gamma}) \sum_{t=0}^{\infty} \tilde{\gamma}^t \mathbb{P}_{\pi_\theta}^{(\lambda)}(s_t = s|s_0), \tag{69}$$

$$d_{\pi_\theta}^\lambda(s) = \mathbb{E}_{s_0 \sim \rho_0(\cdot)} \left[ d_{\pi_\theta}^{s_0, \lambda}(s) \right], \tag{70}$$

$$\mathbf{d}_{\pi_\theta}^\lambda[s] = d_{\pi_\theta}^\lambda(s), \tag{71}$$

where $\mathbb{P}_{\pi_\theta}^{(\lambda)}(s_t = s|s_0)$ is the $(s_0, s)$-th component of the matrix $\left( \mathbf{P}_{\pi_\theta}^{(\lambda)} \right)^t$, i.e.,

$$\mathbb{P}_{\pi_\theta}^{(\lambda)}(s_t = s|s_0) =: \left( \mathbf{P}_{\pi_\theta}^{(\lambda)} \right)^t [s_0, s].$$

Similarly, $\mathbb{P}_{\pi_\theta}^{(\lambda)}(s_t = s'|s)$ is the $(s, s')$-th component of the matrix $\left( \mathbf{P}_{\pi_\theta}^{(\lambda)} \right)^t$, i.e.,

$$\mathbb{P}_{\pi_\theta}^{(\lambda)}(s_t = s'|s) =: \left( \mathbf{P}_{\pi_\theta}^{(\lambda)} \right)^t [s, s'].$$

Finally, we rewrite $\mathbf{d}_{\pi_\theta}^{\rho_0, \lambda}$ as the following matrix version,

$$\mathbf{d}_{\pi_\theta}^\lambda = (1 - \tilde{\gamma}) \sum_{t=0}^{\infty} \left( \gamma \mathbf{P}_{\pi_\theta}^{(\lambda)} \right)^t \boldsymbol{\rho}_0 = (1 - \tilde{\gamma}) \left( \mathbf{I} - \tilde{\gamma} \mathbf{P}_{\pi_\theta}^{(\lambda)} \right)^{-1} \boldsymbol{\rho}_0. \tag{72}$$

**Remark 5** ($\lambda$-Return Version of Bellman Equation). *According to Bellman equation (61), $\mathbf{v}_{\pi_\theta}$ is fixed point of $\lambda$-operator $\mathcal{B}_{\pi_\theta}^\lambda$, i.e.,*

$$\mathbf{v}_{\pi_\theta} = \mathbf{r}_{\pi_\theta}^{(\lambda)} + \tilde{\gamma}\mathbf{P}_{\pi_\theta}^{(\lambda)}\mathbf{v}_{\pi_\theta}. \tag{73}$$

*Recall $\tau = \{s_t, a_t, r_{t+1}\}_{t \geq 0} \sim \pi_\theta$, according to (73), the value function of initial state $s_0$ is*

$$\begin{aligned}
V_{\pi_\theta}(s_0) &= \mathbf{v}_{\pi_\theta}[s_0] = \mathbf{r}_{\pi_\theta}^{(\lambda)}[s_0] + \tilde{\gamma}\mathbf{P}_{\pi_\theta}^{(\lambda)}\mathbf{v}_{\pi_\theta}[s_0] \\
&= R_{\pi_\theta}^{(\lambda)}(s_0) + \tilde{\gamma}\sum_{s' \in \mathcal{S}}\mathbb{P}_{\pi_\theta}^{(\lambda)}(s_1 = s'|s_0)V_{\pi_\theta}(s').
\end{aligned} \tag{74}$$

We unroll the expression of (74) repeatedly, then we have

$$\begin{aligned}
&V_{\pi_\theta}(s_0) \\
=&R_{\pi_\theta}^{(\lambda)}(s_0) + \tilde{\gamma}\sum_{s' \in \mathcal{S}}\mathbb{P}_{\pi_\theta}^{(\lambda)}(s_1 = s'|s_0)\underbrace{\left(R_{\pi_\theta}^{(\lambda)}(s') + \tilde{\gamma}\sum_{s'' \in \mathcal{S}}\mathbb{P}_{\pi_\theta}^{(\lambda)}(s_2 = s''|s_1 = s')V_{\pi_\theta}(s'')\right)}_{=V_{\pi_\theta}(s')} \\
=&R_{\pi_\theta}^{(\lambda)}(s_0) + \tilde{\gamma}\sum_{s' \in \mathcal{S}}\mathbb{P}_{\pi_\theta}^{(\lambda)}(s_1 = s'|s_0)R_{\pi_\theta}^{(\lambda)}(s') \\
&\qquad + \tilde{\gamma}^2\sum_{s'' \in \mathcal{S}}\underbrace{\left(\sum_{s' \in \mathcal{S}}\mathbb{P}_{\pi_\theta}^{(\lambda)}(s_1 = s'|s_0)\mathbb{P}_{\pi_\theta}^{(\lambda)}(s_2 = s''|s_1 = s')\right)}_{\overset{(67)}{=}:\mathbb{P}_{\pi_\theta}^{(\lambda)}(s_2 = s''|s_0)}V_{\pi_\theta}(s'') \\
=&R_{\pi_\theta}^{(\lambda)}(s_0) + \tilde{\gamma}\sum_{s \in \mathcal{S}}\mathbb{P}_{\pi_\theta}^{(\lambda)}(s_1 = s|s_0)R_{\pi_\theta}^{(\lambda)}(s) + \tilde{\gamma}^2\sum_{s \in \mathcal{S}}\mathbb{P}_{\pi_\theta}^{(\lambda)}(s_2 = s|s_0)V_{\pi_\theta}(s) \\
=&R_{\pi_\theta}^{(\lambda)}(s_0) + \tilde{\gamma}\sum_{s \in \mathcal{S}}\mathbb{P}_{\pi_\theta}^{(\lambda)}(s_1 = s|s_0)R_{\pi_\theta}^{(\lambda)}(s) \\
&\qquad + \tilde{\gamma}^2\sum_{s \in \mathcal{S}}\mathbb{P}_{\pi_\theta}^{(\lambda)}(s_2 = s|s_0)\left(R_{\pi_\theta}^{(\lambda)}(s) + \tilde{\gamma}\sum_{s' \in \mathcal{S}}\mathbb{P}_{\pi_\theta}^{(\lambda)}(s_3 = s'|s_2 = s)V_{\pi_\theta}(s')\right) \\
=&R_{\pi_\theta}^{(\lambda)}(s_0) + \tilde{\gamma}\sum_{s \in \mathcal{S}}\mathbb{P}_{\pi_\theta}^{(\lambda)}(s_1 = s|s_0)R_{\pi_\theta}^{(\lambda)}(s) + \tilde{\gamma}^2\sum_{s \in \mathcal{S}}\mathbb{P}_{\pi_\theta}^{(\lambda)}(s_2 = s|s_0)R_{\pi_\theta}^{(\lambda)}(s) \\
&\qquad + \tilde{\gamma}^3\sum_{s' \in \mathcal{S}}\underbrace{\left(\sum_{s \in \mathcal{S}}\mathbb{P}_{\pi_\theta}^{(\lambda)}(s_2 = s|s_0)\mathbb{P}_{\pi_\theta}^{(\lambda)}(s_3 = s'|s_2 = s)\right)}_{=\mathbb{P}_{\pi_\theta}^{(\lambda)}(s_3 = s'|s_0)}V_{\pi_\theta}(s') \\
=&R^{(\lambda)}(s_0) + \tilde{\gamma}\sum_{s \in \mathcal{S}}\mathbb{P}_{\pi_\theta}^{(\lambda)}(s_1 = s|s_0)R_{\pi_\theta}^{(\lambda)}(s) + \tilde{\gamma}^2\sum_{s \in \mathcal{S}}\mathbb{P}_{\pi_\theta}^{(\lambda)}(s_2 = s|s_0)R_{\pi_\theta}^{(\lambda)}(s) \\
&\qquad + \tilde{\gamma}^3\sum_{s \in \mathcal{S}}\mathbb{P}_{\pi_\theta}^{(\lambda)}(s_3 = s|s_0)V_{\pi_\theta}(s) \\
=&\cdots \\
=&\sum_{s \in \mathcal{S}}\sum_{t=0}^{\infty}\tilde{\gamma}^t\mathbb{P}_{\pi_\theta}^{(\lambda)}(s_t = s|s_0)R_{\pi_\theta}^{(\lambda)}(s) \overset{(69)}{=} \frac{1}{1 - \tilde{\gamma}}\sum_{s \in \mathcal{S}}d_{\pi_\theta}^{s_0,\lambda}(s)R_{\pi_\theta}^{(\lambda)}(s).
\end{aligned} \tag{75}$$

According to (57) and (75), we have

$$J(\pi_{\boldsymbol{\theta}}) = \sum_{s_0 \in \mathcal{S}} \rho_0(s_0) V_{\pi_{\boldsymbol{\theta}}}(s_0) \overset{(75)}{=} \frac{1}{1-\tilde{\gamma}} \sum_{s_0 \in \mathcal{S}} \rho_0(s_0) \sum_{s \in \mathcal{S}} d_{\pi_{\boldsymbol{\theta}}}^{s_0, \lambda}(s) R_{\pi_{\boldsymbol{\theta}}}^{(\lambda)}(s)$$

$$= \frac{1}{1-\tilde{\gamma}} \sum_{s \in \mathcal{S}} \underbrace{\left( \sum_{s_0 \in \mathcal{S}} \rho_0(s_0) d_{\pi_{\boldsymbol{\theta}}}^{s_0, \lambda}(s) \right)}_{=d_{\pi_{\boldsymbol{\theta}}}^{\lambda}(s)} R_{\pi_{\boldsymbol{\theta}}}^{(\lambda)}(s)$$

$$= \frac{1}{1-\tilde{\gamma}} \sum_{s \in \mathcal{S}} d_{\pi_{\boldsymbol{\theta}}}^{\lambda}(s) R_{\pi_{\boldsymbol{\theta}}}^{(\lambda)}(s) = \frac{1}{1-\tilde{\gamma}} \mathbb{E}_{s \sim d_{\pi_{\boldsymbol{\theta}}}^{\lambda}(\cdot)} \left[ R_{\pi_{\boldsymbol{\theta}}}^{(\lambda)}(s) \right]. \tag{76}$$

Finally, we summarize above results in the following Lemma 1.

**Lemma 1.** *The objective* $J(\pi_{\boldsymbol{\theta}})$ *(58) can be rewritten as the following version:*

$$J(\pi_{\boldsymbol{\theta}}) = \frac{1}{1-\tilde{\gamma}} \sum_{s \in \mathcal{S}} d_{\pi_{\boldsymbol{\theta}}}^{\lambda}(s) R_{\pi_{\boldsymbol{\theta}}}^{(\lambda)}(s) = \frac{1}{1-\tilde{\gamma}} \mathbb{E}_{s \sim d_{\pi_{\boldsymbol{\theta}}}^{\lambda}(\cdot)} \left[ R_{\pi_{\boldsymbol{\theta}}}^{(\lambda)}(s) \right].$$

# E  Proof of Theorem 1

We need the following Proposition 4 to prove Theorem 1, which illustrates an identity for the objective function of policy optimization.

**Proposition 4.** *For any function $\varphi(\cdot) : \mathcal{S} \to \mathbb{R}$, for any policy $\pi_{\boldsymbol{\theta}}$, for any trajectory satisfies $\tau = \{s_t, a_t, r_{t+1}\}_{t \geq 0} \sim \pi_{\boldsymbol{\theta}}$, let*

$$\delta_t^{\varphi} = r(s_{t+1}|s_t, a_t) + \gamma \varphi(s_{t+1}) - \varphi(s_t),$$

$$\delta_{\pi_{\boldsymbol{\theta}},t}^{\varphi}(s) = \mathbb{E}_{s_t \sim \mathbb{P}_{\pi_{\boldsymbol{\theta}}}(\cdot|s), a_t \sim \pi_{\boldsymbol{\theta}}(\cdot|s_t), s_{t+1} \sim \mathbb{P}(\cdot|s_t, a_t)} \left[ \delta_t^{\varphi} \right],$$

*then, the objective $J(\pi_{\boldsymbol{\theta}})$ (76) can be rewritten as the following version:*

$$
\begin{aligned}
J(\pi_{\boldsymbol{\theta}}) =& \mathbb{E}_{s_0 \sim \rho_0(\cdot)}[\varphi(s_0)] + \frac{1}{1-\tilde{\gamma}} \sum_{s \in \mathcal{S}} d_{\pi_{\boldsymbol{\theta}}}^{\lambda}(s) \left( \sum_{t=0}^{\infty} \gamma^t \lambda^t \delta_{\pi_{\boldsymbol{\theta}},t}^{\varphi}(s) \right) \\
=& \mathbb{E}_{s_0 \sim \rho_0(\cdot)}[\varphi(s_0)] + \frac{1}{1-\tilde{\gamma}} \mathbb{E}_{s \sim d_{\pi_{\boldsymbol{\theta}}}^{\lambda}(\cdot)} \left[ \sum_{t=0}^{\infty} \gamma^t \lambda^t \delta_{\pi_{\boldsymbol{\theta}},t}^{\varphi}(s) \right].
\end{aligned}
\tag{77}
$$

We present the proof of of Proposition 4 at the end of this section, see Section E.2.

We introduce a vector $\boldsymbol{\delta}_{\pi_{\boldsymbol{\theta}},t}^{\varphi} \in \mathbb{R}^{|\mathcal{S}|}$ and its components are: for any $s \in \mathcal{S}$

$$\boldsymbol{\delta}_{\pi_{\boldsymbol{\theta}},t}^{\varphi}[s] = \delta_{\pi_{\boldsymbol{\theta}},t}^{\varphi}(s). \tag{78}$$

Then, we rewrite the objective as the following vector version

$$J(\pi_{\boldsymbol{\theta}}) = \mathbb{E}_{s_0 \sim \rho_0(\cdot)}[\varphi(s_0)] + \frac{1}{1-\tilde{\gamma}} \sum_{t=0}^{\infty} \gamma^t \lambda^t \langle \mathbf{d}_{\pi_{\boldsymbol{\theta}}}^{\lambda}, \boldsymbol{\delta}_{\pi_{\boldsymbol{\theta}},t}^{\varphi} \rangle, \tag{79}$$

where $\langle \cdot, \cdot \rangle$ denotes inner production between two vectors.

## E.1  Proof of Theorem 1

**Theorem 1** (Generalized Policy Performance Difference) *For any function $\varphi(\cdot) : \mathcal{S} \to \mathbb{R}$, for two arbitrary policy $\pi_{\boldsymbol{\theta}}$ and $\pi_{\boldsymbol{\theta}'}$, for any $p, q \in [1, \infty)$ such that $\frac{1}{p} + \frac{1}{q} = 1$, The following bound holds:*

$$\frac{1}{1-\tilde{\gamma}} \sum_{t=0}^{\infty} \gamma^t \lambda^t M_{p,q,t}^{\varphi,-}(\pi_{\boldsymbol{\theta}}, \pi_{\boldsymbol{\theta}'}) \leq J(\pi_{\boldsymbol{\theta}}) - J(\pi_{\boldsymbol{\theta}'}) \leq \frac{1}{1-\tilde{\gamma}} \sum_{t=0}^{\infty} \gamma^t \lambda^t M_{p,q,t}^{\varphi,+}(\pi_{\boldsymbol{\theta}}, \pi_{\boldsymbol{\theta}'}), \tag{80}$$

*where the terms $M_{p,q,t}^{\varphi,-}$ and $M_{p,q,t}^{\varphi,+}$ are defined in (96)-(97).*

*Proof.* (of Theorem 1)

We consider two arbitrary policies $\pi_{\boldsymbol{\theta}}$ and $\pi_{\boldsymbol{\theta}'}$ with different parameters $\boldsymbol{\theta}$ and $\boldsymbol{\theta}'$, let

$$D_t^{\varphi,(\lambda)}(\pi_{\boldsymbol{\theta}}, \pi_{\boldsymbol{\theta}'}) =: \langle \mathbf{d}_{\pi_{\boldsymbol{\theta}}}^{\lambda}, \boldsymbol{\delta}_{\pi_{\boldsymbol{\theta}},t}^{\varphi} \rangle - \langle \mathbf{d}_{\pi_{\boldsymbol{\theta}'}}^{\lambda}, \boldsymbol{\delta}_{\pi_{\boldsymbol{\theta}'},t}^{\varphi} \rangle. \tag{81}$$

According to (79), we obtain performance difference as follows,

$$
\begin{aligned}
J(\pi_{\boldsymbol{\theta}}) - J(\pi_{\boldsymbol{\theta}'}) =& \frac{1}{1-\tilde{\gamma}} \sum_{t=0}^{\infty} \gamma^t \lambda^t \left( \langle \mathbf{d}_{\pi_{\boldsymbol{\theta}}}^{\lambda}, \boldsymbol{\delta}_{\pi_{\boldsymbol{\theta}},t}^{\varphi} \rangle - \langle \mathbf{d}_{\pi_{\boldsymbol{\theta}'}}^{\lambda}, \boldsymbol{\delta}_{\pi_{\boldsymbol{\theta}'},t}^{\varphi} \rangle \right) \\
=& \frac{1}{1-\tilde{\gamma}} \sum_{t=0}^{\infty} \gamma^t \lambda^t D_t^{\varphi,(\lambda)}(\pi_{\boldsymbol{\theta}}, \pi_{\boldsymbol{\theta}'}),
\end{aligned}
\tag{82}
$$

which requires us to consider the boundedness of the difference $D_t^{\varphi,(\lambda)}(\pi_{\boldsymbol{\theta}}, \pi_{\boldsymbol{\theta}'})$ (81) .

**Step 1: Bound the term $D_t^{\varphi,(\lambda)}(\pi_{\boldsymbol{\theta}}, \pi_{\boldsymbol{\theta}'})$ (81).**

We rewrite the first term of (81) as follows,

$$\langle \mathbf{d}_{\pi_{\boldsymbol{\theta}}}^{\lambda}, \boldsymbol{\delta}_{\pi_{\boldsymbol{\theta}},t}^{\varphi} \rangle = \langle \mathbf{d}_{\pi_{\boldsymbol{\theta}'}}^{\lambda}, \boldsymbol{\delta}_{\pi_{\boldsymbol{\theta}},t}^{\varphi} \rangle + \langle \mathbf{d}_{\pi_{\boldsymbol{\theta}}}^{\lambda} - \mathbf{d}_{\pi_{\boldsymbol{\theta}'}}^{\lambda}, \boldsymbol{\delta}_{\pi_{\boldsymbol{\theta}},t}^{\varphi} \rangle, \tag{83}$$

which is bounded by applying Hölder's inequality to the term $\langle \mathbf{d}_{\pi_{\boldsymbol{\theta}}}^{\lambda} - \mathbf{d}_{\pi_{\boldsymbol{\theta}'}}^{\lambda}, \boldsymbol{\delta}_{\pi_{\boldsymbol{\theta}},t}^{\varphi} \rangle$, we rewrite (83) as follows,

$$\begin{aligned}
&\langle \mathbf{d}_{\pi_{\boldsymbol{\theta}'}}^{\lambda}, \boldsymbol{\delta}_{\pi_{\boldsymbol{\theta}},t}^{\varphi} \rangle - \|\mathbf{d}_{\pi_{\boldsymbol{\theta}}}^{\lambda} - \mathbf{d}_{\pi_{\boldsymbol{\theta}'}}^{\lambda}\|_p \|\boldsymbol{\delta}_{\pi_{\boldsymbol{\theta}},t}^{\varphi}\|_q \\
\leq & \langle \mathbf{d}_{\pi_{\boldsymbol{\theta}}}^{\lambda}, \boldsymbol{\delta}_{\pi_{\boldsymbol{\theta}},t}^{\varphi} \rangle \leq \langle \mathbf{d}_{\pi_{\boldsymbol{\theta}'}}^{\lambda}, \boldsymbol{\delta}_{\pi_{\boldsymbol{\theta}},t}^{\varphi} \rangle + \|\mathbf{d}_{\pi_{\boldsymbol{\theta}}}^{\lambda} - \mathbf{d}_{\pi_{\boldsymbol{\theta}'}}^{\lambda}\|_p \|\boldsymbol{\delta}_{\pi_{\boldsymbol{\theta}},t}^{\varphi}\|_q,
\end{aligned} \tag{84}$$

where $p, q \in [1, \infty)$ and $\frac{1}{p} + \frac{1}{q} = 1$. Let

$$\epsilon_{p,q,t}^{\varphi,(\lambda)}(\pi_{\boldsymbol{\theta}}, \pi_{\boldsymbol{\theta}'}) =: \|\mathbf{d}_{\pi_{\boldsymbol{\theta}}}^{\lambda} - \mathbf{d}_{\pi_{\boldsymbol{\theta}'}}^{\lambda}\|_p \|\boldsymbol{\delta}_{\pi_{\boldsymbol{\theta}},t}^{\varphi}\|_q,$$

then we rewrite Eq.(84) as follows,

$$\langle \mathbf{d}_{\pi_{\boldsymbol{\theta}'}}^{\lambda}, \boldsymbol{\delta}_{\pi_{\boldsymbol{\theta}},t}^{\varphi} \rangle - \epsilon_{p,q,t}^{\varphi,(\lambda)}(\pi_{\boldsymbol{\theta}}, \pi_{\boldsymbol{\theta}'}) \leq \langle \mathbf{d}_{\pi_{\boldsymbol{\theta}}}^{\lambda}, \boldsymbol{\delta}_{\pi_{\boldsymbol{\theta}},t}^{\varphi} \rangle \leq \langle \mathbf{d}_{\pi_{\boldsymbol{\theta}'}}^{\lambda}, \boldsymbol{\delta}_{\pi_{\boldsymbol{\theta}},t}^{\varphi} \rangle + \epsilon_{p,q,t}^{\varphi,(\lambda)}(\pi_{\boldsymbol{\theta}}, \pi_{\boldsymbol{\theta}'}). \tag{85}$$

Let

$$M_t^{\varphi}(\pi_{\boldsymbol{\theta}}, \pi_{\boldsymbol{\theta}'}) =: \underbrace{\langle \mathbf{d}_{\pi_{\boldsymbol{\theta}'}}^{\lambda}, \boldsymbol{\delta}_{\pi_{\boldsymbol{\theta}},t}^{\varphi} \rangle}_{\text{Term-I}} - \underbrace{\langle \mathbf{d}_{\pi_{\boldsymbol{\theta}'}}^{\lambda}, \boldsymbol{\delta}_{\pi_{\boldsymbol{\theta}'},t}^{\varphi} \rangle}_{\text{Term-II}}, \tag{86}$$

combining the (81) and (85), we achieve the boundedness of $D_t^{\varphi}(\pi_{\boldsymbol{\theta}}, \pi_{\boldsymbol{\theta}'})$ as follows

$$M_t^{\varphi}(\pi_{\boldsymbol{\theta}}, \pi_{\boldsymbol{\theta}'}) - \epsilon_{p,q,t}^{\varphi,(\lambda)}(\pi_{\boldsymbol{\theta}}, \pi_{\boldsymbol{\theta}'}) \leq D_t^{\varphi}(\pi_{\boldsymbol{\theta}}, \pi_{\boldsymbol{\theta}'}) \leq M_t^{\varphi}(\pi_{\boldsymbol{\theta}}, \pi_{\boldsymbol{\theta}'}) + \epsilon_{p,q,t}^{\varphi,(\lambda)}(\pi_{\boldsymbol{\theta}}, \pi_{\boldsymbol{\theta}'}). \tag{87}$$

## Step 2: Analyze the term $M_t^{\varphi}(\pi_{\boldsymbol{\theta}}, \pi_{\boldsymbol{\theta}'})$ (86).

To analyze (87) further, we need to consider the first term appears in $M_t^{\varphi}(\pi_{\boldsymbol{\theta}}, \pi_{\boldsymbol{\theta}'})$ (86):

$$\begin{aligned}
\text{Term-I (86)} &= \langle \mathbf{d}_{\pi_{\boldsymbol{\theta}'}}^{\lambda}, \boldsymbol{\delta}_{\pi_{\boldsymbol{\theta}},t}^{\varphi} \rangle \\
&= \sum_{s \in \mathcal{S}} d_{\pi_{\boldsymbol{\theta}'}}^{\lambda}(s) \delta_{\pi_{\boldsymbol{\theta}},t}^{\varphi}(s) = \mathbb{E}_{s \sim d_{\pi_{\boldsymbol{\theta}'}}^{\lambda}(\cdot)} \left[ \delta_{\pi_{\boldsymbol{\theta}},t}^{\varphi}(s) \right]
\end{aligned} \tag{88}$$

$$\overset{(78)}{=} \mathbb{E}_{s \sim d_{\pi_{\boldsymbol{\theta}'}}^{\lambda}(\cdot)} \left[ \mathbb{E}_{s_t \sim \mathbb{P}_{\pi_{\boldsymbol{\theta}}}(\cdot|s)} [\delta_{\pi_{\boldsymbol{\theta}}}^{\varphi}(s_t)] \right]. \tag{89}$$

We notice the following relationship

$$\delta_{\pi_{\boldsymbol{\theta}},t}^{\varphi}(s) = \underset{\substack{s_t \sim \mathbb{P}_{\pi_{\boldsymbol{\theta}}}(\cdot|s) \\ a_t \sim \pi_{\boldsymbol{\theta}}(\cdot|s_t) \\ s_{t+1} \sim \mathbb{P}(\cdot|s_t, a_t)}}{\mathbb{E}} [\delta_t^{\varphi}] = \underset{\substack{s_t \sim \mathbb{P}_{\pi_{\boldsymbol{\theta}'}}(\cdot|s) \\ a_t \sim \pi_{\boldsymbol{\theta}'}(\cdot|s_t) \\ s_{t+1} \sim \mathbb{P}(\cdot|s_t, a_t)}}{\mathbb{E}} \left[ \frac{\pi_{\boldsymbol{\theta}}(a_t|s_t)}{\pi_{\boldsymbol{\theta}'}(a_t|s_t)} \delta_t^{\varphi} \right], \tag{90}$$

which holds since we use importance sampling: for any distribution $p(\cdot)$ and $q(\cdot)$, for any random variable function $f(\cdot)$,

$$\mathbb{E}_{x \sim p(x)}[f(x)] = \mathbb{E}_{x \sim q(x)} \left[ \frac{p(x)}{q(x)} f(x) \right].$$

According to (88), (90), we rewrite the term $\langle \mathbf{d}_{\pi_{\boldsymbol{\theta}'}}^{\lambda}, \boldsymbol{\delta}_{\pi_{\boldsymbol{\theta}},t}^{\varphi} \rangle$ in Eq.(86) as follows,

$$\text{Term-I (86)} = \langle \mathbf{d}_{\pi_{\boldsymbol{\theta}'}}^{\lambda}, \boldsymbol{\delta}_{\pi_{\boldsymbol{\theta}},t}^{\varphi} \rangle = \sum_{s \in \mathcal{S}} d_{\pi_{\boldsymbol{\theta}'}}^{\lambda}(s) \left( \underset{\substack{s_t \sim \mathbb{P}_{\pi_{\boldsymbol{\theta}'}}(\cdot|s) \\ a_t \sim \pi_{\boldsymbol{\theta}'}(\cdot|s_t) \\ s_{t+1} \sim \mathbb{P}(\cdot|s_t, a_t)}}{\mathbb{E}} \left[ \frac{\pi_{\boldsymbol{\theta}}(a_t|s_t)}{\pi_{\boldsymbol{\theta}'}(a_t|s_t)} \delta_t^{\varphi} \right] \right). \tag{91}$$

Now, we consider the second term appears in $M_t^\varphi(\pi_{\boldsymbol{\theta}}, \pi_{\boldsymbol{\theta}'})$ (86):

$$\text{Term-II (86)} = \langle \mathbf{d}_{\pi_{\boldsymbol{\theta}'}}^\lambda, \boldsymbol{\delta}_{\pi_{\boldsymbol{\theta}'},t}^\varphi \rangle$$

$$= \sum_{s \in \mathcal{S}} d_{\pi_{\boldsymbol{\theta}'}}^\lambda(s) \delta_{\pi_{\boldsymbol{\theta}'},t}^\varphi(s) = \sum_{s \in \mathcal{S}} d_{\pi_{\boldsymbol{\theta}'}}^\lambda(s) \left( \mathop{\mathbb{E}}_{\substack{s_t \sim \mathbb{P}_{\pi_{\boldsymbol{\theta}'}}(\cdot|s) \\ a_t \sim \pi_{\boldsymbol{\theta}'}(\cdot|s_t) \\ s_{t+1} \sim \mathbb{P}(\cdot|s_t,a_t)}} [\delta_t^\varphi] \right). \tag{92}$$

Finally, take the results (91) and (92) to (86), we obtain the difference between $\langle \mathbf{d}_{\pi_{\boldsymbol{\theta}'}}^\lambda, \boldsymbol{\delta}_{\pi_{\boldsymbol{\theta}},t}^\varphi \rangle$ and $\langle \mathbf{d}_{\pi_{\boldsymbol{\theta}'}}^\lambda, \boldsymbol{\delta}_{\pi_{\boldsymbol{\theta}'},t}^\varphi \rangle$, i.e., we achieve a identity for $M_t^\varphi(\pi_{\boldsymbol{\theta}}, \pi_{\boldsymbol{\theta}'})$ (86) as follows,

$$M_t^\varphi(\pi_{\boldsymbol{\theta}}, \pi_{\boldsymbol{\theta}'}) \overset{(86)}{=} \langle \mathbf{d}_{\pi_{\boldsymbol{\theta}'}}^\lambda, \boldsymbol{\delta}_{\pi_{\boldsymbol{\theta}},t}^\varphi \rangle - \langle \mathbf{d}_{\pi_{\boldsymbol{\theta}'}}^\lambda, \boldsymbol{\delta}_{\pi_{\boldsymbol{\theta}'},t}^\varphi \rangle$$

$$\overset{(91),(92)}{=} \sum_{s \in \mathcal{S}} d_{\pi_{\boldsymbol{\theta}'}}^\lambda(s) \left( \mathop{\mathbb{E}}_{\substack{s_t \sim \mathbb{P}_{\pi_{\boldsymbol{\theta}'}}(\cdot|s) \\ a_t \sim \pi_{\boldsymbol{\theta}'}(\cdot|s_t) \\ s_{t+1} \sim \mathbb{P}(\cdot|s_t,a_t)}} \left[ \left( \frac{\pi_{\boldsymbol{\theta}}(a_t|s_t)}{\pi_{\boldsymbol{\theta}'}(a_t|s_t)} - 1 \right) \delta_t^\varphi \right] \right). \tag{93}$$

To simplify expression, we introduce a notation as follows,

$$\Delta_t^\varphi(\pi_{\boldsymbol{\theta}}, \pi_{\boldsymbol{\theta}'}, s) =: \mathop{\mathbb{E}}_{\substack{s_t \sim \mathbb{P}_{\pi_{\boldsymbol{\theta}'}}(\cdot|s) \\ a_t \sim \pi_{\boldsymbol{\theta}'}(\cdot|s_t) \\ s_{t+1} \sim \mathbb{P}(\cdot|s_t,a_t)}} \left[ \left( \frac{\pi_{\boldsymbol{\theta}}(a_t|s_t)}{\pi_{\boldsymbol{\theta}'}(a_t|s_t)} - 1 \right) \delta_t^\varphi \right], \tag{94}$$

and we use a vector $\boldsymbol{\Delta}_t^\varphi(\pi_{\boldsymbol{\theta}}, \pi_{\boldsymbol{\theta}'}) \in \mathbb{R}^{|\mathcal{S}|}$ to store all the values $\{\Delta_t^\varphi(\pi_{\boldsymbol{\theta}}, \pi_{\boldsymbol{\theta}'}, s)\}_{s \in \mathcal{S}}$:
$$\boldsymbol{\Delta}_t^\varphi(\pi_{\boldsymbol{\theta}}, \pi_{\boldsymbol{\theta}'})[s] = \Delta_t^\varphi(\pi_{\boldsymbol{\theta}}, \pi_{\boldsymbol{\theta}'}, s).$$

Then we rewrite $\langle \mathbf{d}_{\pi_{\boldsymbol{\theta}'}}^\lambda, \boldsymbol{\delta}_{\pi_{\boldsymbol{\theta}},t}^\varphi \rangle - \langle \mathbf{d}_{\pi_{\boldsymbol{\theta}'}}^\lambda, \boldsymbol{\delta}_{\pi_{\boldsymbol{\theta}'},t}^\varphi \rangle$ (93) as follows,

$$M_t^\varphi(\pi_{\boldsymbol{\theta}}, \pi_{\boldsymbol{\theta}'}) = \langle \mathbf{d}_{\pi_{\boldsymbol{\theta}'}}^\lambda, \boldsymbol{\delta}_{\pi_{\boldsymbol{\theta}},t}^\varphi \rangle - \langle \mathbf{d}_{\pi_{\boldsymbol{\theta}'}}^\lambda, \boldsymbol{\delta}_{\pi_{\boldsymbol{\theta}'},t}^\varphi \rangle$$

$$\overset{(93)}{=} \sum_{s \in \mathcal{S}} d_{\pi_{\boldsymbol{\theta}'}}^\lambda(s) \Delta_t^\varphi(\pi_{\boldsymbol{\theta}}, \pi_{\boldsymbol{\theta}'}, s) = \langle \mathbf{d}_{\pi_{\boldsymbol{\theta}'}}^\lambda, \boldsymbol{\Delta}_t^\varphi(\pi_{\boldsymbol{\theta}}, \pi_{\boldsymbol{\theta}'}) \rangle.$$

## Step 3: Bound on $J(\pi_{\boldsymbol{\theta}}) - J(\pi_{\boldsymbol{\theta}'})$.

Recall (87), taking above result in it, we obtain

$$\langle \mathbf{d}_{\pi_{\boldsymbol{\theta}'}}^\lambda, \boldsymbol{\Delta}_t^\varphi(\pi_{\boldsymbol{\theta}}, \pi_{\boldsymbol{\theta}'}) \rangle - \epsilon_{p,q,t}^{\varphi,(\lambda)}(\pi_{\boldsymbol{\theta}}, \pi_{\boldsymbol{\theta}'}) \leq D_t^\varphi(\pi_{\boldsymbol{\theta}}, \pi_{\boldsymbol{\theta}'}) \leq \langle \mathbf{d}_{\pi_{\boldsymbol{\theta}'}}^\lambda, \boldsymbol{\Delta}_t^\varphi(\pi_{\boldsymbol{\theta}}, \pi_{\boldsymbol{\theta}'}) \rangle + \epsilon_{p,q,t}^{\varphi,(\lambda)}(\pi_{\boldsymbol{\theta}}, \pi_{\boldsymbol{\theta}'}). \tag{95}$$

Finally, let

$$M_{p,q,t}^{\varphi,-}(\pi_{\boldsymbol{\theta}}, \pi_{\boldsymbol{\theta}'}) = \langle \mathbf{d}_{\pi_{\boldsymbol{\theta}'}}^\lambda, \boldsymbol{\Delta}_t^\varphi(\pi_{\boldsymbol{\theta}}, \pi_{\boldsymbol{\theta}'}) \rangle - \epsilon_{p,q,t}^{\varphi,(\lambda)}(\pi_{\boldsymbol{\theta}}, \pi_{\boldsymbol{\theta}'}) \tag{96}$$

$$= \sum_{s \in \mathcal{S}} d_{\pi_{\boldsymbol{\theta}'}}^\lambda(s) \left( \mathop{\mathbb{E}}_{\substack{s_t \sim \mathbb{P}_{\pi_{\boldsymbol{\theta}'}}(\cdot|s) \\ a_t \sim \pi_{\boldsymbol{\theta}'}(\cdot|s_t) \\ s_{t+1} \sim \mathbb{P}(\cdot|s_t,a_t)}} \left[ \left( \frac{\pi_{\boldsymbol{\theta}}(a_t|s_t)}{\pi_{\boldsymbol{\theta}'}(a_t|s_t)} - 1 \right) \delta_t^\varphi \right] \right) - \|\mathbf{d}_{\pi_{\boldsymbol{\theta}}}^\lambda - \mathbf{d}_{\pi_{\boldsymbol{\theta}'}}^\lambda\|_p \|\boldsymbol{\delta}_{\pi_{\boldsymbol{\theta}},t}^\varphi\|_q$$

$$= \mathbb{E}_{s \sim d_{\pi_{\boldsymbol{\theta}'}}^\lambda(\cdot)} \left[ \mathop{\mathbb{E}}_{\substack{s_t \sim \mathbb{P}_{\pi_{\boldsymbol{\theta}'}}(\cdot|s) \\ a_t \sim \pi_{\boldsymbol{\theta}'}(\cdot|s_t) \\ s_{t+1} \sim \mathbb{P}(\cdot|s_t,a_t)}} \left[ \left( \frac{\pi_{\boldsymbol{\theta}}(a_t|s_t)}{\pi_{\boldsymbol{\theta}'}(a_t|s_t)} - 1 \right) \delta_t^\varphi \right] \right] - \|\mathbf{d}_{\pi_{\boldsymbol{\theta}}}^\lambda - \mathbf{d}_{\pi_{\boldsymbol{\theta}'}}^\lambda\|_p \|\boldsymbol{\delta}_{\pi_{\boldsymbol{\theta}},t}^\varphi\|_q.$$

and

$$M_{p,q,t}^{\varphi,+}(\pi_{\boldsymbol{\theta}}, \pi_{\boldsymbol{\theta}'}) = \langle \mathbf{d}_{\pi_{\boldsymbol{\theta}'}}^{\lambda}, \boldsymbol{\Delta}_t^{\varphi}(\pi_{\boldsymbol{\theta}}, \pi_{\boldsymbol{\theta}'}) \rangle + \epsilon_{p,q,t}^{\varphi,(\lambda)}(\pi_{\boldsymbol{\theta}}, \pi_{\boldsymbol{\theta}'}) \tag{97}$$

$$= \sum_{s \in \mathcal{S}} d_{\pi_{\boldsymbol{\theta}'}}^{\lambda}(s) \left( \underset{\substack{s_t \sim \mathbb{P}_{\pi_{\boldsymbol{\theta}'}}(\cdot|s) \\ a_t \sim \pi_{\boldsymbol{\theta}'}(\cdot|s_t) \\ s_{t+1} \sim \mathbb{P}(\cdot|s_t, a_t)}}{\mathbb{E}} \left[ \left( \frac{\pi_{\boldsymbol{\theta}}(a_t|s_t)}{\pi_{\boldsymbol{\theta}'}(a_t|s_t)} - 1 \right) \delta_t^{\varphi} \right] \right) + \| \mathbf{d}_{\pi_{\boldsymbol{\theta}}}^{\lambda} - \mathbf{d}_{\pi_{\boldsymbol{\theta}'}}^{\lambda} \|_p \| \boldsymbol{\delta}_{\pi_{\boldsymbol{\theta}},t}^{\varphi} \|_q$$

$$= \mathbb{E}_{s \sim d_{\pi_{\boldsymbol{\theta}'}}^{\lambda}(\cdot)} \left[ \underset{\substack{s_t \sim \mathbb{P}_{\pi_{\boldsymbol{\theta}'}}(\cdot|s) \\ a_t \sim \pi_{\boldsymbol{\theta}'}(\cdot|s_t) \\ s_{t+1} \sim \mathbb{P}(\cdot|s_t, a_t)}}{\mathbb{E}} \left[ \left( \frac{\pi_{\boldsymbol{\theta}}(a_t|s_t)}{\pi_{\boldsymbol{\theta}'}(a_t|s_t)} - 1 \right) \delta_t^{\varphi} \right] \right] + \| \mathbf{d}_{\pi_{\boldsymbol{\theta}}}^{\lambda} - \mathbf{d}_{\pi_{\boldsymbol{\theta}'}}^{\lambda} \|_p \| \boldsymbol{\delta}_{\pi_{\boldsymbol{\theta}},t}^{\varphi} \|_q.$$

According to (82) and (95), we achieve the boundedness of performance difference between two arbitrary policies $\pi_{\boldsymbol{\theta}}$ and $\pi_{\boldsymbol{\theta}'}$:

$$\underbrace{\frac{1}{1-\tilde{\gamma}} \sum_{t=0}^{\infty} \gamma^t \lambda^t M_{p,q,t}^{\varphi,-}(\pi_{\boldsymbol{\theta}}, \pi_{\boldsymbol{\theta}'})}_{=:L_{p,q,}^{\varphi,-}} \le J(\pi_{\boldsymbol{\theta}}) - J(\pi_{\boldsymbol{\theta}'}) \le \underbrace{\frac{1}{1-\tilde{\gamma}} \sum_{t=0}^{\infty} \gamma^t \lambda^t M_{p,q,t}^{\varphi,+}(\pi_{\boldsymbol{\theta}}, \pi_{\boldsymbol{\theta}'})}_{=:L_{p,q,}^{\varphi,+}}. \tag{98}$$

$\square$

### E.2 Proof of Proposition 4

*Proof.* (of Proposition 4).

**Step 1: Rewrite the objective $J(\pi_{\boldsymbol{\theta}})$ in Eq.(76).**

We rewrite the discounted distribution $\mathbf{d}_{\pi_{\boldsymbol{\theta}}}^{\lambda}$ (72) as follows,

$$\boldsymbol{\rho}_0 - \frac{1}{1-\tilde{\gamma}} \mathbf{d}_{\pi_{\boldsymbol{\theta}}}^{\lambda} + \frac{\tilde{\gamma}}{1-\tilde{\gamma}} \mathbf{P}_{\pi_{\boldsymbol{\theta}}}^{(\lambda)} \mathbf{d}_{\pi_{\boldsymbol{\theta}}}^{\lambda} = \mathbf{0}. \tag{99}$$

Let $\varphi(\cdot)$ be a real number function defined on the state space $\mathcal{S}$, i.e., $\varphi : \mathcal{S} \to \mathbb{R}$. Then we define a vector function $\boldsymbol{\phi}(\cdot) \in \mathbb{R}^{|\mathcal{S}|}$ to collect all the values $\{\varphi(s)\}_{s \in \mathcal{S}}$, and its components are

$$\boldsymbol{\phi}[s] = \varphi(s), \quad s \in \mathcal{S}.$$

Now, we take the inner product between the vector $\boldsymbol{\phi}$ and (99), we have

$$0 = \langle \boldsymbol{\rho}_0 - \frac{1}{1-\tilde{\gamma}} \mathbf{d}_{\pi_{\boldsymbol{\theta}}}^{\lambda} + \frac{\tilde{\gamma}}{1-\tilde{\gamma}} \mathbf{P}_{\pi_{\boldsymbol{\theta}}}^{(\lambda)} \mathbf{d}_{\pi_{\boldsymbol{\theta}}}^{\lambda}, \boldsymbol{\phi} \rangle$$

$$= \langle \boldsymbol{\rho}_0, \boldsymbol{\phi} \rangle - \frac{1}{1-\tilde{\gamma}} \langle \mathbf{d}_{\pi_{\boldsymbol{\theta}}}^{\lambda}, \boldsymbol{\phi} \rangle + \frac{\tilde{\gamma}}{1-\tilde{\gamma}} \langle \mathbf{P}_{\pi_{\boldsymbol{\theta}}}^{(\lambda)} \mathbf{d}_{\pi_{\boldsymbol{\theta}}}^{\lambda}, \boldsymbol{\phi} \rangle. \tag{100}$$

We express the first term $\langle \boldsymbol{\rho}_0, \boldsymbol{\phi} \rangle$ of (100) as follows,

$$\langle \boldsymbol{\rho}_0, \boldsymbol{\phi} \rangle = \sum_{s \in \mathcal{S}} \rho_0(s) \varphi(s) = \mathbb{E}_{s \sim \rho_0(\cdot)}[\varphi(s)]. \tag{101}$$

We express the second term $\langle \mathbf{d}_{\pi_{\boldsymbol{\theta}}}^{\lambda}, \boldsymbol{\phi} \rangle$ of (100) as follows,

$$-\frac{1}{1-\tilde{\gamma}} \langle \mathbf{d}_{\pi_{\boldsymbol{\theta}}}^{\lambda}, \boldsymbol{\phi} \rangle = -\frac{1}{1-\tilde{\gamma}} \sum_{s \in \mathcal{S}} d_{\pi_{\boldsymbol{\theta}}}^{\lambda}(s) \varphi(s) = -\frac{1}{1-\tilde{\gamma}} \mathbb{E}_{s \sim d_{\pi_{\boldsymbol{\theta}}}^{\lambda}(\cdot)}[\varphi(s)]. \tag{102}$$

We express the third term $\langle \tilde{\gamma} \mathbf{P}_{\pi_{\boldsymbol{\theta}}}^{(\lambda)} \mathbf{d}_{\pi_{\boldsymbol{\theta}}}^{\lambda}, \boldsymbol{\phi} \rangle$ of (100) as follows,

$$\frac{\tilde{\gamma}}{1-\tilde{\gamma}} \langle \mathbf{P}_{\pi_{\boldsymbol{\theta}}}^{(\lambda)} \mathbf{d}_{\pi_{\boldsymbol{\theta}}}^{\lambda}, \boldsymbol{\phi} \rangle = \frac{\tilde{\gamma}}{1-\tilde{\gamma}} \sum_{s' \in \mathcal{S}} \left( \mathbf{P}_{\pi_{\boldsymbol{\theta}}}^{(\lambda)} \mathbf{d}_{\pi_{\boldsymbol{\theta}}}^{\lambda} \right)[s'] \varphi(s')$$

$$= \frac{\tilde{\gamma}}{1-\tilde{\gamma}} \sum_{s' \in \mathcal{S}} \left( \sum_{s \in \mathcal{S}} \mathbb{P}_{\pi_{\boldsymbol{\theta}}}^{(\lambda)}(s'|s) d_{\pi_{\boldsymbol{\theta}}}^{\lambda}(s) \right) \varphi(s'). \tag{103}$$

According to Lemma 1, put the results (76) and (100) together, we have

$$J(\pi_{\boldsymbol{\theta}}) \overset{(76),(100)}{=} \frac{1}{1-\tilde{\gamma}} \sum_{s\in\mathcal{S}} d_{\pi_{\boldsymbol{\theta}}}^{\lambda}(s) R_{\pi_{\boldsymbol{\theta}}}^{(\lambda)}(s) + \langle \boldsymbol{\rho}_0 - \frac{1}{1-\tilde{\gamma}} \mathbf{d}_{\pi_{\boldsymbol{\theta}}}^{\lambda} + \frac{\tilde{\gamma}}{1-\tilde{\gamma}} \mathbf{P}_{\pi_{\boldsymbol{\theta}}}^{(\lambda)} \mathbf{d}_{\pi_{\boldsymbol{\theta}}}^{\lambda}, \boldsymbol{\phi} \rangle$$

$$= \mathbb{E}_{s_0\sim\rho_0(\cdot)}[\varphi(s_0)] + \frac{1}{1-\tilde{\gamma}} \sum_{s\in\mathcal{S}} d_{\pi_{\boldsymbol{\theta}}}^{\lambda}(s) \left( R_{\pi_{\boldsymbol{\theta}}}^{(\lambda)}(s) + \tilde{\gamma} \sum_{s'\in\mathcal{S}} \mathbb{P}_{\pi_{\boldsymbol{\theta}}}^{(\lambda)}(s'|s)\varphi(s') - \varphi(s) \right),$$

(104)

where the last equation holds since we unfold (100) according to (101)-(103).

**Step 2: Rewrite the term** $\left( R_{\pi_{\boldsymbol{\theta}}}^{(\lambda)}(s) + \tilde{\gamma} \sum_{s'\in\mathcal{S}} \mathbb{P}_{\pi_{\boldsymbol{\theta}}}^{(\lambda)}(s'|s)\varphi(s') - \varphi(s) \right)$ **in Eq.(104).**

Then, we unfold the second term of (104) as follows,

$$R_{\pi_{\boldsymbol{\theta}}}^{(\lambda)}(s) + \tilde{\gamma} \sum_{s'\in\mathcal{S}} \mathbb{P}_{\pi_{\boldsymbol{\theta}}}^{(\lambda)}(s'|s)\varphi(s') - \varphi(s)$$

(105)

$$\overset{(66),(68)}{=} \sum_{t=0}^{\infty} (\gamma\lambda \mathbf{P}_{\pi_{\boldsymbol{\theta}}})^t \mathbf{r}_{\pi_{\boldsymbol{\theta}}}[s] + \tilde{\gamma}(1-\gamma\lambda) \sum_{s'\in\mathcal{S}} \sum_{t=0}^{\infty} (\gamma\lambda)^t \left( \mathbf{P}_{\pi_{\boldsymbol{\theta}}}^{t+1}[s,s'] \right) \varphi(s') - \varphi(s)$$

$$\overset{(63)}{=} \sum_{t=0}^{\infty} (\gamma\lambda \mathbf{P}_{\pi_{\boldsymbol{\theta}}})^t \mathbf{r}_{\pi_{\boldsymbol{\theta}}}[s] + \gamma(1-\lambda) \sum_{s'\in\mathcal{S}} \sum_{t=0}^{\infty} (\gamma\lambda)^t \mathbb{P}_{\pi_{\boldsymbol{\theta}}}(s_{t+1} = s'|s)\varphi(s') - \varphi(s).$$

(106)

Recall the terms $\mathbf{P}_{\pi_{\boldsymbol{\theta}}}^{(\lambda)}$, $\mathbf{r}_{\pi_{\boldsymbol{\theta}}}^{(\lambda)}[s]$ defined in (63)-(68),

$$R_{\pi_{\boldsymbol{\theta}}}^{(\lambda)}(s) + \gamma(1-\lambda) \sum_{s'\in\mathcal{S}} \mathbb{P}_{\pi_{\boldsymbol{\theta}}}^{(\lambda)}(s'|s)\varphi(s') - \varphi(s)$$

(107)

We consider the first term $R_{\pi_{\boldsymbol{\theta}}}^{(\lambda)}(s)$ of (105) as follows,

$$R_{\pi_{\boldsymbol{\theta}}}^{(\lambda)}(s) \overset{(63)-(68)}{=} \mathbf{r}_{\pi_{\boldsymbol{\theta}}}^{(\lambda)}[s] = \sum_{t=0}^{\infty} (\gamma\lambda)^t \mathbf{P}_{\pi_{\boldsymbol{\theta}}}^t \mathbf{r}_{\pi_{\boldsymbol{\theta}}}[s] = \sum_{t=0}^{\infty} \sum_{s_t\in\mathcal{S}} (\gamma\lambda)^t \mathbb{P}_{\pi_{\boldsymbol{\theta}}}(s_t|s) R_{\pi_{\boldsymbol{\theta}}}(s_t).$$

(108)

We consider the second term $\tilde{\gamma} \sum_{s\in\mathcal{S}} \mathbb{P}_{\pi_{\boldsymbol{\theta}}}^{(\lambda)}(s'|s)\varphi(s) - \varphi(s)$ of (105) as follows,

$$\tilde{\gamma} \sum_{s'\in\mathcal{S}} \mathbb{P}_{\pi_{\boldsymbol{\theta}}}^{(\lambda)}(s'|s)\varphi(s') - \varphi(s)$$

$$\overset{(66)}{=} \tilde{\gamma}(1-\gamma\lambda) \sum_{s'\in\mathcal{S}} \sum_{t=0}^{\infty} (\gamma\lambda)^t \mathbb{P}_{\pi_{\boldsymbol{\theta}}}(s_{t+1} = s'|s)\varphi(s') - \varphi(s)$$

(109)

$$\overset{(63)}{=} \gamma(1-\lambda) \sum_{s'\in\mathcal{S}} \sum_{t=0}^{\infty} (\gamma\lambda)^t \mathbb{P}_{\pi_{\boldsymbol{\theta}}}(s_{t+1} = s'|s)\varphi(s') - \varphi(s)$$

(110)

$$= \gamma \sum_{s'\in\mathcal{S}} \sum_{t=0}^{\infty} (\gamma\lambda)^t \mathbb{P}_{\pi_{\boldsymbol{\theta}}}(s_{t+1} = s'|s)\varphi(s') - \sum_{s'\in\mathcal{S}} \underbrace{\left( \sum_{t=0}^{\infty} (\gamma\lambda)^{t+1} \mathbb{P}_{\pi_{\boldsymbol{\theta}}}(s_{t+1} = s'|s)\varphi(s') \right)}_{=\sum_{t=1}^{\infty} (\gamma\lambda)^t \mathbb{P}_{\pi_{\boldsymbol{\theta}}}(s_t = s'|s)\varphi(s')} - \varphi(s)$$

$$= \gamma \sum_{s'\in\mathcal{S}} \sum_{t=0}^{\infty} (\gamma\lambda)^t \mathbb{P}_{\pi_{\boldsymbol{\theta}}}(s_{t+1} = s'|s)\varphi(s') - \underbrace{\left( \sum_{s'\in\mathcal{S}} \sum_{t=1}^{\infty} (\gamma\lambda)^t \mathbb{P}_{\pi_{\boldsymbol{\theta}}}(s_t = s'|s)\varphi(s') + \varphi(s) \right)}_{=\sum_{s'\in\mathcal{S}} \sum_{t=0}^{\infty} (\gamma\lambda)^t \mathbb{P}_{\pi_{\boldsymbol{\theta}}}(s_t = s'|s)\varphi(s')}$$

(111)

$$= \gamma \sum_{s'\in\mathcal{S}} \sum_{t=0}^{\infty} (\gamma\lambda)^t \mathbb{P}_{\pi_{\boldsymbol{\theta}}}(s_{t+1} = s'|s)\varphi(s') - \sum_{s_t\in\mathcal{S}} \sum_{t=0}^{\infty} (\gamma\lambda)^t \mathbb{P}_{\pi_{\boldsymbol{\theta}}}(s_t|s)\varphi(s),$$

(112)

where the equation from Eq.(111) to Eq.(112) holds since: according to (54), we use the following identity

$$\sum_{s' \in \mathcal{S}} \mathbb{P}_{\pi_{\boldsymbol{\theta}}}(s_0 = s'|s)\varphi(s') = \varphi(s).$$

Furthermore, take the result (108) and (112) to (107), we have

$$R_{\pi_{\boldsymbol{\theta}}}^{(\lambda)}(s) + \tilde{\gamma} \sum_{s' \in \mathcal{S}} \mathbb{P}_{\pi_{\boldsymbol{\theta}}}^{(\lambda)}(s'|s)\varphi(s') - \varphi(s)$$

$$= \sum_{t=0}^{\infty} (\gamma\lambda)^t \left( \sum_{s_t \in \mathcal{S}} \mathbb{P}_{\pi_{\boldsymbol{\theta}}}(s_t|s) R_{\pi_{\boldsymbol{\theta}}}(s_t) + \gamma \sum_{s' \in \mathcal{S}} \underbrace{\mathbb{P}_{\pi_{\boldsymbol{\theta}}}(s_{t+1} = s'|s)\varphi(s')}_{\stackrel{(55)}{=} \sum_{s_t \in \mathcal{S}} \mathbb{P}_{\pi_{\boldsymbol{\theta}}}(s_{t+1}=s'|s_t)\mathbb{P}_{\pi_{\boldsymbol{\theta}}}(s_t|s)\varphi(s')} \right.$$

$$\left. - \sum_{s_t \in \mathcal{S}} \mathbb{P}_{\pi_{\boldsymbol{\theta}}}(s_t|s)\varphi(s_t) \right) \qquad (113)$$

$$= \sum_{t=0}^{\infty} (\gamma\lambda)^t \left( \sum_{s_t \in \mathcal{S}} \mathbb{P}_{\pi_{\boldsymbol{\theta}}}(s_t|s) R_{\pi_{\boldsymbol{\theta}}}(s_t) + \gamma \sum_{s_t \in \mathcal{S}} \mathbb{P}_{\pi_{\boldsymbol{\theta}}}(s_t|s) \sum_{s_{t+1} \in \mathcal{S}} \mathbb{P}_{\pi_{\boldsymbol{\theta}}}(s_{t+1}|s_t)\varphi(s_{t+1}) \right.$$

$$\left. - \sum_{s_t \in \mathcal{S}} \mathbb{P}_{\pi_{\boldsymbol{\theta}}}(s_t|s)\varphi(s_t) \right) \qquad (114)$$

$$= \sum_{t=0}^{\infty} (\gamma\lambda)^t \sum_{s_t \in \mathcal{S}} \mathbb{P}_{\pi_{\boldsymbol{\theta}}}(s_t|s) \left( \underbrace{\sum_{a_t \in \mathcal{A}} \pi_{\boldsymbol{\theta}}(a_t|s_t) \sum_{s_{t+1} \in \mathcal{S}} \mathbb{P}(s_{t+1}|s_t, a_t) r(s_{t+1}|s_t, a_t)}_{=R_{\pi_{\boldsymbol{\theta}}}(s_t)} \right.$$

$$\left. + \gamma \underbrace{\sum_{a_t \in \mathcal{A}} \pi_{\boldsymbol{\theta}}(a_t|s_t) \sum_{s_{t+1} \in \mathcal{S}} \mathbb{P}(s_{t+1}|s_t, a_t) \varphi(s_{t+1}) - \varphi(s_t)}_{=\mathbb{P}_{\pi_{\boldsymbol{\theta}}}(s_{t+1}|s_t)} \right)$$

$$= \sum_{t=0}^{\infty} (\gamma\lambda)^t \sum_{s_t \in \mathcal{S}} \mathbb{P}_{\pi_{\boldsymbol{\theta}}}(s_t|s) \sum_{a_t \in \mathcal{A}} \pi_{\boldsymbol{\theta}}(a_t|s_t) \sum_{s_{t+1} \in \mathcal{S}} \mathbb{P}(s_{t+1}|s_t, a_t) \left( r(s_{t+1}|s_t, a_t) + \gamma\varphi(s_{t+1}) - \varphi(s_t) \right)$$

$$\qquad (115)$$

$$= \sum_{t=0}^{\infty} (\gamma\lambda)^t \mathbb{E}_{s_t \sim \mathbb{P}_{\pi_{\boldsymbol{\theta}}}(\cdot|s), a_t \sim \pi_{\boldsymbol{\theta}}(\cdot|s_t), s_{t+1} \sim \mathbb{P}(\cdot|s_t, a_t)} \left[ r(s_{t+1}|s_t, a_t) + \gamma\varphi(s_{t+1}) - \varphi(s_t) \right], \qquad (116)$$

the equation from Eq.(112) to Eq.(113) holds since:

$$\mathbb{P}_{\pi_{\boldsymbol{\theta}}}(s_{t+1}|s) \stackrel{(55)}{=} \sum_{s_t \in \mathcal{S}} \mathbb{P}_{\pi_{\boldsymbol{\theta}}}(s_{t+1}|s_t)\mathbb{P}_{\pi_{\boldsymbol{\theta}}}(s_t|s);$$

the equation from Eq.(113) to Eq.(114) holds since we use the Markov property of the definition of MDP: for each time $t \in \mathbb{N}$,

$$\mathbb{P}_{\pi_{\boldsymbol{\theta}}}(s_{t+1} = s'|s_t = s) = \mathbb{P}_{\pi_{\boldsymbol{\theta}}}(s'|s);$$

the equation (115) the following identity:

$$\sum_{a_t \in \mathcal{A}} \pi_{\boldsymbol{\theta}}(a_t|s_t) = 1, \qquad \sum_{s_{t+1} \in \mathcal{S}} \mathbb{P}(s_{t+1}|s_t, a_t) = 1,$$

then

$$\varphi(s_t) = \sum_{a_t \in \mathcal{A}} \pi_{\boldsymbol{\theta}}(a_t|s_t) \sum_{s_{t+1} \in \mathcal{S}} \mathbb{P}(s_{t+1}|s_t, a)\varphi(s_t).$$

**Step 3: Put all the result together.**

Finally, let

$$\delta_t^{\varphi} = r(s_{t+1}|s_t, a_t) + \gamma\varphi(s_{t+1}) - \varphi(s_t),$$
$$\delta_{\pi_{\boldsymbol{\theta}},t}^{\varphi}(s) = \mathbb{E}_{s_t \sim \mathbb{P}_{\pi_{\boldsymbol{\theta}}}(\cdot|s), a_t \sim \pi_{\boldsymbol{\theta}}(\cdot|s_t), s_{t+1} \sim \mathbb{P}(\cdot|s_t, a_t)} \left[\delta_t^{\varphi}\right],$$

combining the results (104) and (116), we have

$$J(\pi_{\boldsymbol{\theta}}) = \mathbb{E}_{s_0 \sim \rho_0(\cdot)}[\varphi(s_0)] + \frac{1}{1-\tilde{\gamma}} \sum_{s \in \mathcal{S}} d_{\pi_{\boldsymbol{\theta}}}^{\lambda}(s) \left(\sum_{t=0}^{\infty} \gamma^t \lambda^t \delta_{\pi_{\boldsymbol{\theta}},t}^{\varphi}(s)\right) \tag{117}$$
$$= \mathbb{E}_{s_0 \sim \rho_0(\cdot)}[\varphi(s_0)] + \frac{1}{1-\tilde{\gamma}} \mathbb{E}_{s \sim d_{\pi_{\boldsymbol{\theta}}}^{\lambda}(\cdot)} \left[\sum_{t=0}^{\infty} \gamma^t \lambda^t \delta_{\pi_{\boldsymbol{\theta}},t}^{\varphi}(s)\right].$$

This concludes the proof of Proposition 4. $\qquad\qquad\square$

### E.3 Proposition 3

All above bound results appear in (11) and (13) can be extended for a total variational divergence to KL-divergence between policies, which are desirable for policy optimization.

We obtain

$$\mathbb{E}_{s \sim d_{\pi_{\boldsymbol{\theta}'}}^{\lambda}(\cdot)}[D_{\mathrm{TV}}(\pi_{\boldsymbol{\theta}'}, \pi_{\boldsymbol{\theta}})[s]] \le \mathbb{E}_{s \sim d_{\pi_{\boldsymbol{\theta}'}}^{\lambda}(\cdot)} \left[\sqrt{\frac{1}{2}\mathrm{KL}(\pi_{\boldsymbol{\theta}'}, \pi_{\boldsymbol{\theta}})[s]}\right] \le \sqrt{\frac{1}{2}\mathbb{E}_{s \sim d_{\pi_{\boldsymbol{\theta}'}}^{\lambda}(\cdot)}[\mathrm{KL}(\pi_{\boldsymbol{\theta}'}, \pi_{\boldsymbol{\theta}})[s]]},$$
$$\tag{118}$$

where $\mathrm{KL}(\cdot, \cdot)$ is KL-divergence, and

$$\mathrm{KL}(\pi_{\boldsymbol{\theta}'}, \pi_{\boldsymbol{\theta}})[s] = \mathrm{KL}(\pi_{\boldsymbol{\theta}'}(\cdot|s), \pi_{\boldsymbol{\theta}}(\cdot|s));$$

the first inequality follows Pinsker's inequality [Csiszár and Körner, 2011] and the second inequality follows Jensen's inequality. According to (118), we obtain the next Proposition 3.

**Proposition 3**. *All the bounds in (11) and (13) hold if we make the following substitution:*

$$\mathbb{E}_{s \sim d_{\pi_{\boldsymbol{\theta}'}}^{\lambda}(\cdot)}[D_{\mathrm{TV}}(\pi_{\boldsymbol{\theta}'}, \pi_{\boldsymbol{\theta}})[s]] \leftarrow \sqrt{\frac{1}{2}\mathbb{E}_{s \sim d_{\pi_{\boldsymbol{\theta}'}}^{\lambda}(\cdot)}[\mathrm{KL}(\pi_{\boldsymbol{\theta}'}, \pi_{\boldsymbol{\theta}})[s]]}.$$

# F  Lemma 2

In this section, we show Lemma 2 that presents an upper bound to the difference between two $\lambda$-version of normalized discounted distribution. Before we present our main results, we review the norms induced by $p$-norms for matrix.

## F.1  Norms Induced by $p$-norms for Matrix

If the $p$-norm for vectors ($1 \le p \le \infty$) is used for both spaces $\mathbb{R}^n$ and $\mathbb{R}^m$, then the corresponding operator norm is:

$$\|\mathbf{A}\|_p = \sup_{\mathbf{x} \neq \mathbf{0}} \frac{\|\mathbf{A}\mathbf{x}\|_p}{\|\mathbf{x}\|_p}.$$

These induced norms are different from the "entry-wise" $p$-norms and the *Schatten* $p$-norms for matrices treated below, which are also usually denoted by $\|\mathbf{A}\|_p$.

In the special cases of $p = 1$ and $p = \infty$, the induced matrix norms can be computed or estimated by

$$\|\mathbf{A}\|_1 = \max_{1 \le j \le n} \sum_{i=1}^{m} |a_{ij}|,$$

which is simply the maximum absolute column sum of the matrix;

$$\|\mathbf{A}\|_\infty = \max_{1 \le i \le m} \sum_{j=1}^{n} |a_{ij}|,$$

which is simply the maximum absolute row sum of the matrix. Thus, the following equation holds

$$\|\mathbf{A}^\top\|_\infty = \|\mathbf{A}\|_1. \tag{119}$$

## F.2  Lemma 2

**Lemma 2.** *The divergence between discounted future state visitation distributions,* $\|\mathbf{d}^\lambda_{\pi_{\boldsymbol{\theta}'}} - \mathbf{d}^\lambda_{\pi_{\boldsymbol{\theta}}}\|_1$ *is bounded as follows,*

$$\|\mathbf{d}^\lambda_{\pi_{\boldsymbol{\theta}'}} - \mathbf{d}^\lambda_{\pi_{\boldsymbol{\theta}}}\|_1 \le \frac{1}{1 - \tilde{\gamma}} \cdot \frac{\tilde{\gamma}\left(\gamma\lambda(|\mathcal{S}| - 1) + 1\right)}{1 - \gamma\lambda} \mathbb{E}_{s \sim d^\lambda_{\pi_{\boldsymbol{\theta}}}(\cdot)}\left[2D_{\mathrm{TV}}(\pi_{\boldsymbol{\theta}'}, \pi_{\boldsymbol{\theta}})[s]\right],$$

*where $D_{\mathrm{TV}}(\pi_{\boldsymbol{\theta}'}, \pi_{\boldsymbol{\theta}})[s]$ is the total variational divergence between action distributions at state $s$, i.e.,*

$$2D_{\mathrm{TV}}(\pi_{\boldsymbol{\theta}'}, \pi_{\boldsymbol{\theta}})[s] = \sum_{a \in \mathcal{A}} |\pi_{\boldsymbol{\theta}'}(a|s) - \pi_{\boldsymbol{\theta}}(a|s)|.$$

*Proof.* (of Lemma 2). Recall Eq.(72), we know,

$$\mathbf{d}^\lambda_{\pi_{\boldsymbol{\theta}}} = (1 - \tilde{\gamma}) \sum_{t=0}^{\infty} \left(\gamma \mathbf{P}^{(\lambda)}_{\pi_{\boldsymbol{\theta}}}\right)^t \boldsymbol{\rho}_0 = (1 - \tilde{\gamma}) \left(\mathbf{I} - \tilde{\gamma}\mathbf{P}^{(\lambda)}_{\pi_{\boldsymbol{\theta}}}\right)^{-1} \boldsymbol{\rho}_0.$$

To short the expression, we introduce some additional notations as follows.

$$\mathbf{G}_{\pi_{\boldsymbol{\theta}}} = \left(\mathbf{I} - \tilde{\gamma}\mathbf{P}^{(\lambda)}_{\pi_{\boldsymbol{\theta}}}\right)^{-1}, \ \mathbf{G}_{\pi_{\boldsymbol{\theta}'}} = \left(\mathbf{I} - \tilde{\gamma}\mathbf{P}^{(\lambda)}_{\pi_{\boldsymbol{\theta}'}}\right)^{-1}, \ \mathbf{D} = \mathbf{P}^{(\lambda)}_{\pi_{\boldsymbol{\theta}'}} - \mathbf{P}^{(\lambda)}_{\pi_{\boldsymbol{\theta}}}. \tag{120}$$

Then, after some simple algebra, the following holds

$$\mathbf{G}^{-1}_{\pi_{\boldsymbol{\theta}}} - \mathbf{G}^{-1}_{\pi_{\boldsymbol{\theta}'}} = \left(\mathbf{I} - \tilde{\gamma}\mathbf{P}^{(\lambda)}_{\pi_{\boldsymbol{\theta}}}\right) - \left(\mathbf{I} - \tilde{\gamma}\mathbf{P}^{(\lambda)}_{\pi_{\boldsymbol{\theta}'}}\right) = \tilde{\gamma}\mathbf{D}. \tag{121}$$

Furthermore, by left-multiplying by $\mathbf{G}_{\pi_{\boldsymbol{\theta}}}$ and right-multiplying by $\mathbf{G}_{\pi_{\boldsymbol{\theta}'}}$, we achieve

$$\mathbf{G}_{\pi_{\boldsymbol{\theta}'}} - \mathbf{G}_{\pi_{\boldsymbol{\theta}}} = \tilde{\gamma}\mathbf{G}_{\pi_{\boldsymbol{\theta}'}}\mathbf{D}\mathbf{G}_{\pi_{\boldsymbol{\theta}}}. \tag{122}$$

Grouping all the results from (120)-(122), recall (72),

$$\mathbf{d}^\lambda_{\pi_\theta} = (1-\tilde{\gamma})\sum_{t=0}^{\infty}\left(\gamma\mathbf{P}^{(\lambda)}_{\pi_\theta}\right)^t\boldsymbol{\rho}_0 = (1-\tilde{\gamma})\left(\mathbf{I}-\tilde{\gamma}\mathbf{P}^{(\lambda)}_{\pi_\theta}\right)^{-1}\boldsymbol{\rho}_0 = (1-\tilde{\gamma})\mathbf{G}_{\pi_\theta}\boldsymbol{\rho}_0, \tag{123}$$

then we have

$$\begin{aligned}
\mathbf{d}^\lambda_{\pi_{\theta'}} - \mathbf{d}^\lambda_{\pi_\theta} =& (1-\tilde{\gamma})\left(\mathbf{G}_{\pi_{\theta'}} - \mathbf{G}_{\pi_\theta}\right)\boldsymbol{\rho}_0 \\
\overset{(122)}{=}& (1-\tilde{\gamma})\tilde{\gamma}\mathbf{G}_{\pi_{\theta'}}\mathbf{D}\mathbf{G}_{\pi_\theta}\boldsymbol{\rho}_0 \\
\overset{(123)}{=}& \tilde{\gamma}\mathbf{G}_{\pi_{\theta'}}\mathbf{D}\mathbf{d}^\lambda_{\pi_\theta}.
\end{aligned} \tag{124}$$

Applying (124), we have

$$\|\mathbf{d}^\lambda_{\pi_{\theta'}} - \mathbf{d}^\lambda_{\pi_\theta}\|_1 \overset{(124)}{\leq} \tilde{\gamma}\|\mathbf{G}_{\pi_{\theta'}}\|_1\|\mathbf{D}\mathbf{d}^\lambda_{\pi_\theta}\|_1. \tag{125}$$

Firstly, we bound the term $\|\mathbf{G}_{\pi_{\theta'}}\|_1$ as follows,

$$\|\mathbf{G}_{\pi_{\theta'}}\|_1 = \left\|\left(\mathbf{I}-\tilde{\gamma}\mathbf{P}^{(\lambda)}_{\pi_{\theta'}}\right)^{-1}\right\|_1 \leq \sum_{t=0}^{\infty}\tilde{\gamma}^t\left\|\mathbf{P}^{(\lambda)}_{\pi_{\theta'}}\right\|_1 = \frac{1}{1-\tilde{\gamma}}. \tag{126}$$

Thus, recall $\tilde{\gamma} = \dfrac{\gamma(1-\lambda)}{1-\gamma\lambda}$, we obtain

$$\|\mathbf{d}^\lambda_{\pi_{\theta'}} - \mathbf{d}^\lambda_{\pi_\theta}\|_1 \leq \tilde{\gamma}\|\mathbf{G}_{\pi_{\theta'}}\|_1\|\mathbf{D}\mathbf{d}^\lambda_{\pi_\theta}\|_1 \leq \frac{\tilde{\gamma}}{1-\tilde{\gamma}}\|\mathbf{D}\mathbf{d}^\lambda_{\pi_\theta}\|_1 \tag{127}$$

$$\leq \frac{1}{1-\tilde{\gamma}}\cdot\frac{\tilde{\gamma}\left(\gamma\lambda(|\mathcal{S}|-1)+1\right)}{1-\gamma\lambda}\mathbb{E}_{s\sim d^\lambda_{\pi_\theta}(\cdot)}\left[2D_{\mathrm{TV}}(\pi_{\theta'},\pi_\theta)[s]\right], \tag{128}$$

where the last equation holds due to Lemma 3, this concludes the proof of Lemma 2 .  $\square$

**Lemma 3.** *The term* $\|\mathbf{D}\mathbf{d}^\lambda_{\pi_\theta}\|_1$ *is bounded as follows,*

$$\|\mathbf{D}\mathbf{d}^\lambda_{\pi_\theta}\|_1 \leq \frac{\gamma\lambda(|\mathcal{S}|-1)+1}{1-\gamma\lambda}\mathbb{E}_{s\sim d^\lambda_{\pi_\theta}(\cdot)}\left[2D_{\mathrm{TV}}(\pi_{\theta'},\pi_\theta)[s]\right].$$

*Proof.* Now, we analyze $\|\mathbf{D}\mathbf{d}^\lambda_{\pi_\theta}\|_1$ as follows,

$$\|\mathbf{D}\mathbf{d}^\lambda_{\pi_\theta}\|_1 = \sum_{s\in\mathcal{S}}\left|\sum_{s'\in\mathcal{S}}\mathbf{D}(s'|s)d^\lambda_{\pi_\theta}(s)\right| \overset{(66)}{=} \sum_{s\in\mathcal{S}}\left|\sum_{s'\in\mathcal{S}}\left(\mathbb{P}^{(\lambda)}_{\pi_{\theta'}}(s'|s) - \mathbb{P}^{(\lambda)}_{\pi_\theta}(s'|s)\right)\right|d^\lambda_{\pi_\theta}(s)$$

$$\overset{(66)}{=} \sum_{s\in\mathcal{S}}\left|(1-\gamma\lambda)\sum_{t=0}^{\infty}(\gamma\lambda)^t\sum_{s'\in\mathcal{S}}\left(\mathbb{P}_{\pi_{\theta'}}(s_{t+1}=s'|s) - \mathbb{P}_{\pi_\theta}(s_{t+1}=s'|s)\right)\right|d^\lambda_{\pi_\theta}(s),$$

which implies that to bound $\|\mathbf{D}\mathbf{d}^\lambda_{\pi_\theta}\|_1$, we need to bound the following difference

$$\sum_{t=0}^{\infty}(\gamma\lambda)^t\sum_{s'\in\mathcal{S}}\left(\mathbb{P}_{\pi_{\theta'}}(s_{t+1}=s'|s) - \mathbb{P}_{\pi_\theta}(s_{t+1}=s'|s)\right).$$

**Step 1: Rewrite** $\sum_{t=0}^{\infty}(\gamma\lambda)^t\sum_{s'\in\mathcal{S}}\left(\mathbb{P}_{\pi_{\theta'}}(s_{t+1}=s'|s) - \mathbb{P}_{\pi_\theta}(s_{t+1}=s'|s)\right).$

Let $s_0 = s$, then

$$\mathbb{P}_{\pi_\theta}(s_{t+1}=s'|s) \overset{(55)}{=} \sum_{s_1\in\mathcal{S}}\mathbb{P}_{\pi_\theta}(s_{t+1}=s'|s_1)\mathbb{P}_{\pi_\theta}(s_1|s_0) \tag{129}$$

$$= \sum_{s_1\in\mathcal{S}}\mathbb{P}_{\pi_\theta}(s_{t+1}=s'|s_1)\left(\sum_{a\in\mathcal{A}}\pi_\theta(a|s_0)\mathbb{P}(s_1|s_0,a)\right). \tag{130}$$

Similarly,

$$\mathbb{P}_{\pi_{\theta'}}(s_{t+1} = s'|s) \overset{(55)}{=} \sum_{s_1 \in \mathcal{S}} \mathbb{P}_{\pi_{\theta'}}(s_{t+1} = s'|s_1)\mathbb{P}_{\pi_{\theta'}}(s_1|s_0) \tag{131}$$

$$= \sum_{s_1 \in \mathcal{S}} \mathbb{P}_{\pi_{\theta'}}(s_{t+1} = s'|s_1)\left(\sum_{a \in \mathcal{A}} \pi_{\theta'}(a|s_0)\mathbb{P}(s_1|s_0, a)\right). \tag{132}$$

Firstly, we consider the following term

$$\sum_{t=0}^{\infty}(\gamma\lambda)^t \sum_{s' \in \mathcal{S}} \mathbb{P}_{\pi_{\theta}}(s_{t+1} = s'|s) = \sum_{t=0}^{\infty}(\gamma\lambda)^t \sum_{s' \in \mathcal{S}} \sum_{s_1 \in \mathcal{S}} \mathbb{P}_{\pi_{\theta}}(s_{t+1} = s'|s_1)\left(\sum_{a \in \mathcal{A}} \pi_{\theta}(a|s)\mathbb{P}(s_1|s, a)\right)$$

$$= \sum_{t=0}^{\infty}(\gamma\lambda)^t \sum_{s' \in \mathcal{S}} \sum_{s_1 \in \mathcal{S}} \mathbf{P}_{\pi_{\theta}}^t[s_1, s']\left(\sum_{a \in \mathcal{A}} \pi_{\theta}(a|s)\mathbb{P}(s_1|s, a)\right)$$

$$= \sum_{s' \in \mathcal{S}} \sum_{s_1 \in \mathcal{S}} \left(\sum_{t=0}^{\infty}(\gamma\lambda\mathbf{P}_{\pi_{\theta}})^t\right)[s_1, s']\left(\sum_{a \in \mathcal{A}} \pi_{\theta}(a|s)\mathbb{P}(s_1|s, a)\right)$$

$$= \sum_{s' \in \mathcal{S}} \sum_{s_1 \in \mathcal{S}} (\mathbf{I} - \gamma\lambda\mathbf{P}_{\pi_{\theta}})^{-1}[s_1, s']\left(\sum_{a \in \mathcal{A}} \pi_{\theta}(a|s)\mathbb{P}(s_1|s, a)\right). \tag{133}$$

To short expression, we introduce a new notation as follows,

$$\mathbf{F}_{\pi_{\theta}} = (\mathbf{I} - \gamma\lambda\mathbf{P}_{\pi_{\theta}})^{-1}. \tag{134}$$

Then, we rewrite (133) as follows,

$$\sum_{t=0}^{\infty}(\gamma\lambda)^t \sum_{s' \in \mathcal{S}} \mathbb{P}_{\pi_{\theta}}(s_{t+1} = s'|s) = \sum_{s' \in \mathcal{S}} \sum_{s_1 \in \mathcal{S}} \mathbf{F}_{\pi_{\theta}}[s_1, s']\left(\sum_{a \in \mathcal{A}} \pi_{\theta}(a|s)\mathbb{P}(s_1|s, a)\right). \tag{135}$$

Furthermore, according to (135), we obtain

$$\sum_{t=0}^{\infty}(\gamma\lambda)^t \sum_{s' \in \mathcal{S}} \left(\mathbb{P}_{\pi_{\theta'}}(s_{t+1} = s'|s) - \mathbb{P}_{\pi_{\theta}}(s_{t+1} = s'|s)\right)$$

$$= \sum_{s' \in \mathcal{S}} \sum_{s_1 \in \mathcal{S}} \mathbf{F}_{\pi_{\theta'}}[s_1, s']\left(\sum_{a \in \mathcal{A}} \pi_{\theta'}(a|s)\mathbb{P}(s_1|s, a)\right) - \sum_{s' \in \mathcal{S}} \sum_{s_1 \in \mathcal{S}} \mathbf{F}_{\pi_{\theta}}[s_1, s']\left(\sum_{a \in \mathcal{A}} \pi_{\theta}(a|s)\mathbb{P}(s_1|s, a)\right)$$

$$= \sum_{s' \in \mathcal{S}} \sum_{s_1 \in \mathcal{S}} \mathbf{F}_{\pi_{\theta'}}[s_1, s']\left(\sum_{a \in \mathcal{A}} \pi_{\theta'}(a|s)\mathbb{P}(s_1|s, a)\right) - \sum_{s' \in \mathcal{S}} \sum_{s_1 \in \mathcal{S}} \mathbf{F}_{\pi_{\theta'}}[s_1, s']\left(\sum_{a \in \mathcal{A}} \pi_{\theta}(a|s)\mathbb{P}(s_1|s, a)\right)$$

$$+ \sum_{s' \in \mathcal{S}} \sum_{s_1 \in \mathcal{S}} \mathbf{F}_{\pi_{\theta'}}[s_1, s']\left(\sum_{a \in \mathcal{A}} \pi_{\theta}(a|s)\mathbb{P}(s_1|s, a)\right) - \sum_{s' \in \mathcal{S}} \sum_{s_1 \in \mathcal{S}} \mathbf{F}_{\pi_{\theta}}[s_1, s']\left(\sum_{a \in \mathcal{A}} \pi_{\theta}(a|s)\mathbb{P}(s_1|s, a)\right)$$

$$= \sum_{s' \in \mathcal{S}} \sum_{s_1 \in \mathcal{S}} \mathbf{F}_{\pi_{\theta'}}[s_1, s']\left(\sum_{a \in \mathcal{A}} \left(\pi_{\theta'}(a|s) - \pi_{\theta}(a|s)\right)\mathbb{P}(s_1|s, a)\right) \tag{136}$$

$$+ \sum_{s' \in \mathcal{S}} \sum_{s_1 \in \mathcal{S}} \left(\mathbf{F}_{\pi_{\theta'}}[s_1, s'] - \mathbf{F}_{\pi_{\theta}}[s_1, s']\right)\left(\sum_{a \in \mathcal{A}} \pi_{\theta}(a|s)\mathbb{P}(s_1|s, a)\right), \tag{137}$$

which implies that to bound the following difference

$$\sum_{t=0}^{\infty}(\gamma\lambda)^t \sum_{s' \in \mathcal{S}} \left(\mathbb{P}_{\pi_{\theta'}}(s_{t+1} = s'|s) - \mathbb{P}_{\pi_{\theta}}(s_{t+1} = s'|s)\right),$$

we need to bound (136) and (137).

Due to the simple fact: for any inverse matrices $\mathbf{A}$ and $\mathbf{B}$, then the following identity holds

$$\mathbf{A}^{-1} - \mathbf{B}^{-1} = \mathbf{A}^{-1}(\mathbf{B} - \mathbf{A})\mathbf{B}^{-1},$$

we rewrite the difference $\mathbf{F}_{\pi_{\theta'}} - \mathbf{F}_{\pi_\theta}$ as follows,

$$\mathbf{F}_{\pi_{\theta'}} - \mathbf{F}_{\pi_\theta} = \left(\mathbf{I} - \gamma\lambda\mathbf{P}_{\pi_{\theta'}}\right)^{-1} - (\mathbf{I} - \gamma\lambda\mathbf{P}_{\pi_\theta})^{-1}$$

$$= \gamma\lambda\left(\mathbf{I} - \gamma\lambda\mathbf{P}_{\pi_{\theta'}}\right)^{-1}\left(\mathbf{P}_{\pi_{\theta'}} - \mathbf{P}_{\pi_\theta}\right)(\mathbf{I} - \gamma\lambda\mathbf{P}_{\pi_\theta})^{-1}.$$

Then, we rewrite (137) as the following matrix version

$$\sum_{s'\in\mathcal{S}}\sum_{s_1\in\mathcal{S}}\left(\mathbf{F}_{\pi_{\theta'}}[s_1, s'] - \mathbf{F}_{\pi_\theta}[s_1, s']\right)\left(\sum_{a\in\mathcal{A}}\pi_\theta(a|s)\mathbb{P}(s_1|s,a)\right)$$

$$= \sum_{s''\in\mathcal{S}}\sum_{s'\in\mathcal{S}}\left(\mathbf{F}_{\pi_{\theta'}}[s', s''] - \mathbf{F}_{\pi_\theta}[s', s'']\right)\left(\sum_{a\in\mathcal{A}}\pi_\theta(a|s)\mathbb{P}(s'|s,a)\right) = \left\|\left(\mathbf{F}_{\pi_{\theta'}}^\top - \mathbf{F}_{\pi_\theta}^\top\right)\mathbf{p}_{\pi_\theta}(s)\right\|_1,$$

where $\mathbf{p}_{\pi_\theta}(s) \in \mathbb{R}^{|\mathcal{S}|}$, and

$$\mathbf{p}_{\pi_\theta}(s) = \left(\mathbb{P}_{\pi_\theta}(s_1|s), \mathbb{P}_{\pi_\theta}(s_2|s), \cdots, \mathbb{P}_{\pi_\theta}\left(s_{|\mathcal{S}|}|s\right)\right)^\top.$$

According to (119), we obtain

$$\left\|\left(\mathbf{F}_{\pi_{\theta'}}^\top - \mathbf{F}_{\pi_\theta}^\top\right)\mathbf{p}_{\pi_\theta}(s)\right\|_1 = \left\|\mathbf{p}_{\pi_\theta}^\top(s)\left(\mathbf{F}_{\pi_{\theta'}} - \mathbf{F}_{\pi_\theta}\right)\right\|_\infty$$

$$= \gamma\lambda\left\|\mathbf{p}_{\pi_\theta}^\top(s)\left(\mathbf{I} - \gamma\lambda\mathbf{P}_{\pi_{\theta'}}\right)^{-1}\left(\mathbf{P}_{\pi_{\theta'}} - \mathbf{P}_{\pi_\theta}\right)(\mathbf{I} - \gamma\lambda\mathbf{P}_{\pi_\theta})^{-1}\right\|_\infty$$

$$= \frac{\gamma\lambda}{1-\gamma\lambda}\left\|\underbrace{\mathbf{p}_{\pi_\theta}^\top(s)(1-\gamma\lambda)\left(\mathbf{I} - \gamma\lambda\mathbf{P}_{\pi_{\theta'}}\right)^{-1}}_{\mathbf{f}_s^\top}\left(\mathbf{P}_{\pi_{\theta'}} - \mathbf{P}_{\pi_\theta}\right)(\mathbf{I} - \gamma\lambda\mathbf{P}_{\pi_\theta})^{-1}\right\|_\infty \qquad (138)$$

$$\leq \frac{\gamma\lambda}{1-\gamma\lambda}\left\|\mathbf{f}_s^\top\left(\mathbf{P}_{\pi_{\theta'}} - \mathbf{P}_{\pi_\theta}\right)\right\|_\infty\left\|(\mathbf{I} - \gamma\lambda\mathbf{P}_{\pi_\theta})^{-1}\right\|_\infty$$

$$= \frac{\gamma\lambda}{(1-\gamma\lambda)^2}\left\|\mathbf{f}_s^\top\left(\mathbf{P}_{\pi_{\theta'}} - \mathbf{P}_{\pi_\theta}\right)\right\|_\infty \qquad (139)$$

$$= \frac{2\gamma\lambda}{(1-\gamma\lambda)^2}\sum_{s\in\mathcal{S}}D_{\mathrm{TV}}(\pi_{\theta'}, \pi_\theta)[s], \qquad (140)$$

where in Eq.(138), we introduce a notation $\mathbf{f}_s^\top \in \mathbb{R}^{|\mathcal{S}|}$ as follows,

$$\mathbf{f}_s^\top =: \mathbf{p}_{\pi_\theta}^\top(s)(1-\gamma\lambda)\left(\mathbf{I} - \gamma\lambda\mathbf{P}_{\pi_{\theta'}}\right)^{-1};$$

Eq.(139) holds since:

$$\left\|(\mathbf{I} - \gamma\lambda\mathbf{P}_{\pi_\theta})^{-1}\right\|_\infty = \frac{1}{1-\gamma\lambda};$$

Eq.(140) holds since:

$$\left\|\mathbf{f}^\top\left(\mathbf{P}_{\pi_{\theta'}} - \mathbf{P}_{\pi_\theta}\right)\right\|_\infty = \sum_{s\in\mathcal{S}}\sum_{s'\in\mathcal{S}}\mathbf{f}_s[s']\left|\mathbb{P}_{\pi_{\theta'}}(s'|s) - \mathbb{P}_{\pi_\theta}(s'|s)\right|$$

$$= \sum_{s\in\mathcal{S}}\sum_{s'\in\mathcal{S}}\mathbf{f}_s[s']\left|\sum_{a\in\mathcal{A}}\mathbb{P}(s'|s,a)\left(\pi_{\theta'}(a|s) - \pi_\theta(a|s)\right)\right|$$

$$\leq \sum_{s\in\mathcal{S}}\sum_{s'\in\mathcal{S}}\mathbf{f}_s[s']\sum_{a\in\mathcal{A}}|\pi_{\theta'}(a|s) - \pi_\theta(a|s)|$$

$$= \sum_{s\in\mathcal{S}}\sum_{a\in\mathcal{A}}|\pi_{\theta'}(a|s) - \pi_\theta(a|s)|,$$

where the last equation holds due to the following fact

$$\sum_{s^{'}\in\mathcal{S}}\mathbf{f}_s[s^{'}]=\sum_{s^{'}\in\mathcal{S}}\mathbf{p}_{\pi_{\boldsymbol{\theta}}}^{\top}(s)(1-\gamma\lambda)\left(\mathbf{I}-\gamma\lambda\mathbf{P}_{\pi_{\boldsymbol{\theta}'}}\right)^{-1}[s^{'}]=1.$$

Thus, the difference (137) is bounded as follows,

$$\sum_{s^{'}\in\mathcal{S}}\sum_{s_1\in\mathcal{S}}\left(\mathbf{F}_{\pi_{\boldsymbol{\theta}'}}[s_1,s^{'}]-\mathbf{F}_{\pi_{\boldsymbol{\theta}}}[s_1,s^{'}]\right)\left(\sum_{a\in\mathcal{A}}\pi_{\boldsymbol{\theta}}(a|s)\mathbb{P}(s_1|s,a)\right)\le\frac{2\gamma\lambda}{(1-\gamma\lambda)^2}\sum_{s\in\mathcal{S}}D_{\mathrm{TV}}(\pi_{\boldsymbol{\theta}'},\pi_{\boldsymbol{\theta}})[s].$$

**Step 3: Bound the difference (136).**

We turn to consider (136):

$$\sum_{s^{'}\in\mathcal{S}}\sum_{s_1\in\mathcal{S}}\mathbf{F}_{\pi_{\boldsymbol{\theta}'}}[s_1,s^{'}]\left(\sum_{a\in\mathcal{A}}\left(\pi_{\boldsymbol{\theta}'}(a|s)-\pi_{\boldsymbol{\theta}}(a|s)\right)\mathbb{P}(s_1|s,a)\right)$$

$$=\sum_{s^{''}\in\mathcal{S}}\sum_{s^{'}\in\mathcal{S}}\mathbf{F}_{\pi_{\boldsymbol{\theta}'}}[s^{'},s^{''}]\left(\sum_{a\in\mathcal{A}}\left(\pi_{\boldsymbol{\theta}'}(a|s)-\pi_{\boldsymbol{\theta}}(a|s)\right)\mathbb{P}(s^{'}|s,a)\right)$$

$$=\left\|\mathbf{F}_{\pi_{\boldsymbol{\theta}'}}^{\top}\left(\mathbf{p}_{\pi_{\boldsymbol{\theta}'}}(s)-\mathbf{p}_{\pi_{\boldsymbol{\theta}}}(s)\right)\right\|_1$$

$$\le\left\|\mathbf{F}_{\pi_{\boldsymbol{\theta}'}}^{\top}\right\|_1\left\|\mathbf{p}_{\pi_{\boldsymbol{\theta}'}}(s)-\mathbf{p}_{\pi_{\boldsymbol{\theta}}}(s)\right\|_1$$

$$=\left\|\mathbf{p}_{\pi_{\boldsymbol{\theta}'}}(s)-\mathbf{p}_{\pi_{\boldsymbol{\theta}}}(s)\right\|_1\le\frac{2}{1-\gamma\lambda}D_{\mathrm{TV}}(\pi_{\boldsymbol{\theta}'},\pi_{\boldsymbol{\theta}})[s], \tag{141}$$

where the last Eq.(141) holds since:

$$\left\|\mathbf{F}_{\pi_{\boldsymbol{\theta}'}}^{\top}\right\|_1=\frac{1}{1-\gamma\lambda},$$

and

$$\left\|\mathbf{p}_{\pi_{\boldsymbol{\theta}'}}(s)-\mathbf{p}_{\pi_{\boldsymbol{\theta}}}(s)\right\|_1=\sum_{s^{'}\in\mathcal{S}}\left|\mathbb{P}_{\pi_{\boldsymbol{\theta}'}}(s^{'}|s)-\mathbb{P}_{\pi_{\boldsymbol{\theta}}}(s^{'}|s)\right|$$

$$=\sum_{s^{'}\in\mathcal{S}}\left|\sum_{a\in\mathcal{A}}\mathbb{P}(s^{'}|s,a)\left(\pi_{\boldsymbol{\theta}'}(a|s)-\pi_{\boldsymbol{\theta}}(a|s)\right)\right|$$

$$\le\sum_{a\in\mathcal{A}}\sum_{s^{'}\in\mathcal{S}}\mathbb{P}(s^{'}|s,a)\left|\pi_{\boldsymbol{\theta}'}(a|s)-\pi_{\boldsymbol{\theta}}(a|s)\right|$$

$$=\sum_{a\in\mathcal{A}}\left|\pi_{\boldsymbol{\theta}'}(a|s)-\pi_{\boldsymbol{\theta}}(a|s)\right|=2D_{\mathrm{TV}}(\pi_{\boldsymbol{\theta}'},\pi_{\boldsymbol{\theta}})[s].$$

**Step 4: Put all the result together.**

Finally, according to (136), (137), (140), and (141), we obtain

$$\|\mathbf{D}\mathbf{d}_{\pi_{\boldsymbol{\theta}}}^{\lambda}\|_1=\sum_{s\in\mathcal{S}}\left|(1-\gamma\lambda)\sum_{t=0}^{\infty}(\gamma\lambda)^t\sum_{s^{'}\in\mathcal{S}}\left(\mathbb{P}_{\pi_{\boldsymbol{\theta}'}}(s_{t+1}=s^{'}|s)-\mathbb{P}_{\pi_{\boldsymbol{\theta}}}(s_{t+1}=s^{'}|s)\right)\right|d_{\pi_{\boldsymbol{\theta}}}^{\lambda}(s)$$

$$\le\sum_{s\in\mathcal{S}}d_{\pi_{\boldsymbol{\theta}}}^{\lambda}(s)\left[\frac{2\gamma\lambda}{1-\gamma\lambda}\sum_{s\in\mathcal{S}}D_{\mathrm{TV}}(\pi_{\boldsymbol{\theta}'},\pi_{\boldsymbol{\theta}})[s]+2D_{\mathrm{TV}}(\pi_{\boldsymbol{\theta}'},\pi_{\boldsymbol{\theta}})[s]\right]$$

$$=\sum_{s\in\mathcal{S}}d_{\pi_{\boldsymbol{\theta}}}^{\lambda}(s)\left[\frac{2\gamma\lambda|\mathcal{S}|}{1-\gamma\lambda}D_{\mathrm{TV}}(\pi_{\boldsymbol{\theta}'},\pi_{\boldsymbol{\theta}})[s]+2D_{\mathrm{TV}}(\pi_{\boldsymbol{\theta}'},\pi_{\boldsymbol{\theta}})[s]\right]$$

$$=\frac{\gamma\lambda(|\mathcal{S}|-1)+1}{1-\gamma\lambda}\mathbb{E}_{s\sim d_{\pi_{\boldsymbol{\theta}}}^{\lambda}(\cdot)}\left[2D_{\mathrm{TV}}(\pi_{\boldsymbol{\theta}'},\pi_{\boldsymbol{\theta}})[s]\right].$$

This concludes the result of Lemma 3. $\qquad\qquad\square$

# G   Proof of Theorem 2

Before we present the main result, we define some notations.

$$\chi_k = \mathbb{E}_{s \sim d^\lambda_{\pi_{\boldsymbol{\theta}_k}}(\cdot)} \left[ \mathrm{KL} \left( \pi_{\boldsymbol{\theta}_k}, \pi_{\boldsymbol{\theta}_{k+\frac{1}{2}}} \right) [s] \right], \tag{142}$$

$$\iota = \frac{\tilde{\gamma}\left(\gamma\lambda(|\mathcal{S}| - 1) + 1\right)}{(1 - \tilde{\gamma})(1 - \gamma\lambda)}. \tag{143}$$

*Proof.* (of Theorem 2)

According to Bregman divergence, if policy $\pi_{\boldsymbol{\theta}_k}$ is feasible, policy $\pi_{\boldsymbol{\theta}_{k+1}}$ is generated according to (15), then the following

$$\mathrm{KL}\left(\pi_{\boldsymbol{\theta}_k}, \pi_{\boldsymbol{\theta}_{k+\frac{1}{2}}}\right) \geq \mathrm{KL}\left(\pi_{\boldsymbol{\theta}_k}, \pi_{\boldsymbol{\theta}_{k+1}}\right) + \mathrm{KL}\left(\pi_{\boldsymbol{\theta}_{k+1}}, \pi_{\boldsymbol{\theta}_{k+\frac{1}{2}}}\right)$$

implies

$$\chi_k = \mathbb{E}_{s \sim d^\lambda_{\pi_{\boldsymbol{\theta}_k}}(\cdot)} \left[ \mathrm{KL}\left(\pi_{\boldsymbol{\theta}_k}, \pi_{\boldsymbol{\theta}_{k+\frac{1}{2}}}\right)[s] \right] \geq \mathbb{E}_{s \sim d^\lambda_{\pi_{\boldsymbol{\theta}_k}}(\cdot)} \left[ \mathrm{KL}\left(\pi_{\boldsymbol{\theta}_{k+1}}, \pi_{\boldsymbol{\theta}_k}\right)[s] \right].$$

According to the asymptotically symmetry of KL divergence if we update the policy within a local region, then, we have

$$\chi_k \geq \mathbb{E}_{s \sim d^\lambda_{\pi_{\boldsymbol{\theta}_k}}(\cdot)} \left[ \mathrm{KL}\left(\pi_{\boldsymbol{\theta}_{k+\frac{1}{2}}}, \pi_{\boldsymbol{\theta}_k}\right)[s] \right] \geq \mathbb{E}_{s \sim d^\lambda_{\pi_{\boldsymbol{\theta}_k}}(\cdot)} \left[ \mathrm{KL}\left(\pi_{\boldsymbol{\theta}_{k+1}}, \pi_{\boldsymbol{\theta}_k}\right)[s] \right].$$

Furthermore, according to Proposition 1 and Proposition 3, we have

$$J(\pi_{\boldsymbol{\theta}_{k+1}}) - J(\pi_{\boldsymbol{\theta}_k})$$
$$\geq \frac{1}{1 - \tilde{\gamma}} \mathbb{E}_{s \sim d^\lambda_{\pi_{\boldsymbol{\theta}_k}}(\cdot), a \sim \pi_{\boldsymbol{\theta}_{k+1}}(\cdot|s)} \left[ A^{\mathtt{GAE}(\gamma,\lambda)}_{\pi_{\boldsymbol{\theta}_k}}(s, a) - \iota \epsilon^V_{\pi_{\boldsymbol{\theta}_{k+1}}}(\pi_{\boldsymbol{\theta}_k}) D_{\mathrm{TV}}(\pi_{\boldsymbol{\theta}_k}, \pi_{\boldsymbol{\theta}_{k+1}})[s] \right]$$
$$\geq \frac{1}{1 - \tilde{\gamma}} \mathbb{E}_{s \sim d^\lambda_{\pi_{\boldsymbol{\theta}_k}}(\cdot), a \sim \pi_{\boldsymbol{\theta}_{k+1}}(\cdot|s)} \left[ -\iota \alpha_k \epsilon^V_{\pi_{\boldsymbol{\theta}_{k+1}}}(\pi_{\boldsymbol{\theta}_k}) \sqrt{\frac{1}{2} \mathrm{KL}(\pi_{\boldsymbol{\theta}_k}, \pi_{\boldsymbol{\theta}_{k+1}})[s]} \right]$$
$$\geq -\frac{\iota}{1 - \tilde{\gamma}} \alpha_k \sqrt{2\chi_k} \epsilon^V_{\pi_{\boldsymbol{\theta}_{k+1}}}(\pi_{\boldsymbol{\theta}_k}).$$

Similarly, according to Proposition 1 and Proposition 2, and since policy $\pi_{\boldsymbol{\theta}_{k+1}}$ satisfies

$$J^c(\pi_{\boldsymbol{\theta}_k}) + \frac{1}{1 - \tilde{\gamma}} \mathbb{E}_{s \sim d^\lambda_{\pi_{\boldsymbol{\theta}_k}}(\cdot), a \sim \pi_{\boldsymbol{\theta}_{k+1}}(\cdot|s)} \left[ A^{\mathtt{GAE}(\gamma,\lambda)}_{\pi_{\boldsymbol{\theta}_k}, C}(s, a) \right] + \beta_k \sqrt{\mathbb{E}_{s \sim d^\lambda_{\pi_{\boldsymbol{\theta}_k}}(\cdot)} \left[ \mathrm{KL}(\pi_{\boldsymbol{\theta}_k}, \pi_{\boldsymbol{\theta}_{k+1}})[s] \right]} \leq b, \tag{144}$$

and

$$J^c(\pi_{\boldsymbol{\theta}_{k+1}}) - J^c(\pi_{\boldsymbol{\theta}_k}) \tag{145}$$
$$\leq \frac{1}{1 - \tilde{\gamma}} \mathbb{E}_{s \sim d^\lambda_{\pi_{\boldsymbol{\theta}_k}}(\cdot), a \sim \pi_{\boldsymbol{\theta}_{k+1}}(\cdot|s)} \left[ A^{\mathtt{GAE}(\gamma,\lambda)}_{\pi_{\boldsymbol{\theta}_k}, C}(s, a) + \iota \beta_k \epsilon^C_{\pi_{\boldsymbol{\theta}_{k+1}}} D_{\mathrm{TV}}(\pi_{\boldsymbol{\theta}_k}, \pi_{\boldsymbol{\theta}_{k+1}})[s] \right].$$

Combining (144)- (146), we have

$$J^c(\pi_{\boldsymbol{\theta}_{k+1}}) - J^c(\pi_{\boldsymbol{\theta}_k}) \tag{146}$$
$$\leq b + \frac{1}{1 - \tilde{\gamma}} \mathbb{E}_{s \sim d^\lambda_{\pi_{\boldsymbol{\theta}_k}}(\cdot), a \sim \pi_{\boldsymbol{\theta}_{k+1}}(\cdot|s)} \left[ \iota \beta_k \epsilon^C_{\pi_{\boldsymbol{\theta}_{k+1}}} \sqrt{\frac{1}{2} \mathbb{E}_{s \sim d^\lambda_{\pi_{\boldsymbol{\theta}_k}}(\cdot)} \left[ \mathrm{KL}(\pi_{\boldsymbol{\theta}_k}, \pi_{\boldsymbol{\theta}_{k+1}})[s] \right]} \right]$$
$$\leq b + \frac{1}{1 - \tilde{\gamma}} \mathbb{E}_{s \sim d^\lambda_{\pi_{\boldsymbol{\theta}_k}}(\cdot), a \sim \pi_{\boldsymbol{\theta}_{k+1}}(\cdot|s)} \left[ \iota \beta_k \sqrt{2\chi_k} \epsilon^C_{\pi_{\boldsymbol{\theta}_{k+1}}} \right]. \tag{147}$$

$\square$

# H  Experiments

The Python code for our implementation of CUP is provided along with this submission in the supplementary material.

All experiments were implemented in Pytorch 1.7.0 with CUDA 11.0 and conducted on an Ubuntu 20.04.2 LTS (GNU/Linux 5.8.0-59-generic x86 64) with 40 CPU cores (Intel(R) Xeon(R) Silver 4210R CPU @ 2.40GHz), 251G memory and 4 GPU cards (GeForce RTX 3080). The baseline algorithm FOCOPS based on the open-source `https://github.com/ymzhang01/focops`, which were offical code library. The other baseline algorithms include CPO, TRPO-L, PPO-L based on `https://github.com/openai/safety-starter-agents`, which published by openai.

## H.1  Algorithm Parameters

| Hyperparameter | CUP | PPO-L | TRPO-L | CPO | FOCOPS |
|---|---|---|---|---|---|
| No. of hidden layers | 2 | 2 | 2 | 2 | 2 |
| No. of hidden nodes | 64 | 64 | 64 | 64 | 64 |
| Activation | tanh | tanh | tanh | tanh | tanh |
| Initial log std | -0.5 | -0.5 | -1 | -0.5 | -0.5 |
| Discount for reward $\gamma$ | 0.99 | 0.99 | 0.99 | 0.99 | 0.99 |
| Discount for cost $\gamma_C$ | 0.99 | 0.99 | 0.99 | 0.99 | 0.99 |
| Batch size | 5000 | 5000 | 5000 | 5000 | 5000 |
| Minibatch size | 64 | 64 | N/A | N/A | 64 |
| No. of optimization epochs | 10 | 10 | N/A | N/A | 10 |
| Maximum episode length | 1000 | 1000 | 1000 | 1000 | 1000 |
| GAE parameter (reward) | 0.95 | 0.95 | 0.95 | 0.95 | 0.95 |
| GAE parameter (cost) | 0.95 | 0.95 | 0.95 | 0.95 | 0.95 |
| Learning rate for policy | $3 \times 10^{-4}$ | $3 \times 10^{-4}$ | N/A | N/A | $3 \times 10^{-4}$ |
| Learning rate for reward value net | $3 \times 10^{-4}$ | $3 \times 10^{-4}$ | $3 \times 10^{-4}$ | $3 \times 10^{-4}$ | $3 \times 10^{-4}$ |
| Learning rate for cost value net | $3 \times 10^{-4}$ | $3 \times 10^{-4}$ | $3 \times 10^{-4}$ | $3 \times 10^{-4}$ | $3 \times 10^{-4}$ |
| Learning rate for $\nu$ | 0.01 | 0.01 | 0.01 | N/A | 0.01 |
| $L2$-regularization coeff. for value net | $10^{-3}$ | $3 \times 10^{-3}$ | $3 \times 10^{-3}$ | $3 \times 10^{-3}$ | $10^{-3}$ |
| Clipping coefficient | N/A | 0.2 | N/A | N/A | N/A |
| Damping coeff. | N/A | N/A | 0.01 | 0.01 | N/A |
| Backtracking coeff. | N/A | N/A | 0.8 | 0.8 | N/A |
| Max backtracking iterations | N/A | N/A | 10 | 10 | N/A |
| Max conjugate gradient iterations | N/A | N/A | 10 | 10 | N/A |
| Iterations for training value net | 1 | 1 | 80 | 80 | 1 |
| Temperature $\lambda$ | 1.5 | N/A | N/A | N/A | 1.5 |
| Trust region bound $\delta$ | 0.02 | N/A | 0.01 | 0.01 | 0.02 |
| Initial $\nu$, $\nu_{\max}$ | 0, 2 | 0, 1 | 0, 2 | N/A | 0, 2 |

Table 3: Hyper-parameters for robots.

## H.2  Environment

### H.2.1  Environment 1: Robots with Speed Limit.

We consider two tasks from MuJoCo [Brockman *et al.*, 2016]: Walker2d-v3 and Hopper-v3, where the setting of cost follows [Zhang *et al.*, 2020]. For agents move on a two-dimensional plane, the cost is calculated as follows,

$$C(s,a) = \sqrt{v_x^2 + v_y^2};$$

for agents move along a straight line, the cost is calculated as

$$C(s,a) = |v_x|,$$

where $v_x$, $v_y$ are the velocities of the agent in the $x$ and $y$ directions respectively.

### H.2.2 Environment 2: Circle.

The Circle Environment follows [Achiam *et al.*, 2017], and we use open-source implementation of the circle environments from `https://github.com/ymzhang01/mujoco-circle`. According to Zhang *et al.* [2020], those experiments were implemented in OpenAI Gym [Brockman *et al.*, 2016] while the circle tasks in Achiam *et al.* [2017] were implemented in rllab [Duan *et al.*, 2016]. We also excluded the Point agent from the original experiments since it is not a valid agent in OpenAI Gym. The first two dimensions in the state space are the $(x, y)$ coordinates of the center mass of the agent, hence the state space for both agents has two extra dimensions compared to the standard Ant-v0 and Humanoid-v0 environments from OpenAI Gym.

Now, we present some necessary details of this environment taken from [Zhang *et al.*, 2020].

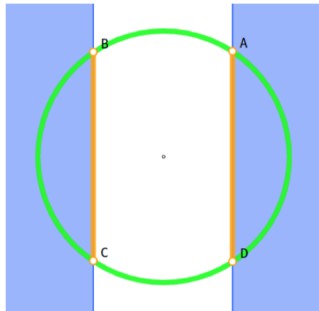

Figure 4: In the Circle task, reward is maximized by moving along the green circle. The agent is not allowed to enter the blue regions, so its optimal constrained path follows the line segments $AD$ and $BC$ (figure and caption taken from [Achiam *et al.*, 2017; Zhang *et al.*, 2020]).

In the circle tasks, the goal is for an agent to move along the circumference of a circle while remaining within a safety region smaller than the radius of the circle. The exact geometry of the task is shown in Figure 4. The reward and cost functions are defined as:

$$R(s) = \frac{-yv_x + xv_y}{1 + |\sqrt{x^2 + y^2} - r|}, \;\; C(s) = \mathbb{I}(|x| > x_{\lim}),$$

where $x, y$ are the positions of the agent on the plane, $v_x, v_y$ are the velocities of the agent along the $x$ and $y$ directions, $r$ is the radius of the circle, and $x_{\lim}$ specifies the range of the safety region. The radius is set to $r = 10$ for both Ant and Humanoid while $x_{\lim}$ is set to 3 and 2.5 for Ant and Humanoid respectively. Note that these settings are identical to those of the circle task in Achiam *et al.* [2017]; Zhang *et al.* [2020].

### H.3 Safety Gym

In Safety Gym environments, the agent perceives the world through a robot's sensors and interacts with the world through its actuators [Ray *et al.*, 2019]. In this section, we consider two robots: Point and Car, where the presentation of those safety environments are taken from [Ray *et al.*, 2019], for more details, please refer to [Ray *et al.*, 2019, Page 8–10]. In this section, we experiment with the Safety Gym environment-builder two tasks: Goal, Button.

### H.3.1 Safety Gym Robots

We consider two robots: Point and Car. All actions for all robots are continuous and linearly scaled to $[-1, +1]$, which is typical for 3D robot-based RL environments and (anecdotally) improves learning with neural nets. Modulo scaling, the action parameterization is based on a mix of hand-tuning and MuJoCo actuator defaults, and we caution that it is not clear if these choices are optimal. Some safe exploration techniques are action-layer interventions, like projecting to the closest predicted safe action [Dalal *et al.*, 2018], and these methods can be sensitive to action parameterization. As a result, action parameterization may merit more careful consideration than is usually given. Future work on

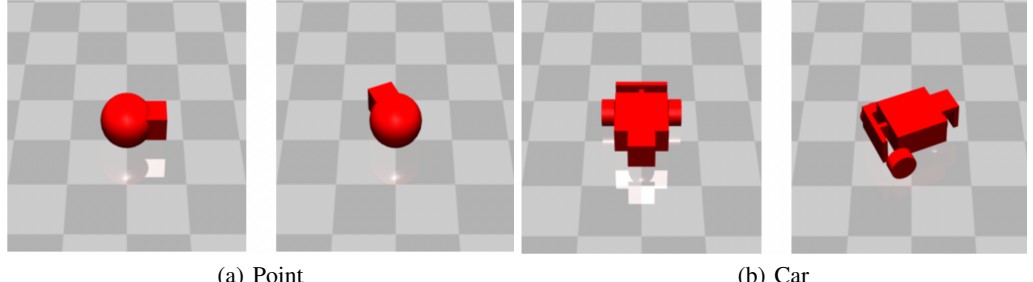

(a) Point                                       (b) Car

Figure 5: Fig (a): a 2D robot that can turn and move; Fig (b): a wheeled robot with a differential drive control, in "Button", the objective is to press the highlighted button (visually indicated with a faint gray cylinder), where figures and caption taken from Safety Gym [Ray *et al.*, 2019].

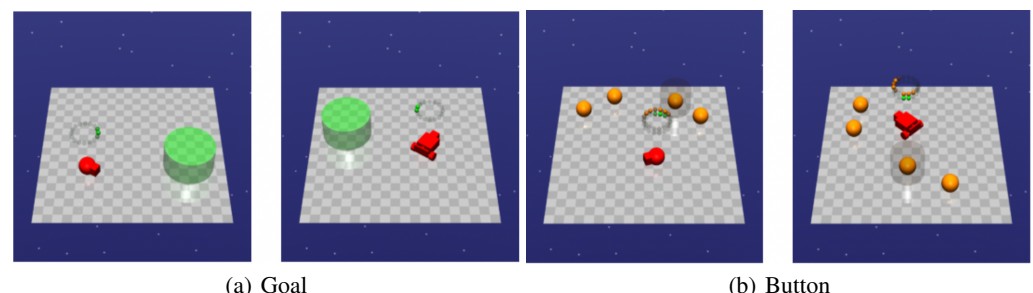

(a) Goal                                        (b) Button

Figure 6: Fig (a): In "Goal," the objective is to move the robot inside the green goal area; Fig (b): In "Button", the objective is to press the highlighted button (visually indicated with a faint gray cylinder), where figures and caption are taken from Safety Gym [Ray *et al.*, 2019].

action space design might be to find action parameterizations that respect physical measures we care about—for example, an action space where a fixed distance corresponds to a fixed amount of energy.

**Point**: A robot constrained to the 2D plane, with one actuator for turning and another for moving forward/backward. This factored control scheme makes the robot particularly easy to control for navigation. Point has a small square in front that makes it easier to visually determine the robot's direction and helps the point push a box element that appears in one of our tasks.

**Car**: The car is a slightly more complex robot that has two independently-driven parallel wheels and a free-rolling rear wheel. The car is not fixed to the 2D plane but mostly resides in it. For this robot, both are turning and moving forward/backward require coordinating both of the actuators. It is similar in design to simple robots used in education.

### H.3.2    Tasks

Tasks in Safety Gym are mutually exclusive, and an individual environment can only use a single task. Reward functions are configurable, allowing rewards to be either sparse (rewards only obtained on task completion) or dense (rewards have helpful, hand-crafted shaping terms). Task details are shown as follows.

**Goal**: Move the robot to a series of goal positions. When a goal is achieved, the goal location is randomly reset to someplace new, while keeping the rest of the layout the same. The sparse reward component is attained on achieving a goal position (robot enters the goal circle). The dense reward component gives a bonus for moving towards the goal (shown in Figure 5).

**Button**: Press a series of goal buttons. Several immobile "buttons" are scattered throughout the environment, and the agent should navigate to and press (contact) the currently-highlighted button, which is the goal button. After the agent presses the correct button, the environment will select and highlight a new goal button, keeping everything else fixed. The sparse reward component is attained

| Environment | | CPO | TRPO-L | PPO-L | FOCOPS | CUP |
|---|---|---|---|---|---|---|
| Safexp-PointGoal1-v0 | Return | $21.29 \pm 3.49$ | $19.23 \pm 1.45$ | $16.17 \pm 5.89$ | $12.46 \pm 1.49$ | $\mathbf{23.74 \pm 0.12}$ |
| Cost limit (25.0) | Constraint | $39.00 \pm 5.19$ | $28.20 \pm 5.21$ | $21.82 \pm 6.31$ | $34.67 \pm 2.62$ | $24.74 \pm 0.91$ |
| Safexp-PointButton1-v0 | Return | $17.69 \pm 1.22$ | $5.39 \pm 1.02$ | $4.74 \pm 2.73$ | $8.36 \pm 0.34$ | $\mathbf{19.52 \pm 1.38}$ |
| Cost limit (25.0) | Constraint | $69.61 \pm 8.29$ | $25.15 \pm 4.88$ | $30.37 \pm 7.58$ | $18.56 \pm 1.31$ | $26.67 \pm 1.84$ |
| Safexp-CarGoal1-v0 | Return | $\mathbf{33.00 \pm 0.00}$ | $17.78 \pm 2.34$ | $19.93 \pm 1.13$ | $17.73 \pm 3.50$ | $27.41 \pm 1.80$ |
| Cost limit 25.0) | Constraint | $30.50 \pm 1.44$ | $23.00 \pm 4.11$ | $29.64 \pm 4.79$ | $25.50 \pm 1.43$ | $30.81 \pm 1.60$ |
| Safexp-CarButton1-v0 | Return | $5.80 \pm 1.06$ | $0.48 \pm 0.15$ | $0.41 \pm 0.13$ | $9.47 \pm 1.67$ | $\mathbf{12.12 \pm 1.91}$ |
| Cost limit (25.0) | Constraint | $93.88 \pm 13.90$ | $23.17 \pm 9.76$ | $16.23 \pm 15.55$ | $19.60 \pm 1.52$ | $29.41 \pm 0.40$ |

Table 4: Average results for CPO, PPO-L, TRPO-L, FOCOPS, CUP over 10 seeds after 500 iterations on Safety-Gym. The agent interacts with the environment 5000 times per iteration. Constraint limits are in brackets under the environment names.

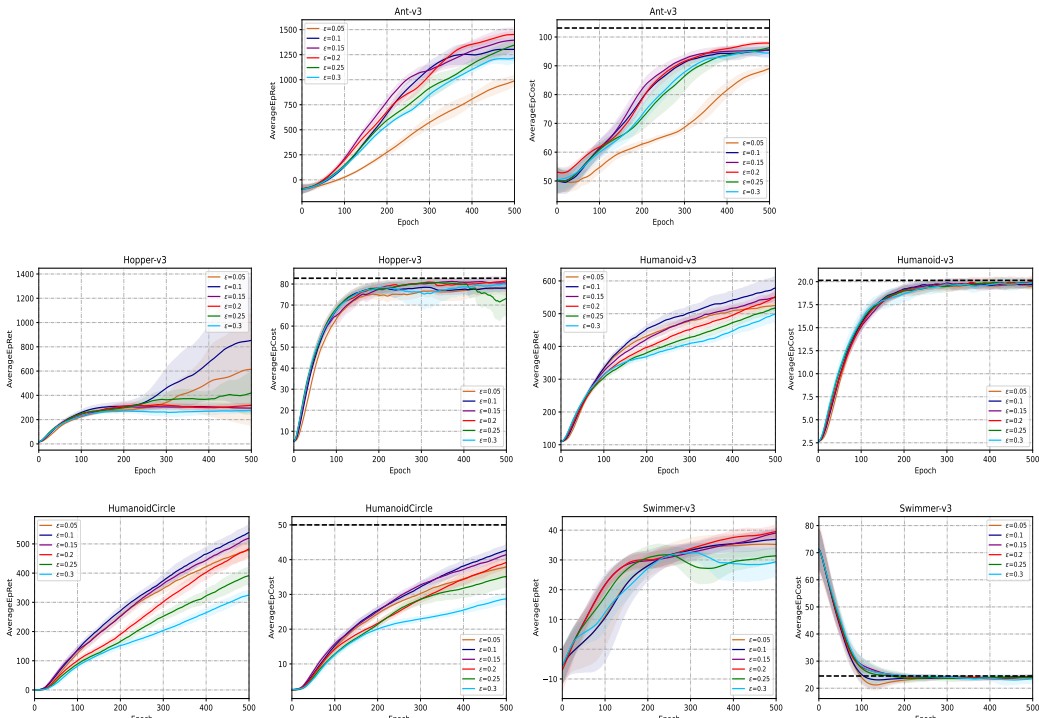

Figure 7: Performance with respect to penalty factor $\epsilon$ appears in Algorithm 1.

on pressing the current goal button. The dense reward component gives a bonus for moving towards the current goal button. We show a visualization in Figure 6).

### H.4 Discussions

Results of Figure 7 show that the performance of CUP is still very stable for different settings of $\epsilon$. Additionally, the constraint value of CUP also still fluctuates around the target value. The different value achieved by CUP in different setting $\epsilon$ is affected by the simulated environment and constraint thresholds, which are easy to control

The results of Table 4 show that the proposed CUP significantly outperforms all the baseline algorithms except on the Safexp-CarGoal1-v0 task. Notably, on the Safexp-PointButton1-v0 task, CUP achieve $21.27 \pm 1.42$ within the safety region, while the best baseline algorithm is CPO that only obtains a reward of $17.69 \pm 1.22$ but it violates the cost limit 25 more than a value of 44. This result is consistent with the result of Figure 2. Besides, from Table 4, we know although CPO achieves a reward of $33 \pm 00$ significantly outperforms the proposed CUP in Safexp-CarGoal1-v0, CPO needs a cost $30.50 \pm 1.44$ higher than CUP.