# OpenReview forum: "Constrained Update Projection Approach to Safe Policy Optimization"
_NeurIPS.cc/2022/Conference — NeurIPS 2022 Accept_

### Official Review · Reviewer_px3X · 2022-07-01

**Rating:** 6
**Confidence:** 4
**Soundness:** 3 good
**Presentation:** 4 excellent
**Contribution:** 3 good

**Summary:**

1. A new GAE style surrogate function for solving CMDP problem
2. The theory covers the previous well-known results
3. Do not require to compute the second oder information, unlike CPO and PCPO.


**Questions:**

Please answer my questions in the weaknesses.

**Limitations:**

1. The similarity to PCPO

**Strengths And Weaknesses:**

Pros
1. The theory looks technically-sound
2. The related work section is good. I like the comparison table of different prior algorithm shown in the appendix.
3. In the experiment section, the claim that PCPO and CPO uses the convex relaxation results of larger errors is back up by the experiment in figure 2.
4. Nice ablation studies on the projection parameter $v$, the step size, and the cost penalty. It clearly shows that the algorithm is stable, easy to optimize, and can follow the cost constraint threshold well no matter how hard the threshold is.

Cons
1. In algorithm 1 in the appendix, the protection step requires to compute the KL divergence between the old policy and the target policy, does this require to compute second order information to do an update? If this is the case, then the contribution of using less computation would not hold.
2. The idea of using projection has been extensively used in PCPO paper. The difference between PCPO and this work is the theoretical analysis. In addition, formulating the GAE problem has been discussed in Zhong et al paper. The difference between Zhong et al and this work is again the theoretical analysis.

---

> ### Author Response · Authors · 2022-07-30
> **Response to questions from Reviewer px3X**
>
> We all thank the reviewer for your professional comments and suggestions. We wish to clear all your concerns as follows.
>
> **Q1: "In algorithm 1 in the appendix, the protection step requires to compute the KL divergence between the old policy and the target policy, does this require to compute second order information to do an update?"**
>
> **A1:** The update of algorithm 1 does **NOT** depend on any second-order information. The algorithm 1 updates all the policy parameters according to the first order optimizer (such as Adam, AadGrad, et.al).
> The key idea of the update rule of algorithm 1 has been shown in Eq.(41)-(42), see page 21 (line 626), which is based on the stochastic gradient descent methods and does not contain any second-order information.
>
>
> **Q2: "The similarity to PCPO."**
>
> **A2:** PCPO provides much inspiration for us to design the proposed CUP algorithm, especially, the **projection** operator of PCPO leads us to propose new algorithm. However, our submission improves safe RL algorithms (not limited in PCPO) at least three aspects:
>
> *  Our rigorous theoretical analysis shows a bound with respect to GAE. Although GAE has been empirically applied to extensive reinforcement learning tasks, **to the best of our knowledge, the result of our submission is the first to show a rigorous theoretical analysis to extend the surrogate functions to GAE.** The proof is NOT a trivial result, which depends on a generalized expression of objective
> $J(\pi)$ as follows:
> $$J(\pi)=\mathbb E_{s_0\sim\rho_0(\cdot)}[\varphi(\cdot)]+\dfrac{1}{1-\gamma}\mathbb E_{s_0\sim\rho_0(\cdot)}\left[\sum_{t=0}^{\infty}\gamma^t\lambda^t\delta_t^{\varphi}\right],~~\text{\color{blue}{(EQ2)}},$$
> where $\delta_t^{\varphi}=E_{s_t\sim P_{\pi}(\cdot|s_t),a_t\sim \pi(\cdot|s_t),s_{t+1}\sim P(\cdot|s_t,a_t)}\left[r_{t+1}+\gamma\varphi(s_{t+1})-\varphi(s_{t})\right]$ is a new expected TD-error with respect to a function $\varphi$. **To the best of our knowledge, such a novel expression $J(\pi)$
>  has not appeared in any literature before our submission.** This novel expression $J(\pi)$ plays an important role for us to show GAE.
>  If $\lambda\ne0,\gamma\ne0$, we see that
> $J(\pi))(\text{\color{blue}{EQ2}})$ has a remarkably simple formula involving a discounted sum of TD errors, which is analogue to TD$(\lambda)$. Significantly, $J(\pi) )(\text{\color{blue}{EQ2}})$is not a simple extension TD$(\lambda)$ since those TD errors $\delta_t^\varphi$ is determined by $\varphi$. Furthermore, if $\phi=V_{\pi}$, then $\delta_t^\varphi$ is reduced to $\delta_t=\mathbb E_{s_t\sim P_{\pi}(\cdot|s_t),a_t\sim \pi(\cdot|s_t),s_{t+1}\sim P(\cdot|s_t,a_t)}\left[r_{t+1}+\gamma V(s_{t+1})-V(s_{t})\right],$ then the following term in $(\text{\color{blue}{EQ2}})$
> $$
> \dfrac{1}{1-\gamma}\mathbb E_{s_0\sim\rho_0(\cdot)}\left[\sum_{t=0}^{\infty}\gamma^t\lambda^t\delta_t^{\varphi}\right]
> $$
> reduces to an estimator of advantage $A_{\pi}$, which is the key idea of GAE.  For more discussion, please refer to our response https://openreview.net/forum?id=22hMrSbQXzt&noteId=pasZh-pX6Ce
>
> * The implementation of PCPO depends on convex approximation and second order information, which makes the proposed algorithm requires less memory and compute efficiently. For this point, we have clear it in **A1**.
>
> * Finally, we also use empirical comparisons between CUP and PCPO. The proposed CUP performs better than PCPO. We believe this is not an accidental phenomenon since CUP optimizes the non-convex objective directly, while PCPO considers a convex approximation than may lead to many error sources and troubles. Besides, the convex approximation may be far away from the original safe policy optimization, while CUP does not require any strong approximation of the convexity to the objectives.
>
>
> **Q3: formulating the GAE problem has been discussed in Zhong et al paper.**
>
> **A3:** The reviewer refers to [Zhong et al], it seems to refer to FOCOPS [Zhang et al, 2020]? not [Zhong et al]? The detailed implementation from FOCOPS covers the GAE. However, the theory part of FOCOPS still lacks a rigorous theoretical analysis with respect to GAE.

---

> > ### Comment · Reviewer_px3X · 2022-08-08
> > **Thanks**
> >
> > Dear Authors,
> > Thank you for the rebuttal. I would like to keep the score the same.

---

### Official Review · Reviewer_8pzD · 2022-07-12

**Rating:** 7
**Confidence:** 3
**Soundness:** 3 good
**Presentation:** 3 good
**Contribution:** 3 good

**Summary:**

This work extends the performance difference bounds for Constrained Policy Optimization (CPO, Achiam et al, 2017) to a generalized setting that includes GAE in the bound. This provides a theoretical justification for using GAE in CPO-like methods as well as allows for a new approach, CUP, that explicitly uses these new general bounds with an implementation that does not depend on the convex approximation. The authors then validate their new approach on a variety of safe RL tasks and baselines and show the effectiveness of their approach.

**Questions:**

- Significance of Remark 1 is not clear. Comparison with Kakade has already been made in CPO. The generalized bounds with GAE don't really provide any new insights w.r.t. that comparison or do they? Will it be better to instead have a remark on comparison with CPO instead?


**Limitations:**

The authors don't really discuss the limitations of their approach. Are there any scenarios where it makes sense (say computationally) to use any of the other baseline rather than the CUP approach?

**Strengths And Weaknesses:**

# Strengths :

- The results in the paper are original and bridge the missing justification for the use of GAE in CPO-based methods. The paper is also mostly well written for the most part.

- The empirical study in the paper is done rigorously which further demonstrates the effectiveness of their method. Moreover, the code is provided in the supplemental that will also help future research in this area.

# Weaknesses:

- The aspects of the paper can be improved when it comes to clarity. For instance, one of the major contributions of the work is how the optimize the objective function via the first-order method (L:208-212, L:245), however, most of the details regarding this have been pushed to the Appendix instead. I would recommend the authors to expand a bit more on these aspects. Though the results are novel, most of the proof methodology seems to stem directly from the CPO proof techniques. Maybe some of the details that are direct extensions of CPO like L:177-182 can be moved to the Appendix instead and space can be provided to move the Algorithm to the main paper.

- More details should be provided on the comparison with FOCOPS. It seems like FOCOPS is the most competitive baseline, with some similarities with CUP approach such as the update rule for $\nu$ is the same as FOCOPS (L:296). In Sec 6.2, while the plots for Fig 3 (a) make sense, they are not particularly interesting as they follow the same rule as FOCOPS. A more interesting comparison could have been on the choice of initial $\nu$ and the choice of $\eta$.

---

> ### Author Response · Authors · 2022-07-30
> **Response to Reviewer 8pzD**
>
> We gratefully appreciate the reviewer for recommending acceptance of our submission and thank his/her valuable suggestions to help improve it. We will answer all the questions that the reviewers concern about.
>
> **Q1: The first comment in Weaknesses:"The aspects of the paper can be improved when it comes to clarity....", and Questions: "Significance of Remark 1 is not clear. Comparison with Kakade has already been made in CPO. The generalized bounds with GAE don't really provide any new insights w.r.t. that comparison or do they? Will it be better to instead have a remark on comparison with CPO instead?"**
>
>
> **A1:** We all thank for your valuable suggestions. In the revision, we will rewrite remark 1 from the aspect of the comparison to CPO. Then, we will provide the insights on how the optimize the objective function via the first-order method. We will update the revision as soon as possible.
>
> **Q2: The second comment in Weaknesses: "A more interesting comparison could have been on the choice of initial $\nu$ and the choice of $\eta$."**
>
> **A2:** We have considered this suggestion, and we will show the comparison could have been on the choice of initial $\nu$ and the choice of $\eta$ in the revision.
>
>
> **Q3: "The authors don't really discuss the limitations of their approach. Are there any scenarios where it makes sense (say computationally) to use any of the other baseline rather than the CUP approach?"**
>
> **A3:** In this submission, we only consider the single constraint, but do not consider the multiple constraints. The multiple constraints are closer to real-world applications, but the multiple constraints are difficult to be learned. Besides, in real-world applications, for example, autonomous vehicle or power systems it is catastrophic if the system plays violations of constraints. Thus, achieving its goal guarantees zero constraint violation, an important problem for safety learning. But the proposed CUP does not satisfy this case.

---

> > ### Comment · Reviewer_8pzD · 2022-08-08
> > **Thanks for the response**
> >
> > Thank you for your response. I would encourage the authors to expand a bit more on the comparison with related work in the main draft (maybe parts of Section B.4 can be moved?). My stance on the paper has not changed, so I'm keeping the same score.

---

> > > ### Author Response · Authors · 2022-08-09
> > > **Response**
> > >
> > > Thanks for your further feedback.
> > >
> > >
> > > We have provided a comparison in the main draft, see page 7, 235-241.

---

> > > ### Author Response · Authors · 2022-08-09
> > > **Re: reversion according to your suggestions.**
> > >
> > > Thanks for your further feedback.
> > >
> > > We have updated a new version of the submission, where we provided comparison in the main draft, see page 7, 235-241, and remove the  Section B.4 according to your suggestions.

---

> ### Author Response · Authors · 2022-08-02
> **Revision revision according to the suggestions.**
>
> We provide a revision according to your suggestions:
>
> * In the revision, we provide some additional experiments for performance with respect to initial $\eta$ and $\nu$, see Appendix H.5, page 44-45;
>
> * In the revision, we rewrite the Remark 1, where we remove original Remark 1 and provide a discussion from the aspect of the comparison to CPO, see Section 3.2, page 5;
>
> * In the revision, we provide the insights on how the optimize the objective function via the first-order method, see page 6, line 195-202.

---

### Official Review · Reviewer_tK1z · 2022-07-12

**Rating:** 5
**Confidence:** 4
**Soundness:** 3 good
**Presentation:** 2 fair
**Contribution:** 3 good

**Summary:**

This paper proposes a new safe RL method based on the constrained updated projection framework(CUP).  The authors use CUP to generalize the surrogate objective and cost(constraints) functions and solve them by the first-order method, which facilities the safety analysis for RL. The authors also compared against some safe RL baselines to show their effectiveness.

**Questions:**

Please refer to the detailed weakness points.

**Limitations:**

Missing a significant literature review and comparison.
The claim of safety guarantees is too strong for the paper.

**Strengths And Weaknesses:**

Strengths:
The paper derives the theoretical bounds for the surrogate functions of objective and constraints to achieve monotonic policy improvement with an asymptotic safety guarantee. They claimed this work is the first to show a rigorous theoretical analysis to extend the surrogate functions w.r.t GAE.

Weaknesses:
1. The paper may miss a significant part of the literature review and experiment comparison.  I did a quick search for the related works.
a. CRPO: A New Approach for Safe Reinforcement Learning with Convergence Guarantee(last year's ICML paper), claims it can achieve optimality and zero constraints violation in a sub-linear rate. Although the authors mentioned it in the paper briefly, they didn't compare it in the experiment. In CRPO, the baselines include primal-dual optimization (PDO) and interior point optimization (IPO), the author should explain why they didn't compare with these methods either.

My understanding of safe RL does not only include the solutions for the constrained optimization problem(like CPO) but also some methods leveraging the formal methods(such as barrier function, and control barrier function, temporal logic), which this paper didn't cover at all. Some references here:
b. IPO: Interior-point Policy Optimization under Constraints (by adding the barrier penalty)
c. End-to-End Safe Reinforcement Learning through Barrier Functions for Safety-Critical Continuous Control Tasks.
d. Temporal Logic Guided Safe Reinforcement Learning Using Control Barrier Functions.

2. Another major concern is how the safety 'guarantee' is expressed in this paper. For example, in CRPO, they show that the safety violation is in a sub-linear rate while this paper only claimed they can have asymptotic constraint satisfaction, without quantitative results. It is even worse when compared to the formal methods-based approaches where they can really achieve rigorous safety guarantees.  Thus, the claim of the safety guarantee in the paper is too strong.

---

> ### Author Response · Authors · 2022-08-01
> **Re to Weaknesses 1**
>
> We all thank the reviewer for his/her careful comments and suggestions. We wish to clear all your concerns in Weaknesses 1 as follows.
>
> **Q1:**  About "primal-dual optimization (PDO)".
>
> **A1:**  After a careful reading of  CRPO paper, we find the PDO algorithm in CRPO is as same as PPO-L in our submission, i.e., we have implemented and compared the PDO algorithm. Both PPO-L and PDO are combining the Lagrange method with PPO. To our best knowledge, the key idea of PPO-L (or PDO) firstly appears in [Ray, et, al. 2019].
>
> [Ray,et,al. 2019] Alex Ray, Joshua Achiam, and Dario Amodei. Benchmarking Safe Exploration in Deep Reinforcement Learning. 2019.
>
>
> **Q2:**  About "CRPO: A New Approach for Safe Reinforcement Learning with Convergence Guarantee"
>
> **A2:**  It is still very difficult for us to reproduce CRPO according to its pseudo-code. Recently, we also try our best to code (or refine it according to our understanding) the algorithm according to CRPO and tuning extensive parameters, but CRPO does **NOT** work on any MuJoco tasks.
> Now, we present our main difficulty to reproduce CRPO as follows.
>
> * The Algorithm 1 relates to $\rho_{j,t}$, however, CRPO paper does NOT present how to set $\rho_{j,t}$.
>
> * The update rule w.r.t, the parameter of CRPO also confuses us very much. Let us recall Eq.(4) in the CRPO (https://arxiv.org/pdf/2011.05869.pdf), it requires updating $\theta$ as follows,
> $$\theta^i_{k+1}(s,a)\leftarrow \theta^i_{k}(s,a)+\beta_{k}[c_{i}(s,a,s')+\gamma\theta^i_{k}(s',a')-\theta^i_{k}(s,a)],{\color{blue}{(\text{EQ3})}}$$
> which confuses us with at least 3 expects:**1.)** this update does not work on continuous control tasks since it is a tabular learning algorithm. However, according to [CRPO,https://arxiv.org/pdf/2011.05869.pdf, Appendix A], they run the continuous control tasks;
> **2.)** If CRPO uses a linear function for policy evaluation, the update rule should share the following formulation:
> $$\theta^i_{k+1}(s,a)\leftarrow \theta^i_{k}(s,a)+\beta_{k}[c_{i}(s,a,s')+\gamma\theta^i_{k}(s',a')x(s',a')-\theta^i_{k}(s,a)x(s,a)],$$
> where $x(\cdot)$ is feature map; besides CRPO paper also needs to show how to set feature map;
> **3.)** If CRPO uses non-linear function for policy evaluation, it needs to use stochastic gradient descent to update $\theta$, and it seems to be very difficult to share the update rule as ${\color{blue}{(\text{EQ3})}}$.
>
>
> **Q3:**  About "IPO: Interior-point Policy Optimization under Constraints"
>
> **A3:** We refer to IPO according to https://ojs.aaai.org//index.php/AAAI/article/view/5932. We find a bug in IPO.
> Recall the definition of $\hat{J}^{\pi_{\theta}}_{C_i}$ in Page 4942,
>
> if the current policy $\pi_\theta$ gets stuck in an unsafe region,
> then
>
> $\hat{J}^{\pi_{\theta}}_{C_i}<0$,
>
>
> else   $\hat{J}^{\pi_{\theta}}_{C_i}\ge0$.
>
> Then the bug lies on the page 4943, the following term $\phi$ is **ill-defined**
>
> $$\phi=\dfrac{\log\left(-\hat{J}^{\pi_{\theta}}_{C_i}\right)}{t}$$
>
> since $\hat{J}^{\pi_{\theta}}_{C_i}$ can NOT keep negative always. To show this point, we also print the date run by IPO (that implemented by ourselves), which also supports above claims.

---

> ### Author Response · Authors · 2022-08-01
> **Re to Safety in Weaknesses 1-2**
>
> In this comment, we wish to clear all your concerns about safe reinforcement learning.
>
>
> **Q1:** "My understanding of safe RL does not only include the solutions for the constrained optimization problem(like CPO) but also some methods leveraging the formal methods(such as barrier function, and control barrier function, temporal logic), which this paper didn't cover at all."
>
> **A1:** First, we all agree with the reviewer's above comments about safe RL.
> In fact, safe RL is a very broad issue in RL, which relates extensive works. In this submission, we provide one view from constrained MDP to study safe RL. In the revision, we will provide some basic discussion and comparisons with barrier function in the related work section.
>
>
> According to the classic work [Garcia&Fernandez, 2015,https://jmlr.org/papers/volume16/garcia15a/garcia15a.pdf ], this work categorizes and analyze two approaches of Safe Reinforcement Learning. The first is based on **the modification of the optimality criterion**, the classic discounted finite/infinite horizon, with a safety factor. The second is based on **the modification of the exploration process** through the incorporation of external knowledge or the guidance of a risk metric.
>
>
> From this view of  [Garcia&Fernandez, 2015], the our submission, IPO, "End-to-End Safe Reinforcement Learning through Barrier Functions for Safety-Critical Continuous Control Tasks" $(\color{blue}{\text{W1}})$, and "Temporal Logic Guided Safe RL Using Control Barrier Functions"$(\color{blue}{\text{W2}})$
> all belong to belongs to the first categorize: learning from the optimality criterion.
>
>
> [Garcia&Fernandez, 2015] Javier Garcia and Fernando Fernandez, A Comprehensive Survey on Safe Reinforcement Learning,Journal of Machine Learning Research, 2015.
>
>
> Second, The main difference between our submission and IPO/$(\color{blue}{\text{W1}})$/$(\color{blue}{\text{W2}})$ lies in the learning method to solve safe RL. Our submission learns policy via performance bound, while IPO/$(\color{blue}{\text{W1}})$/$(\color{blue}{\text{W2}})$ update policy via  Barrier functions. Our method is parallel to Barrier functions, in the revision, we will provide some basic discussion and comparisons in the related work section.
>
>
> **Q2: "Another major concern is how the safety 'guarantee' is expressed in this paper."**
>
> **A2:" From the view of theory, the asymptotic safety guarantee (such as our submission) is weaker than non-asymptotic safety guarantee (such as CRPO). But our method is also a safe RL algorithm.
>
>
> The main difference between asymptotic safety guarantee and non-asymptotic safety guarantee lies in the following fact:
>
>
>  **non-asymptotic safety guarantee $\Rightarrow$ asymptotic safety guarantee,**
>
> but **asymptotic safety guarantee$\not\Rightarrow$ non-asymptotic safety guarantee**
>
>
> non-asymptotic safety guarantee shows a convergence rate of the algorithm learning a safe policy. While asymptotic safety guarantee only shows the safety guarantee without convergence rate, thus, our method is also with safety guarantee, see Theorem 1 and Remark 2  that provide a condition that keeps safe learning:
> $$\lim_{k\rightarrow \infty}J^{c}(\pi_k)\leq b.$$
> Thus, the our method is a safe RL algorithm.

---

> > ### Comment · Reviewer_tK1z · 2022-08-07
> > **Safety issue**
> >
> > Yeah, your method is a safe RL method. But, as you said, CRPO has stronger safety properties than this paper. My question is what's the advantage of your method when compared to CRPO? Currently, you only show that your implementation of CRPO is not working on some tasks...

---

> > > ### Author Response · Authors · 2022-08-07
> > > **Re to issue of the advantage of our method when compared to CRPO.**
> > >
> > > Thanks for your further feedback, we wish to clear all your concerns as follows.
> > >
> > > In this comment, we express the advantage of our method when compared to CRPO by the following two aspects.
> > >
> > > _____
> > >
> > > **(Part I) For the theory part, our setting is more general than CRPO. CRPO depends on 4 very strong assumptions to obtain its theory result. While our theory result does not depend on any assumption.**
> > >
> > >
> > >
> > > Although CRPO of Theorem 1 shows the non-asymptotic safety guarantee, we notice that the Theorem 1 of CRPO is only a tabular case with softmax policy. Then CRPO of Theorem 2 expends it to neural network learning case with softmax policy. However, we find CRPO uses the **linear function** for policy evaluation (see Eq.(10) in https://arxiv.org/pdf/2011.05869.pdf), which violates its neural network setting clearly.
> > >
> > > Besides, CRPO depends on some very strong assumptions. For example, its Assumption 4 requires the following condition holds:
> > > $$\mathbb{E}\left[\exp(\bar Q_t)-\dfrac{1}{C_f}\mathbb{E}[\bar Q_t]\right]\leq 1.$$
> > > Since the state space is usually very large or even infinite in the function approximation setting, we do not know how to keep this Assumption 4 holds. Similarly, its Assumption 3 shares the same condition as Assumption 4.
> > >
> > > Its Assumption 2 requires $Q$ belongs to the functional class $\mathcal F_\infty$, where $\mathcal F_\infty$ keeps the following form:
> > > $$f(s,a,\theta)=f(s,a,\theta_0)+\int\mathbb{I}(\theta^\top \psi(s,a)>0)\lambda^{\top}(\theta)\psi(s,a)\mathrm{d} p(\theta).$$
> > > But as the authors of CRPO state $\mathcal F_\infty$ needs "is a function class of neural networks with infinite width", see page 7, line 12, https://arxiv.org/pdf/2011.05869.pdf. The neural networks with infinite width is very difficult to be implemented.
> > >
> > >
> > >
> > > **While our safety analysis does not depend on such strong assumptions shown in CRPO. See our Theorem 1 and Remark 1.**
> > >
> > >
> > > _____
> > >
> > >
> > > **(Part II) For the empirical  part, our CUP is much more scalable and adaptable than CRPO.**
> > >
> > > Our CUP performs well on extensive setting with a compressive comparison. However, it is still unknown for us the CRPO could be played on the environment of Mujoco and Safety-Gym. We also have provided some insights of why CRPO can not be scalable to complex environments in the previous response, see https://openreview.net/forum?id=22hMrSbQXzt&noteId=OeWK5K5Zv- .

---

> > > > ### Comment · Reviewer_tK1z · 2022-08-08
> > > > **Response**
> > > >
> > > > Thank you for your clarification on the questions. I will update my score to a 5.
> > > > It would be great if you could make a repo for your submission code. Otherwise, the next safe RL submission may also encounter the same issue as I have with CRPO.

---

> > > > > ### Author Response · Authors · 2022-08-08
> > > > > **Response**
> > > > >
> > > > > Thanks for all your positive comments, we will take consideration to your suggestions.

---

### Official Review · Reviewer_iQjH · 2022-07-13

**Rating:** 5
**Confidence:** 3
**Soundness:** 3 good
**Presentation:** 2 fair
**Contribution:** 2 fair

**Summary:**

This paper proposes a novel strategy for safe reinforcement learning in constrained MDPs (cMDPs). At a high level, they provide a bound on performance difference that, compared to (Achiam et al., 2017), accounts for the approximation error when applying a Taylor approximation. Then, they propose a novel algorithm based on their theory, and experimentally evaluate its performance.

**Questions:**

Can you clarify the novelty compared to (Achiam et al., 2017), both in terms of the theory and the algorithm?

**Strengths And Weaknesses:**

Pros
- Important problem

Cons
- Incremental contribution compared to (Achiam et al., 2017)

The theoretical contributions in this paper appear a bit limited. Compared to prior work (Achiam et al., 2017), they are only providing guarantees on the approximation error. Intuitively, such a guarantee should follow pretty straightforwardly from Taylor’s theorem. If there are additional technical challenges that must be overcome, the authors should make this clear.

The algorithmic contribution is also unclear. They claim to propose a new algorithm that performs a projection based on their theory to ensure correctness. However, prior work (Achiam et al., 2017) uses a similar strategy. I’m wondering if the authors can clarify how their proposed approach differs from this existing strategy.

Finally, their experimental results also appear mixed. On MuJoCo, they do perform well, but there are several cases where existing baselines outperform their approach. The same issue holes for SafetyGym; here, they perform best on 3/4 environments, but CPO apparently significantly outperforms their approach on CarGoal1-v0 (though it is hard to tell since they cut off the graph).

---

> ### Author Response · Authors · 2022-07-29
> **Re "Incremental contribution compared to (Achiam et al., 2017)"  (theory part)**
>
> Thanks for your careful reviewing and valuable suggestions, we will clarify your concerns and provide a revision according your suggestions.
>
>
> Our submission improves the CPO  (Achiam et al., 2017) at the following three aspects. We have discussed the difference between our new algorithm and CPO in Appendix B and Table 2 (page 18).
>
> ${\color{red}{1).}}$From the theory party, **the result of (Achiam et al., 2017) is special case of our proposed bound, i.e., our new bound unifies the result of (Achiam et al., 2017)**. Concretely, recall Proposition 1: for any policies $\pi$ and $\pi\prime$
> $$
> J(\pi)-J(\pi^\prime)\ge\dfrac{1}{1-\tilde{\gamma}}\mathbb{E}_{s\sim d_\pi\prime,a\sim\pi(\cdot|s)}\left[A_\pi\prime^{\text{GAE}(\gamma,\lambda)}(s,a)
> -\dfrac{2\tilde{\gamma}(\gamma\lambda(|\mathcal{S}|-1)+1)\epsilon}{(1-\tilde\gamma)(1-\gamma\lambda)}D_\text{TV}(\pi\prime,\pi)[s]
> \right],~~\text{\color{blue}(EQ1)}
> $$
> where $\tilde{\gamma}=\dfrac{\gamma(1-\lambda)}{1-\gamma\lambda}$.
>
>
> Let $\lambda\rightarrow0$, then the Proposition 1 is reduced to
> $$
> J(\pi)-J(\pi^\prime)\ge\dfrac{1}{1-\gamma}\mathbb{E}_{s\sim d_\pi\prime,a\sim\pi(\cdot|s)}\left[A_\pi\prime(s,a)
> -\dfrac{2\gamma}{1-\gamma}D_\text{TV}(\pi\prime,\pi)[s]
> \right],
> $$
> which matches the result of CPO (Achiam et al., 2017, Corollary 1).
>
>
> ${\color{red}{2).}}$ **Our Theorem 1 is the first to show a bound w.r.t GAE (see ($\text{\color{blue}{EQ1}}$)) with a rigorous theoretical analysis, while CPO  (Achiam et al., 2017) only shows a bound with advantage. Additionally, we provide a fresh objective to achieve this result**.
>
>
> The refined bound ($\text{\color{blue}{EQ1}}$) contains GAE technique that significantly reduces variance while maintaining
> a tolerable level of bias empirically [Schulman et al., 2016], which implies using the bound ($\text{\color{blue}{EQ1}}$) as
> a surrogate function could improve performance potentially for practice. Although GAE has been
> empirically applied to extensive reinforcement learning tasks, to the best of our knowledge, the result
> ($\text{\color{blue}{EQ1}}$) is the first to show a rigorous theoretical analysis to extend the surrogate functions to GAE.
> About discussions and technique never appear in the work CPO  (Achiam et al., 2017).
>
> Additionally, to achieve the results in Theorem 1, we prove Proposition 4 (see Appendix E in our submission) where we propose a generalized expression of objective
> $J(\pi)$ as follows:
> $$J(\pi)=\mathbb E_{s_0\sim\rho_0(\cdot)}[\varphi(\cdot)]+\dfrac{1}{1-\gamma}\mathbb E_{s_0\sim\rho_0(\cdot)}\left[\sum_{t=0}^{\infty}\gamma^t\lambda^t\delta_t^{\varphi}\right],~~\text{\color{blue}{(EQ2)}},$$
> where $\delta_t^{\varphi}=E_{s_t\sim P_{\pi}(\cdot|s_t),a_t\sim \pi(\cdot|s_t),s_{t+1}\sim P(\cdot|s_t,a_t)}\left[r_{t+1}+\gamma\varphi(s_{t+1})-\varphi(s_{t})\right]$ is a new expected TD-error with respect to a function $\varphi$. **To the best of our knowledge, such a novel expression $J(\pi)$
>  has not appeared in any literature.** This novel expression $J(\pi)$
>  unifies many existing results, we provide a discussion as follows:
>
>
> * If $\phi=V_{\pi}$, which leads to some nice properties, e.g., $\delta_t^\varphi$
> is reduced to traditional temporal difference (TD) error (that is also called as Bellman residual term). Furthermore, let $\lambda=0$, then $J(\pi)(\text{\color{blue}{EQ2}})$
> is reduced to $J(\pi)=\mathbb E_{s\sim\rho_0}[V_{\pi}(s)]$
> that is widely used definition of objective in many existing reinforcement learning literature, e.g., we also start from this definition in our submission.
>
> * If $\lambda\ne0,\gamma\ne0$, we see that
> $J(\pi))(\text{\color{blue}{EQ2}})$ has a remarkably simple formula involving a discounted sum of TD errors, which is analogue to TD$(\lambda)$. Significantly, $J(\pi) )(\text{\color{blue}{EQ2}})$is not a simple extension TD$(\lambda)$ since those TD errors $\delta_t^\varphi$ is determined by $\varphi$. Furthermore, if $\phi=V_{\pi}$, then $\delta_t^\varphi$ is reduced to $\delta_t=\mathbb E_{s_t\sim P_{\pi}(\cdot|s_t),a_t\sim \pi(\cdot|s_t),s_{t+1}\sim P(\cdot|s_t,a_t)}\left[r_{t+1}+\gamma V(s_{t+1})-V(s_{t})\right],$ then the following term in $(\text{\color{blue}{EQ2}})$
> $$
> \dfrac{1}{1-\gamma}\mathbb E_{s_0\sim\rho_0(\cdot)}\left[\sum_{t=0}^{\infty}\gamma^t\lambda^t\delta_t^{\varphi}\right]
> $$
> reduces to an estimator of advantage $A_{\pi}$, which is the key idea of GAE. Thus, the proposed $J(\pi)(\text{\color{blue}{EQ2}})$ is general version, and GAE is only a special case if $\varphi=V_{\pi}$.
> Finally, we should emphasize that the proposed Proposition 4 plays a critical role for us to achieve Theorem 1. Concretely, with $(\text{\color{blue}{EQ2}})$, we know
> $$J(\pi)-J(\pi')=(1-\tilde\gamma)\sum_{t=0}^{\infty}\gamma^t\lambda^t \left(\sum_{s\in\mathcal{S}}d_{\pi}^{\lambda}(s)\delta^\varphi_{\pi,t}-d_{\pi'}^{\lambda}(s)\delta^\varphi_{\pi',t}\right).$$
>  This key idea of GAE has appeared in the expression $J(\pi)-J(\pi')$.

---

> > ### Comment · Reviewer_iQjH · 2022-08-07
> > **Thank you for your response**
> >
> > Thank you for your response, I will update my score to a 5

---

> ### Author Response · Authors · 2022-07-30
> **Continue to Re "Incremental contribution compared to (Achiam et al., 2017)" (algorithm part). The CUP and CPO are quite different for both learning framework and practical implementation.**
>
> $\color{red}{3).}$ **The policy update difference between CPO (Achiam et al., 2017) and CUP.** In this section, we compare CPO and our CUP from the following two aspects: Learning framework and Practical implementation.  The CUP and CPO are quite different for both learning framework and practical implementation.
>
>
> To short the expression, we introduce a notation as follows,
>
>
>
> $$
> \bar D (\pi,\pi_k)=:\mathbb{E}_{s\sim d \pi_k(\cdot) }\left[{\text{KL}}(\pi,\pi_k)\right]
> $$
>
> ____
>
> * Comparison to learning framework.
>
>
> **CPO**. **CPO (Achiam et al., 2017) uses their theory result to derive a policy improvement step that guarantees both an increase in reward and satisfaction of constraints on other costs. This step forms the basis for CPO, which computes an approximation to the theoretically-justified update.**
>
>
> Concretely, CPO (Achiam et al., 2017) uses surrogate functions to approximate the objective of CMDP ad follows,
> $$
> \pi_{k+1}=\arg\max_{\pi\in\Pi}\mathbb{E}_{s\sim d \pi_k(\cdot),a\sim \pi(\cdot|s)}\left[A^{\pi_k} (s,a)\right],
> $$
>
> $$
> \text{s.t.}~J^{c}(\pi_k)+\mathbb{E}_{s\sim d \pi_k(\cdot),a\sim \pi(\cdot|s)}\left[A^{\pi_k}_c (s,a)\right]\leq b,
> $$
>
> $$
> \text{}~~~~~~~\bar D (\pi,\pi_k)=\mathbb{E}_{s\sim d \pi_k(\cdot) }\left[{\text{KL}}(\pi,\pi_k)\right]\leq\delta.
> $$
>
> **CUP**.  **The proposed CUP (constrained update projection) first performs a policy improvement, which may produce a temporary policy violates the constraint. Then in the second step, CUP projects the policy back onto the safe region to reconcile the constraint violation.**
>
>
> Concretely, the two-step approach of performance improvement and projection is updated as follows,
>
> Step1 : **Policy Improvement**
>
>
> $$
> \pi_{k+\frac{1}{2}}=\arg\max_{\pi\in\Pi}\left(
> \mathbb{E}_{s\sim d \pi_k(\cdot),a\sim \pi(\cdot|s)}\left[A^{\text{GAE},\pi_k} (s,a)\right]-\alpha\sqrt{\bar D (\pi,\pi_k)}
> \right)
> $$
>
>
> Step2 : **Projection**
>
> $$
> \pi_{k+1}=\arg\min_{\pi\in\Pi} \bar{D}\left(\pi,\pi_{{k+\frac{1}{2}}}\right),
> $$
>
> $$
> \text{s.t}~J^{c}(\pi_k)+\frac{1}{1-\tilde \gamma}\mathbb{E}_{s\sim d \pi_k(\cdot),a\sim \pi(\cdot|s)}\left[A^{\text{GAE},\pi_k}_c (s,a)\right]+\beta\sqrt{\bar D (\pi,\pi_k)}\leq b.
> $$
>
>
>
> ___
>
>
> * Comparison to practical implementation.
>
> **CPO(Achiam et al., 2017) approximates the non-convex objective (or constraints) with first-order or second Taylor expansion, i.e., CPO uses convex approximation to optimization the non-convex function.**
>
> Concretely, CPO consider the following approximations:
>
> $$
> \dfrac{1}{1-\gamma}\mathbb{E}_{s\sim d \pi_k(\cdot),a\sim \pi(\cdot|s)}\left[A^{\pi_k} (s,a)\right]\approx(\theta-\theta_k)\nabla J(\pi),
> $$
>
> $$
> \dfrac{1}{1-\gamma}\mathbb{E}_{s\sim d \pi_k(\cdot),a\sim \pi(\cdot|s)}\left[A^{\pi_k}_c (s,a)\right]\approx(\theta-\theta_k)\nabla J^{c}(\pi),
> $$
>
>
> $$
> \bar D (\pi,\pi_k)=:\mathbb{E}_{s\sim d \pi_k(\cdot) }\left[{\text{KL}}(\pi,\pi_k)\right]\approx(\theta-\theta_k)^{\top}\mathrm{H}(\theta-\theta_k),
> $$
>
> where $\pi$ is short for $\pi_{\theta}$, $\pi_k$ is short for $\pi_{\theta_k}$, $\mathrm H$ is Hessian matrix defined as follows,
>
>
> $$
> \mathrm H [i,j]=\dfrac{\partial}{\partial \theta_i\partial \theta_j }\mathbb{E}_{s\sim d \pi_k(\cdot) }\left[{\text{KL}}(\pi,\pi_k)\right].
> $$
>
>
> Furthermore, according to convex optimization, CPO (Achiam et al., 2017) obtains a closed update rule with $\theta$ as follows,
>
>
> if approximation to CPO is feasible:
> $$\theta_{k+1}=\theta_k +\dfrac{1}{\lambda_\star}\mathrm H ^{-1}( g -\nu_{\star} a),$$
> else
> $$\theta_{k+1}=\theta_k -\sqrt{\dfrac{\delta}{a^{\top}\mathrm H^{-1}a}}\mathrm H^{-1}a,$$
> where
> $$
> a=\nabla_{\theta}\mathbb{E}_{s\sim d \pi_k(\cdot),a\sim \pi(\cdot|s)}\left[A_c^{\pi_k} (s,a)\right],
> $$
>
> $$
> g=\nabla_{\theta}\mathbb{E}_{s\sim d \pi_k(\cdot),a\sim \pi(\cdot|s)}\left[A^{\pi_k} (s,a)\right],
> $$
>
>
> $$
> \lambda_{\star}, \nu_{\star}=\arg\max_{\lambda\ge0,\nu\ge0}
> \left(
> \dfrac{-1}{2\lambda}
> \left(
> g^{\top}\mathrm H g-2\nu g^{\top}\mathrm H a
> +a^{\top}\mathrm H a \nu^2
> \right)
> +\nu(J^{c}(\pi_k)-b)-\dfrac{\lambda\delta}{2}
> \right).
> $$
>
> **Our CUP provides a non-convex implementation via only first-order optimizers, which does not require any strong approximation on the convexity to the objectives.**
>
> Considering the data $(s_t,a_t,r_{t+1})$ collected according to $\pi_{\theta_k}$, let $$\hat D_t=\dfrac{1}{T}\sum_{t=1}^{T} \mathrm{KL}(\pi_{\theta_t}(\cdot|s_t),\pi_{\theta}(\cdot|s_t)).$$
>
>
>
>
> The CUP updates the parameter as follows,
> $$
> \theta_{k+\frac{1}{2}}=\arg\max_{\theta}
> \left(
> \dfrac{1}{T}\sum_{t=1}^{T}\dfrac{\pi_\theta(a_t|s_t)}{\pi_{\theta_k}(a_t|s_t)}\hat{A}_t-\alpha\sqrt{\hat D_t}
> \right);
> $$
>
> $$
> \theta_{k+1}=\arg\min_{\theta}
> \dfrac{1}{T}\sum_{t=1}^{T}
> \left(
> \mathrm{KL}\Big(\pi_{\theta_{k+\frac{1}{2}}}(\cdot|s_t),\pi_{\theta}(\cdot|s_t)\Big)
> +\nu_{k}\dfrac{\pi_\theta(a_t|s_t)}{\pi_{\theta_k}(a_t|s_t)}\hat A_t^{c}
> \right);
> $$
>
> $$
> \nu_{k+1}=\left(\nu_k+\eta(\hat J_k -b)\right)_{+},
> $$
> where $\eta>0$.

---

### Author Response · Authors · 2022-08-02
**Revison**

Thanks for all the careful reviewing and valuable suggestions from the reviewers, we provide a revision according to all the reviewers.

* In the revision, we provide some additional experiments for performance with respect to initial $\eta$ and $\nu$, see Appendix H.5, page 45-46;

* In the revision, we rewrite the Remark 1, where we remove original Remark 1 and provide a discussion from the aspect of the comparison to CPO, see Section 3.2, page 5;

* In the revision, we provide the insights on how the optimize the objective function via the first-order method, see page 6, line 195-202;

* In the revision, we provide some additional discussion to three work w.r.t. barrier function, see page 7, line 210-213;

* Finally, we also polish some sections.

---

### Meta-Review · Area_Chair_DRJh · 2022-08-27

**Recommendation:** Accept
**Confidence:** Certain

**Metareview:**

This paper studies safe reinforcement learning and proposes a new policy optimization method with a rigorous safety guarantee. During the author-reviewer discussion period, the reviewers' concerns were mostly resolved. The reviewers have reached the consensus that the contribution of the proposed method is sufficient to be borderline accepted. I recommend it for acceptance and suggest the authors incorporate the reviewers' comments into the final version.

**Award:**

No

---

### Decision · Program_Chairs · 2022-09-14

Accept